# On the nature of the earliest known lifeforms

**Dheeraj Kanaparthi[1,2,3]\*, Frances Westall[4], Marko Lampe[5], Baoli Zhu[2,6], Thomas Boesen[7], Bettina Scheu[3], Andreas Klingl[8], Petra Schwille[1], Tillmann Lueders[2]\***

[1]Max-Planck Institute for Biochemistry, Munich, Germany; [2]Chair of Ecological Microbiology, BayCeer, University of Bayreuth, Bayreuth, Germany; [3]Earth and environmental sciences, Ludwig Maximilian University, Munich, Germany; [4]CNRS-Centre de Biophysique Moléculaire, Orléans, France; [5]Advanced Light Microscopy Facility, European Molecular Biology Laboratory, Heidelberg, Germany; [6]Key Laboratory of Agro-Ecological Processes in Subtropical Regions, Taoyuan Agroecosystem Research Station, Institute of Subtropical Agriculture, Chinese Academy of Sciences, Changsha, China; [7]Department of Biosciences, Center for Electromicrobiology, Aarhus, Denmark; [8]Department of Botany I, Ludwig Maximilian University, Munich, Germany

**\*For correspondence:**
dheerajbio@gmail.com (DK);
Tillmann.Lueders@uni-bayreuth.
de (TL)

**Competing interest:** The authors declare that no competing interests exist.

Reviewing Editor: George
H Perry, Pennsylvania State
University, United States

## eLife Assessment

This provocative manuscript presents **important** comparisons of the morphologies of Archaean bacterial microfossils to those of microbes transformed under environmental conditions that mimic those present on Earth during the same Eon. The evidence in support of the conclusions is **solid**. The authors' environmental condition selection for their experiment is justified.

**Abstract** Microfossils from the Paleoarchean Eon are the oldest known evidence of life. Despite their significance in understanding the history of life on Earth, any interpretation of the nature of these microfossils has been a point of contention among researchers. Decades of back-and-forth arguments led to the consensus that reconstructing the lifecycles of Archaean Eon organisms is the most promising way of understanding the nature of these microfossils. Here, we transformed a Gram-positive bacterium into a primitive lipid vesicle-like state and studied it under environmental conditions prevalent on early Earth. Using this approach, we successfully reconstructed morphologies and life cycles of Archaean microfossils. In addition to reproducing microfossil morphologies, we conducted experiments that spanned years to understand the process of cell degradation and how Archaean cells could have undergone encrustation of minerals (in this case, salt), leading to their preservation as fossilized organic carbon in the rock record. These degradation products strongly resemble fossiliferous features from Archaean rock formations. Our observations suggest that microfossils aged between 3.8–2.5 Ga most likely were liposome-like protocells that have evolved physiological pathways of energy conservation but not the mechanisms to regulate their morphology. Based on these observations, we propose that morphology is not a reliable indicator of taxonomy in these microfossils.

## Introduction

The Pilbara Greenstone Belt (PGB), Western Australia, and Barberton Greenstone Belt (BGB), South Africa, are the oldest known sedimentary rock successions that have not undergone significant metamorphic alterations (*Shields, 2007*; *Zuilen, 2019*). Hence, these rock formations have been the subject of numerous scientific investigations focused on understanding the biology and biogeochemistry of Archaean Earth (*Shields, 2007*; *Zuilen, 2019*; *Sugitani et al., 2015a*; *Hickman-Lewis and Westall, 2021*; *Oehler et al., 2017*). Over a span of 50 years, these studies have documented various organic structures within these rock formations that resemble fossilized cells and their degradation products, with δ¹³C composition consistent with biologically derived organic carbon (*Walsh, 1992*; *Schopf et al., 2018*). Although these observations suggest that these organic structures were fossil remnants of Archaean microorganisms, any such interpretation, together with the biological origin of these structures, has been a point of contention among researchers (*Schopf and Kudryavtsev, 2012*; *Wacey et al., 2016*; *Wacey et al., 2018a*; *Gee, 2002*). Two factors currently limit wider acceptance of their biological origin – the absence of truly analogous microfossil morphologies among extant prokaryotes and an indication of an ongoing biological process, like cell division, among most microfossils (*Hickman-Lewis and Westall, 2021*; *Knoll and Barghoorn, 1977*). Moreover, most of the described microfossils are larger than present-day prokaryotes and often exhibit considerable cytoplasmic complexity with intracellular alveolar structures (*Sugitani et al., 2015a*; *Oehler et al., 2017*; *Lepot et al., 2013*). These complex morphologies and relatively larger cell sizes of supposedly primitive Archaean Eon cells are not in accordance with our current understanding of how biological complexity evolved through Darwinian evolution (*Adami et al., 2000*; *Wolf et al., 2018*).

Apart from the chemical and δ¹³C-biomass composition (*Lepot et al., 2013*; *Malaterre et al., 2023*; *Wacey et al., 2014*; *Hickman-Lewis et al., 2016*), one key emphasis of the studies arguing for and against the biological origin of Archaean Eon organic structures involves an extensive morphological comparison with extant prokaryotes or abiotically formed minerals (*Schopf and Kudryavtsev, 2012*; *Wacey et al., 2016*; *Schopf, 1993*; *Wacey et al., 2018b*). Cell morphology among extant organisms is maintained by a plethora of intracellular processes and is determined by the information encoded in their genome (*Adams and Errington, 2009*). In our opinion, drawing parallels between the present-day prokaryotes and Archaean Eon organisms inherently involves subscribing to the notion that paleo-Archaean life forms possess all the complex molecular biological mechanisms to regulate their morphology as present-day cells. Any such presumptions are not in tune with the current scientific consensus of how life could have originated on early Earth (*Adami et al., 2000*; *Westall et al., 2018*; *Szostak, 2017*). It is now widely believed that life evolved in the form of protocells devoid of most molecular biological complexity (*Szostak, 2017*). These primitive cells are thought to have undergone slow Darwinian evolution, resulting in present-day cells with intricate intracellular processes (*Lane and Martin, 2010*; *Lane, 2014*). Given the unlikelihood of Archaean cells possessing complex molecular biological processes, we test the possibility that complex morphologies of Archaean microfossils result from the complete absence of intracellular mechanisms regulating their morphology (*Oparin, 1969*).

To test this hypothesis, we used a top-down approach of transforming a Gram-positive bacterium (*Exiguobacterium Strain-Molly*) into a primitive lipid vesicle-like state (*EM-P*). Cells in this state can be described as a simple sack of cytoplasm devoid of all mechanisms to regulate their morphology and reproduction. Although it has not been empirically demonstrated, some studies have suggested that cells in this lipid vesicle-like state may resemble primitive protocells (*Errington, 2013*; *Kanaparthi et al., 2023*; *Kanaparthi et al., 2024*). Given that the reproduction of such cells is shown to be influenced by environmental conditions (*Kanaparthi et al., 2023*; *Kanaparthi et al., 2024*), we studied the life cycle of these cells under experimental conditions resembling the native environment of the Archaean microfossils.

While the precise environmental conditions of early Earth remain uncertain, a growing consensus within the scientific community suggests that surface temperatures on Archaean Earth ranged between 26° and 35 °C (*Knauth, 2005*; *Knauth, 1998*; *Catling and Zahnle, 2020*). Moreover, most, if not all, of the known microfossils from the Archaean Eon are restricted to coastal marine environments (*Walsh, 1992*; *Hickman-Lewis et al., 2019*). Coastal marine environments often exhibit higher salinity due to the constant evaporation of seawater. To replicate the high salinities of the coastal marine environments, *EM-P* was cultivated in half-strength tryptic soy broth supplemented with 7% (w/v) Dead

Sea Salt (TSB-DSS) at 30 °C. We chose Dead Sea Salt over pure NaCl to better emulate complex salt compositions of natural environments.

Given that *EM-P*'s life cycle and the biophysical basis of such a reproduction process is extensively discussed in the previous paper (*Kanaparthi et al., 2024*), the primary focus of this manuscript will be restricted to the morphological comparison of *EM-P* cells and Archaean Eon microfossils. Below, we present the morphological comparison between *EM-P* cells and Archaean Eon microfossils. In addition to this morphological comparison, we also conducted experiments that spanned years (18–28 months) to understand the process of protocell degradation, how they become encrusted in salt, and how they are preserved as fossilized organic carbon in the rock record.

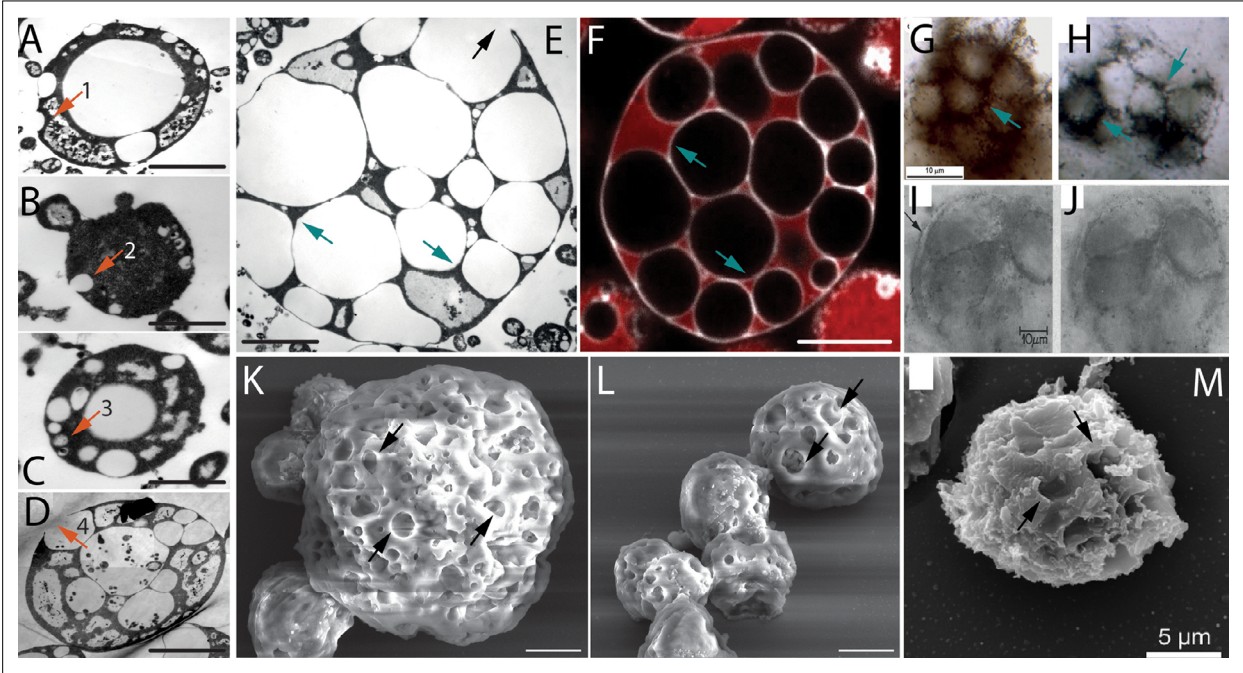

**Figure 1.** Morphological comparison of the Apex Chert and the Strelley Pool Formation microfossils with *EM-P*. Images (**A-D**) are TEM images of *EM-P* cells forming intracellular vesicles (ICVs) and intracellular daughter cells. The numbered arrows in these images point to different stages of ICV formation (see *Figure 1—figure supplement 1 and 2*). Images (**E, F, K & L**) show TEM, SEM, and STED microscope images of *EM-P* cells with ICVs and surface depressions (black arrows). Cells in image F were stained with universal membrane stain, FM5-95 (red), and DNA stain, PicoGreen (green). Images (**G-J & M**) are spherical microfossils reported from the Apex Chert and the Strelley Pool Formation, respectively (originally published by *Schopf and Packer, 1987*; *Delarue et al., 2020*; *Schopf and Packer, 1987*; *Delarue et al., 2020*). Cyan arrows in images (**E-H**) point to cytoplasm sandwiched between large hollow vesicles. The arrow in the image I points to the dual membrane enclosing the microfossil. Morphologically similar images of *EM-P* cells are shown in *Figure 1—figure supplement 3*. Black arrows in images (**K-M**) point to surface depressions in both *EM-P* and the Strelley Pool Formation microfossils, possibly formed by the rupture of ICVs as shown in (**D & E**) (arrows) (also see *Figure 1—figure supplements 4–6*). Scale bars: (**A-D**) (0.5 µm) (**E, K, & L**) (2 µm), and 5 µm (**F**).

The online version of this article includes the following figure supplement(s) for figure 1:

**Figure supplement 1.** *EM-P* cells with intracellular vesicles (ICV).

**Figure supplement 2.** Variation in the morphology of *EM-P* cells and intracellular vesicles.

**Figure supplement 3.** Morphological comparison of the Apex Chert microfossils with *EM-P*.

**Figure supplement 4.** Morphological comparison between *EM-P* and the Strelley Pool Formation (SPF) microfossils.

**Figure supplement 5.** Morphological comparison between *EM-P* and Strelley Pool Formation SPF microfossils.

**Figure supplement 6.** Morphological comparison between *EM-P* and SPF microfossils.

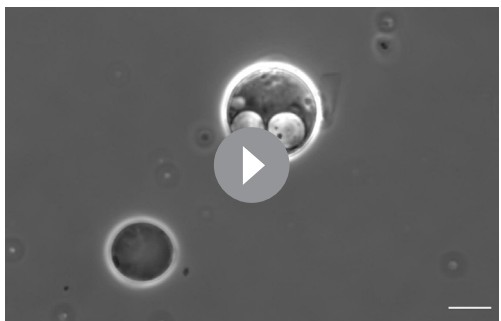

**Video 1.** Video shows an *EM-P* cell with an intracellular daughter cell within its vesicle. Scale bar: 5 μm.
https://elifesciences.org/articles/98637/figures#video1

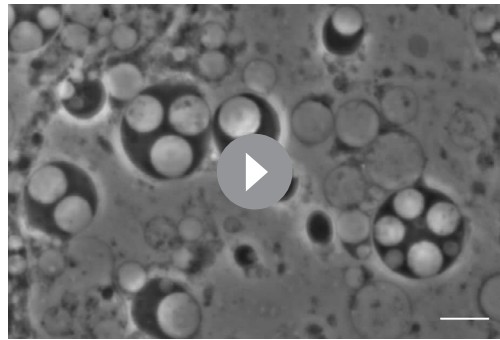

**Video 2.** Videos show a gradual increase in the number of daughter cells within *EM-P*s intracellular vesicles and a corresponding decrease in the volume of the cell cytoplasm. Scale bar: 5 μm.
https://elifesciences.org/articles/98637/figures#video2

## Results

### Morphological comparison of top-down modified cells with fossilized Archaean cells

When cultured under experimental conditions likely resembling coastal marine environments of the Paleoarchean Eon, *EM-P* exhibited cell sizes that were an order of magnitude larger than their original size. They also exhibited complex morphologies and reproduced by a relatively less understood process (*Kanaparthi et al., 2023*; *Kanaparthi et al., 2024*). The life cycle of these cells involves reproduction by two methods – via forming internal or a string of external daughter cells (*Kanaparthi et al., 2024*). *EM-P* reproducing by both these processes bears close morphological resemblance to microfossils reported from the Archaean Eon.

The first step in reproduction by intracellular daughter cells is the formation of hollow intracellular vesicles (*Figure 1A*). These vesicles were formed by a process that resembles endocytosis (*Figure 1A*). A similar process of vesicle formation was previously reported in protoplasts (*Kanaparthi et al., 2024*; *Kapteijn et al., 2022*). Over time, the number of intracellular vesicles (ICVs) within *EM-P* gradually increased (*Figure 1A–F*). No uniformity was observed in the size of ICVs within a cell or the number of ICVs among different cells (*Figure 1E, F*, *Figure 1—figure supplements 1 and 2*). *EM-P* cells with such intracellular vesicles resemble spherical microfossils reported from 3.46 billion-year-old (Ga) Apex chert (*Schopf and Packer, 1987*). Like the Apex-chert microfossils, ICVs of *EM-P* were hollow, and organic carbon (cytoplasm) in these cells is restricted to spaces between the vesicles (*Figure 1F–K* & *Figure 1—figure supplement 3*).

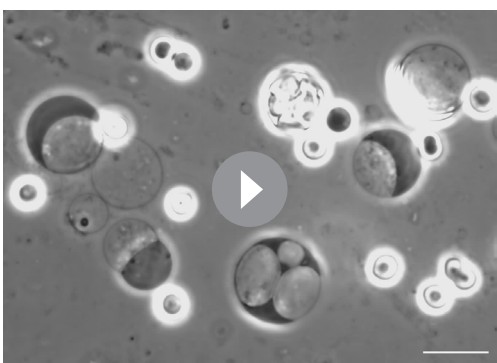

**Video 3.** Movies show a gradual increase in the number of daughter cells within *EM-P*s intracellular vesicles and a corresponding decrease in the volume of the cell cytoplasm. Scale bar: 5mm.
https://elifesciences.org/articles/98637/figures#video3

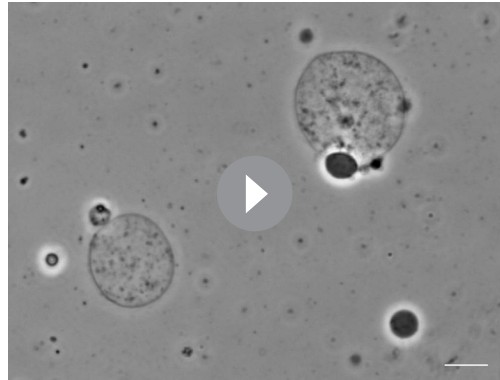

**Video 4.** Movies show a gradual increase in the number of daughter cells within EM-Ps intracellular vesicles and a corresponding decrease in the volume of the cell cytoplasm. Scale bar: 5mm.
https://elifesciences.org/articles/98637/figures#video4

The three-dimensional STED and SEM images of *EM-P* show numerous surface depressions (*Figure 1K*, arrow & *Figure 1—figure supplement 1E*). Such depressions are formed either during vesicle formation (*Figure 1A and B*, arrow) or by the rupture of intracellular vesicles attached to the cell membrane (*Figure 1D, E*, *Figure 1—figure supplement 1G-K*, *Figure 1—figure supplements 4–6*). *EM-P* cells with such surface depressions and intracellular vesicles strongly resemble morphological resemblance to microfossils reported from 3.4 Ga Strelley Pool Formation (SPF) microfossils (*Delarue et al., 2020*; *Figure 1M*, *Figure 1—figure supplements 4–6*). Microfossils reported from other sites, such as the Farrel Quartzite (*Appendix 1—figure 1*; *Retallack et al., 2016*), Turee Creek (*Appendix 1—figure 2*; *Sugitani et al., 2007*), and the Fig Tree Formations (*Appendix 1—figure 3*; *Schopf and Barghoorn, 1967*), likely are morphological variants of Archaean *EM-P*-like cells and the Apex Chert microfossils. For instance, *EM-P* with many but relatively smaller intracellular vesicles resembles the Fig Tree microfossils, both in cell size and shape (*Appendix 1—figure 3*). On the other hand, *EM-P,* with a relatively larger number of intracellular vesicles squeezed into polygonal shapes, resembles microfossils reported from the SPF (*Figure 1G–I* & *Figure 1—figure supplements 4–6*), the Farrel Quartzite microfossils with polygonal alveolar structures (*Appendix 1—figure 1*) and the Turee Creek microfossils (*Appendix 1—figure 2*; *Retallack et al., 2016*).

The second step in this method of reproduction involves the formation of daughter cells into the ICVs (*Appendix 1—figure 4*; *Kanaparthi et al., 2024*). Daughter cells were formed in the intracellular vesicles by a process resembling budding (*Appendix 1—figure 4*). Over time, these bud-like daughter cells detached from the vesicle wall and were released into the ICV (*Video 1*). Due to a gradual loss of cytoplasm to the daughter cells, we observed a gradual reduction of the cytoplasmic volume of the parent *EM-P* cells and a corresponding increase in the number of daughter cells within the ICVs (*Appendix 1—figure 5*, *Videos 2–4*). Over time, *EM-P* cells transformed into hollow vesicles with multiple tiny daughter cells (*Appendix 1—figure 5E-H*, *Videos 3 and 4*).

These intracellular daughter cells were released into the surroundings by a two-step process. In the first step, cells underwent lysis to release the ICVs (*Figure 2*, *Appendix 1—figure 6A and B* and *Videos 5–9*). In the second step, the vesicle membrane underwent lysis (*Appendix 1—figure 6C-E*) to release the daughter cells (Appendix 1—figure 6F–H). *EM-P* cells undergoing this method of reproduction closely resemble fossilized microbial cells discovered from several Archaean Eon rock formations (*Figure 2*, *Figure 2—figure supplements 1–5*, & *Appendix 1—figures 7–12*; *Delarue et al., 2020*; *Sugitani et al., 2013*; *Barlow and Van Kranendonk, 2018*). For instance, microfossils reported from the Mt. Goldsworthy Formation exhibit cells with ICVs containing daughter cells, cells that underwent lysis to release these ICVs (*Figure 2*, *Figure 2—figure supplements 1–5*), and the subsequent rupture of the vesicle membrane and release of the daughter cells (*Figure 2—figure supplement 2D–F*; *Kanaparthi et al., 2024*).

The microfossils reported from sites like the Farrel Quartzite (*Appendix 1—figure 1*; *Retallack et al., 2016*), the Strelley Pool Formation (*Appendix 1—figures 7–9*), the Waterfall locality (*Appendix 1—figures 10 and 11*; *Sugitani et al., 2013*; *Sugitani et al., 2015b*; *Wacey et al., 2011*), the Turee Creek (*Appendix 1—figure 12*; *Barlow and Van Kranendonk, 2018*), and Dresser Formation (*Appendix 1—figures 13–17*; *Wacey et al., 2018a*), bear close morphological resemblance with morphologies of *EM-P* cells reproducing by this process. *EM-P* cells exhibited all the distinctive features of the Dresser formation microfossils, like the presence of hollow regions within the cell (*Appendix 1—figure 14*) and discontinuous or thick-porous cell walls (*Appendix 1—figures 13 and 16*). The step-by-step transformation of *EM-P* cells into these morphologies is shown in *Appendix 1—figure 16*. Due to the lysis and release of daughter cells, most late stationary growth phase *EM-P* cells were deflated with numerous surface depressions. The morphology and surface texture of such deflated *EM-P* cells resemble the morphologies of microfossils reported from the Kromberg Formations (*Appendix 1—figure 18*; *Kaźmierczak and Kremer, 2019*).

Reproduction by external daughter cells happens by two different processes. Tiny daughter cells initially appeared as buds attached to the cell membrane (*Appendix 1—figures 19 and 20*, *Video 10*). These buds subsequently grew in size and detached from the parent cell. Depending on the size of the daughter cells (buds), *EM-P* cells appear to have been reproducing either by budding or binary fission. *EM-P* cells that appear to have been reproducing by budding resemble microfossils reported from the North Pole formations (*Appendix 1—figure 20*). As observed in our incubations, microfossils from this site are a mix of individual spherical cells, spherical cells with pustular protuberances,

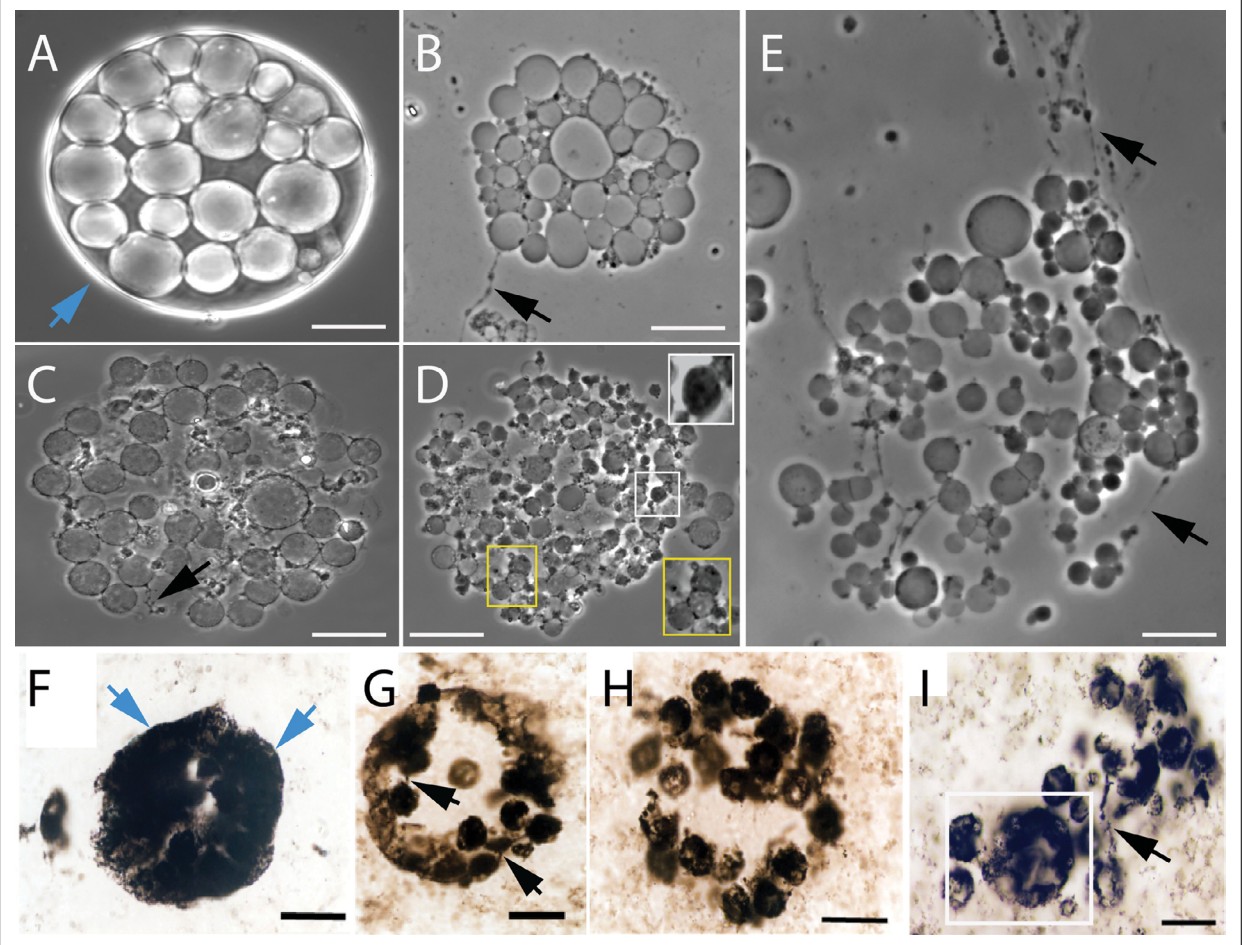

**Figure 2.** Morphological comparison between the Mt. Goldsworthy microfossilsand *EM-P*. Images (**A-E**) show the process of cell lysis and release of intracellular vesicles in *EM-P*. Image (**A**) shows an intact cell with intracellular vesicles. Images (**B-E**) show lysis and gradual dispersion of these vesicles. Insert in image (**D**) shows enlarged images of individual ICVs. Images (**F-I**) show spherical microfossils reported from the Mt. Goldsworthy formation (originally published by *Sugitani et al., 2009*). The arrow in this image, (**A & F**), points to a cell surrounded by an intact membrane. The black arrow in these images points to filamentous extensions connecting individual vesicles. The boxed region in images (**D & I**) highlights a similar discontinuous distribution of organic carbon in ICVs and microfossils. Also see *Figure 2—figure supplements 1–5*. Scale bars: 20 μm (**A–E**) & 20 μm (**F–I**).

The online version of this article includes the following figure supplement(s) for figure 2:

**Figure supplement 1.** Morphological comparison between *EM-P* and the Mt. Goldsworthy Formation microfossils.

**Figure supplement 2.** Morphological comparison between *EM-P* and the Mt. Goldsworthy Formation microfossils.

**Figure supplement 3.** Morphological comparison between *EM-P* and the Mt. Goldsworthy Formation microfossils.

**Figure supplement 4.** Mrphological comparison between *EM-P* and the Mt. Goldsworthy Formation microfossils.

**Figure supplement 5.** Morphological comparison between *EM-P* and the Mt. Goldsworthy Formation microfossils.

cells with bud-like structures, and cells undergoing binary fission (*Appendix 1—figure 20*). Other *EM-P* morphotypes, like individual spherical cells, hourglass-shaped cells undergoing fission, and cells in dyads, bear close morphological resemblance to microfossils reported from both the Swartkopie (*Appendix 1—figure 21*) and the Sheba formations (*Appendix 1—figures 22 and 23*; *Knoll and Barghoorn, 1977*; *Hickman-Lewis et al., 2019*; *Homann, 2019*). Like the Sheba Formation micro-fossils, the cells undergoing binary fission were observed to be in close contact with extracellular organic carbon (clasts of organic carbon in the Sheba Formation). This extracellular organic carbon likely represents the intracellular constituents released during the lysis of cells, as described above (*Videos 5–8* & *Appendix 1—figure 23*).

In some cases, the above-described buds did not detach from the cell surface but transformed into long tentacles (*Videos 11 and 12*). These initially hollow tentacles (*Kanaparthi et al., 2024*) gradually

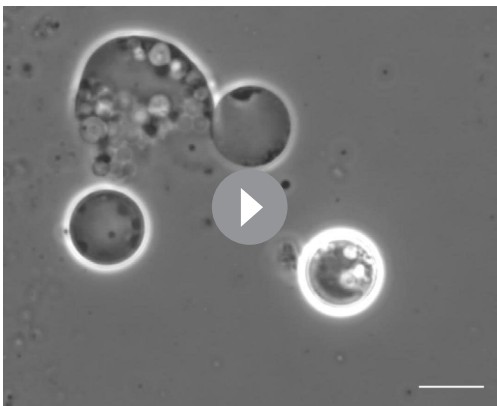

**Video 5.** Videos show sequential stages involved in *EM-P* cell lysis and the release of intracellular vacuoles containing daughter cells. Videos also show the formation of cell debris during the release of ICVs. Scale bar: 5 µm.

https://elifesciences.org/articles/98637/figures#video5

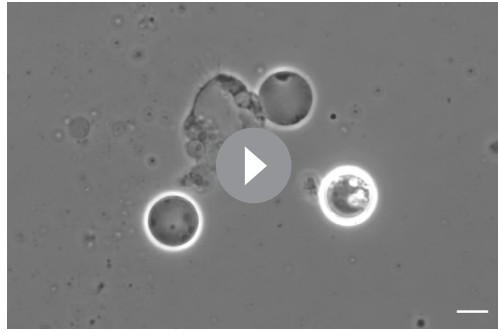

**Video 6.** Movie show EM-P cell lysis and the release of intracellular vacuoles containing daughter cells. Movie also show the formation of cell debris during the release of ICVs. Movies 6-9 show the sequential stages involved in the lysis and the release of intracellular vacuoles containing daughter cells. Scale bar: 5mm.

https://elifesciences.org/articles/98637/figures#video6

received cytoplasm from the parent cells and gradually transformed into 'string-of-spherical daughter cells' (*Figure 3*, *Appendix 1—figures 20A-C and 22*; *Kanaparthi et al., 2024*). Subsequently, these filaments detached from the parent cell, and due to the constant motion of daughter cells within these filaments, the 'string-of-spherical daughter cells' fragmented into smaller and smaller strings and ultimately into multiple individual daughter cells (*Videos 13–15*). Apart from the cells that received cytoplasm (daughter cells), we also observed hollow spherical structures within these tentacles that did not receive cytoplasm from the parent cells (*Figure 3F and G*, black arrows).

All *EM-P* morphotypes observed undergoing this reproduction process bear close morphological resemblance to microfossils reported from the Cleaverville Formation (*Figure 3*). All distinctive features of the Cleaverville microfossils, like the arrangement of cells as pairs within a string

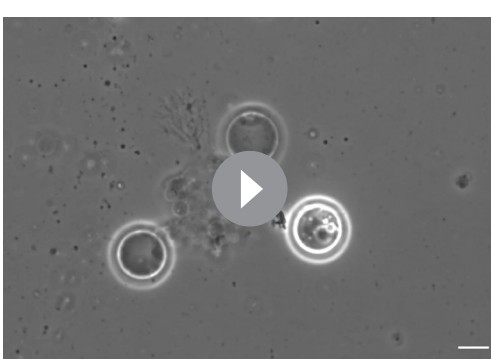

**Video 7.** Movie show EM-P cell lysis and the release of intracellular vacuoles containing daughter cells. Movie also show the formation of cell debris during the release of ICVs (long strings of membrane debris). Movies 6-9 show the sequential stages involved in the lysis and the release of intracellular vacuoles containing daughter cells. Scale bar: 5mm.

https://elifesciences.org/articles/98637/figures#video7

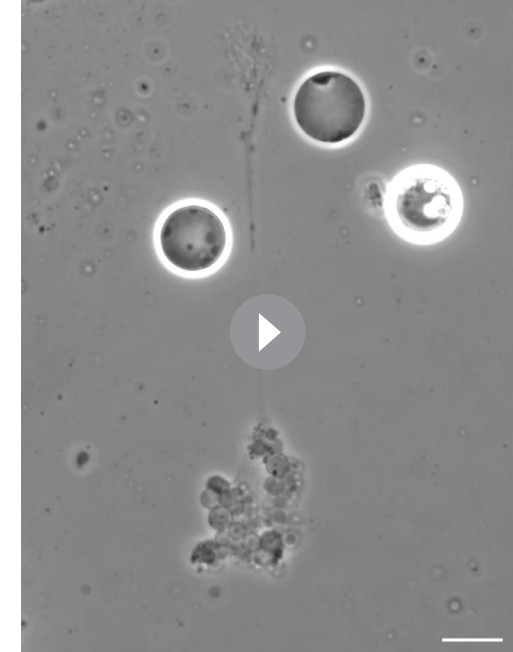

**Video 8.** Movie show EM-P cell lysis and the release of intracellular vacuoles containing daughter cells. Movie also show the formation of cell debris during the release of ICVs (long strings of membrane debris). Movies 6-9 show the sequential stages involved in the lysis and the release of intracellular vacuoles containing daughter cells. Scale bar: 5mm.

https://elifesciences.org/articles/98637/figures#video8

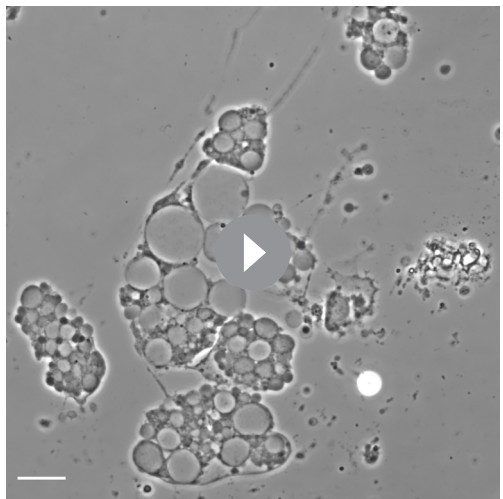

**Video 9.** Movie show EM-P cell lysis and the release of intracellular vacuoles containing daughter cells. Movie also show the formation of cell debris during the release of ICVs (long strings of membrane debris). Movies 6-9 show the sequential stages involved in the lysis and the release of intracellular vacuoles containing daughter cells. Scale bar: 5mm.

https://elifesciences.org/articles/98637/figures#video9

**Video 10.** The video shows cell debris of *EM-P* cells with tiny daughter cells attached to them. Scale bar: 5 μm.

https://elifesciences.org/articles/98637/figures#video10

(*Figure 3H, F and G*), microfossils with a discontinuous layer of organic carbon at the cell periphery (*Figure 3E, D and I–K*), were also observed in *EM-P*. Several hollow spherical structures devoid of organic carbon were reported from the Cleaverville formation (*Figure 3*). As in *EM-P*, these structures could have been the hollow membranous structures that didn't receive the cytoplasm from the parent cells. Similar structures were also reported from other microfossil sites, such as the organic structures reported from the Onverwacht Group (*Appendix 1—figure 24*; *Walsh, 1992*).

In addition to the Cleaverville microfossils, *EM-P* cells in our incubations also resemble filamentous structures with spherical inclusions reported from the Sulphur Spring Formations (*Appendix 1—figures 25–27*; *Wacey et al., 2014*). Based on the morphological similarities, we propose that these structures could have been the filamentous extensions with spherical daughter cells observed in *EM-P*. Similar but smaller filamentous structures were reported from the Mt. Grant Formation (*Appendix 1—figure 28*; *Sugitani et al., 2007*) and could have

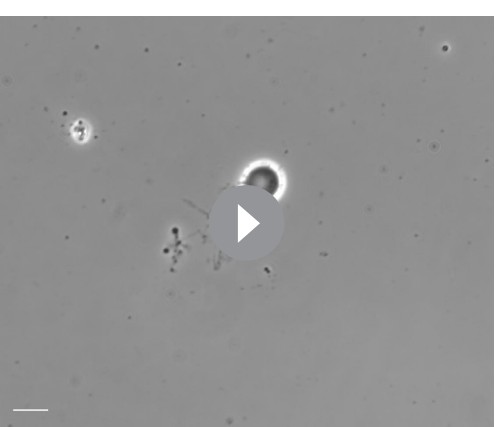

**Video 11.** Movie along with 10-12 show the sequential stages involved in the formation such individual daughter cells and their subsequent transformation into a string-of-daughter cells. Scale bar: 5mm.

https://elifesciences.org/articles/98637/figures#video11

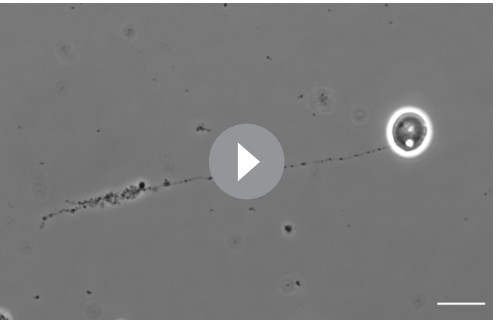

**Video 12.** Movie along with 10-12 show the sequential stages involved in the formation such individual daughter cells and their subsequent transformation into a string-of-daughter cells. Scale bar: 5mm.

https://elifesciences.org/articles/98637/figures#video12

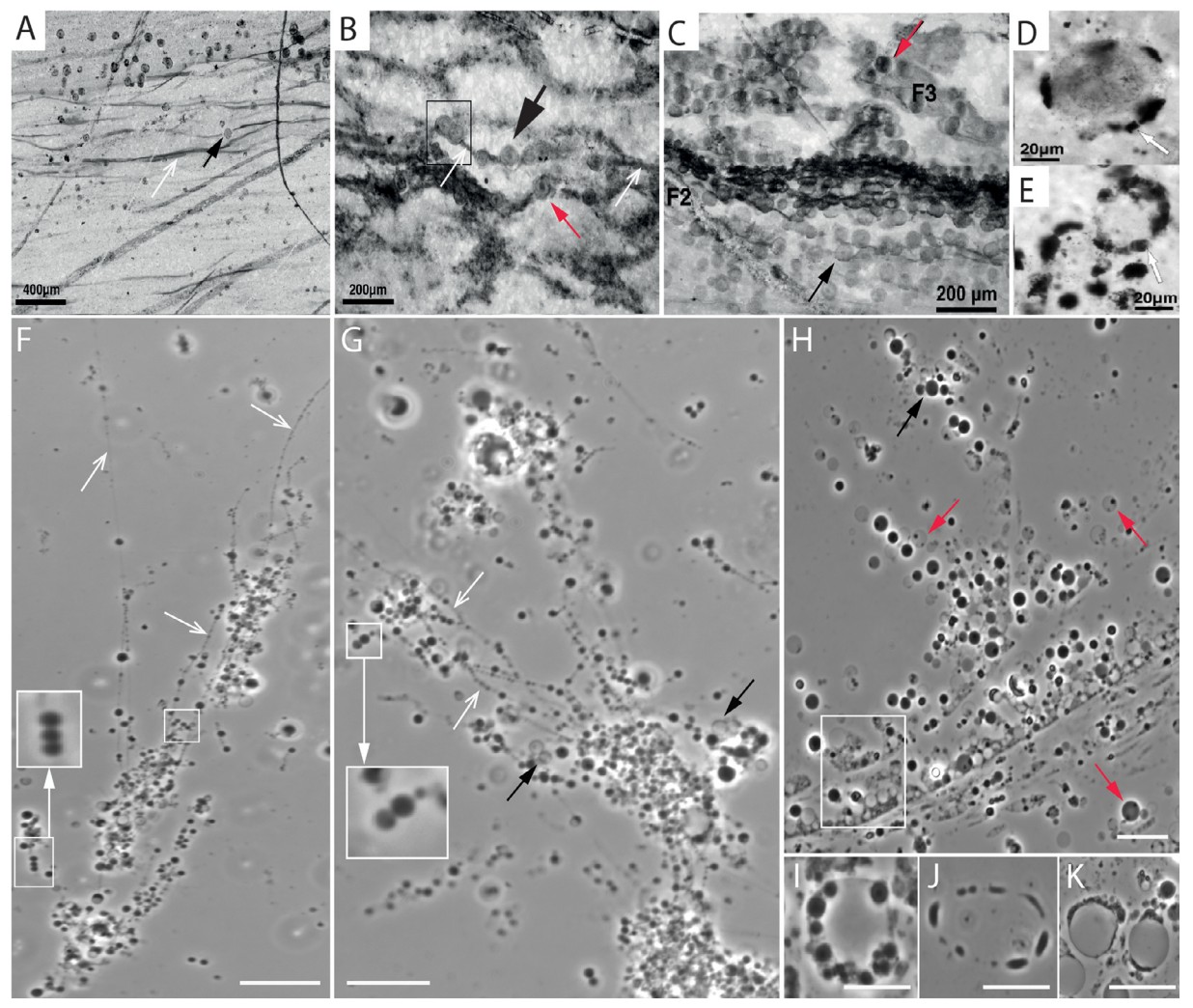

**Figure 3.** Morphological comparison of the Cleaverville microfossils with *EM-P*. Images (**A-E**) are the microfossils reported from Cleaverville formation (originally reported by *Ueno et al., 2006*). Images (**F-K**) are the *EM-P* cells morphologically analogous to the Cleverville Formation microfossils. Open arrows in images (**A, B, F & G**) point to the membrane tethers connecting the spherical cells within the filamentous extensions. Red arrows in the images point to the cells that have a similar distribution of organic carbon within the cells. Boxed and magnified regions in images (**B, F & G**) highlight the arrangement of cells in the filaments in pairs. The boxed region in image (**H**) highlights the cluster of hollow vesicles in *EM-P* incubations similar to the hollow organic structures in the Cleverville Formation, as shown in image (**C**). Images (**D, E**), and (**I-J**) show spherical cells that were largely hollow with organic carbon (cytoplasm) restricted to discontinuous patches at the periphery of the cell. Scale bars: 20 μm (**F–K**).

been the shorter fragments of similar 'strings-of-daughter cells' (*Video 14*). The spherical structures with sparsely distributed organic carbon reported from the Sheba Formations could also have been such cells undergoing fragmentation into smaller and smaller filaments (*Appendix 1—figure 22*). In tune with this proposition, the Sheba Formation microfossils, like the *EM-P* cells, exhibited an ununiform distribution of organic carbon within the cells and filamentous overhangs (*Appendix 1—figure 22*, *Video 16*).

Five to seven days after the start of the experiment, most cells in the incubations were the daughter cells. We observed three distinct types of daughter cells – a string of daughter cells (*Figure 3*, *Appendix 1—figure 28* & *Video 14*), daughter cells that were still attached to the membrane debris of the parent cell (*Figures 4 and 5*, *Figure 5—figure supplements 1–9*, & *Video 17*) and individual daughter cells (*Figure 3h*, *Video 15*). All these daughter cell morphotypes resemble a cluster of tiny spherical globules reported from the SPF (*Appendix 1—figure 7*), a string of daughter cells reported from Mt. Grant formation (*Appendix 1—figure 28*; *Sugitani et al., 2007*; *Sugitani et al., 2013*), and spherical daughter cells still attached to membrane debris of parent cell-like the ones reported

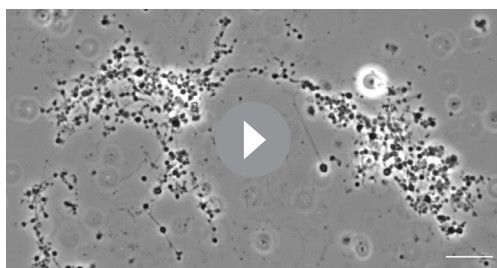

**Video 13.** Movie along with movies 14 & 15 show sequential stages involved in the detachment and fragmentation of these strings of daughter cells into individual daughter cells. Scale bar: 5mm.
https://elifesciences.org/articles/98637/figures#video13

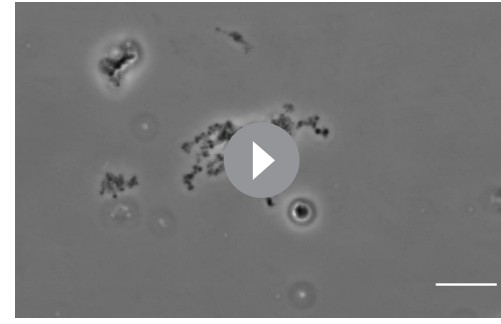

**Video 14.** Movie along with movies 13 & 15 show sequential stages involved in the detachment and fragmentation of these strings of daughter cells into individual daughter cells. Scale bar: 5mm.
https://elifesciences.org/articles/98637/figures#video14

from the Sulphur Spring Formations, Mt. Goldsworthy, the Farrel Quartzite, the Moodies Group, the Dresser Formation, and the SPF (*Figure 4*, *Figure 5—figure supplements 1–9*; *Wacey et al., 2018a*; *Sugitani et al., 2009*; *Sugitani et al., 2013*; *Wacey et al., 2011*; *Homann et al., 2018*).

## The gradual transformation of *EM-P* cells into lamination-like structures and their comparison with Archaean organic structures

Fossilization and preservation of individual cells are considered unlikely due to the absence of rigid structures. However, recent studies indicate such a process could happen under favorable environmental conditions (*Orange et al., 2009*; *Orange et al., 2011*). However, the prevailing observations suggest that a significant portion of cell biomass undergoes taphonomic alteration and is preserved as degraded organic matter. To understand the possibility of *EM-P*-like cells forming structures similar to those observed in Archaean rocks, we studied the morphological transformation of individual *EM-P* cells and biofilms over 12–30 months. Below, we present the step-by-step transformation of individual cells into organic structures and their morphological resemblance to Archaean organic structures.

*EM-P* grew in our incubations as a biofilm at the bottom of the culture flask (*Figure 6* & *Appendix 1—figure 29*). The rapid biofilm formation by *EM-P* can be attributed to the presence of extracellular DNA released during the lysis of *EM-P* cells (*Appendix 1—figure 30*; *Kanaparthi et al., 2024*). DNA released by such processes is known to promote biofilm formation (*Gödeke et al., 2011*). Over time, increased cell numbers resulted in biofilms comprising multiple layers of closely packed individual spherical cells (*Figure 6* & *Appendix 1—figure*

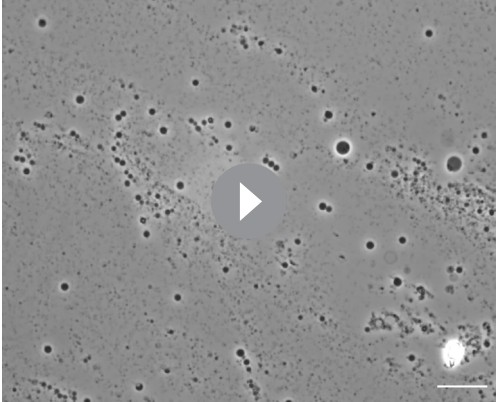

**Video 15.** Movie along with movies 13 & 14 show sequential stages involved in the detachment and fragmentation of these strings of daughter cells into individual daughter cells. Scale bar: 5mm.
https://elifesciences.org/articles/98637/figures#video15

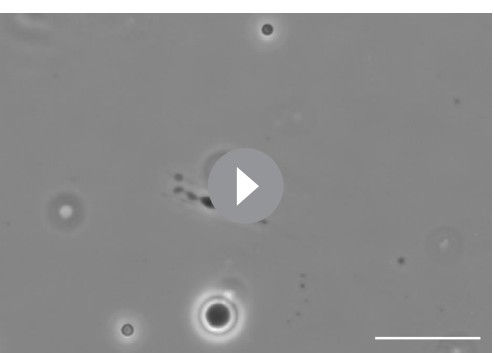

**Video 16.** Movie show the constant movement of the daughter cells; such surface undulations should have provided the kinetic energy required for the fragmentation of strings of daughter cells into individual daughter cells. Scale bar: 10mm.
https://elifesciences.org/articles/98637/figures#video16

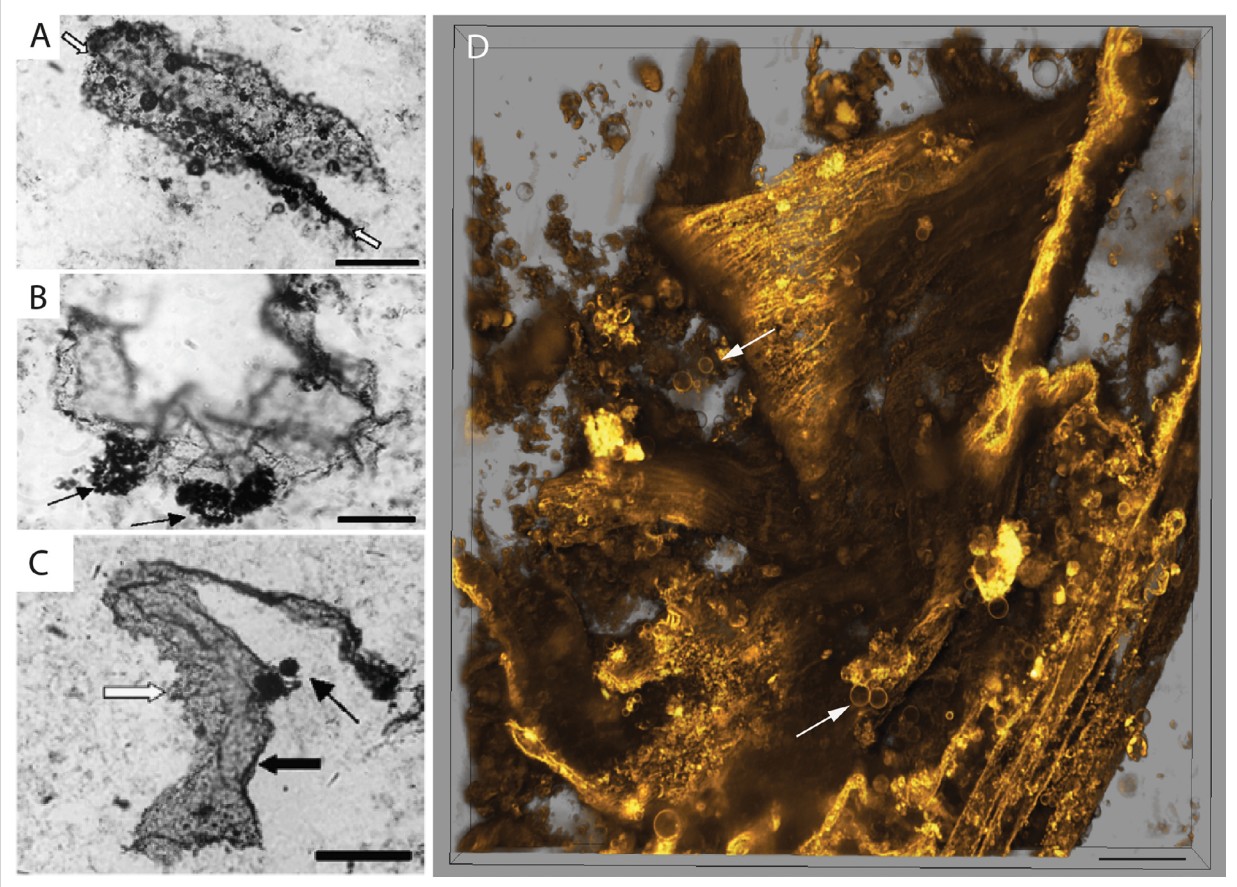

**Figure 4.** Morphological comparison between *EM-P* and the Mt. Goldsworthy microfossils. Images (**A, B & C**) are organic structures reported from the Mt. Goldsworthy Formation (**Sugitani et al., 2007**; **Sugitani et al., 2007**). Image (**D**) shows morphologically analogous film-like membrane debris observed in *EM-P* incubations. Arrows in images A-D point to either clusters or individual spherical structures attached to these film-like structures. Scale bar: 50 µm (**A–C**) & 10 µm (**D**).

*30*). A subsequent increase in the number of cells led to lateral compression and the transformation of spherical cells into a polygonal shape (*Figure 6*). By the late stationary growth phase, most cells underwent such a transformation, resulting in a honeycomb-like biofilm. The step-by-step transformation of individual spherical cells into these structures is shown in *Figure 6*. Morphologically similar organic structures were reported from several paleo-Archaean sites, like the North Pole Formation (*Appendix 1—figures 31–42*; *Buick, 1990*). A similar taphonomic degradation of organic matter associated with a biofilm was demonstrated by previous studies (*Westall et al., 2006*, Figure 10B; *Westall et al., 2006*).

Large aggregations of spherical cells devoid of internal organic carbon were reported from the North Pole Formation (*Buick, 1990*; *Appendix 1—figure 31*). These structures closely resemble the aggregations of hollow ICVs released after the lysis of *EM-P* cells (*Figure 2A–E* & *Appendix 1—figure 11*). As observed in *EM-P*, the distribution of organic carbon in the North-Pole Formation microfossils is restricted to the periphery of the spherical cells (*Appendix 1—figure 32*). Along with the morphological and organizational similarities, *EM-P* also exhibited all the accessory structures associated with the North Pole formation microfossils, such as the large clots of organic carbon (*Appendix 1—figure 31* arrows), filamentous structures originating from the spherical cells, and spherical clots of organic carbon within these filamentous (*Appendix 1—figures 33–35*; *Buick, 1990*). Based on the similarities, we propose that the large clots of organic carbon would have been the membrane debris formed during the lysis and the release of intracellular vesicles (*Appendix 1—figure 31*, *Videos 5–9*). As observed in *EM-P*, the filamentous structures associated with the microfossils could also have formed during the release of intracellular vesicles (*Videos 5–9*). The organic carbon clots within the filamentous structures could have been fossilized daughter cells (*Appendix 1—figures 33 and 34*, arrows).

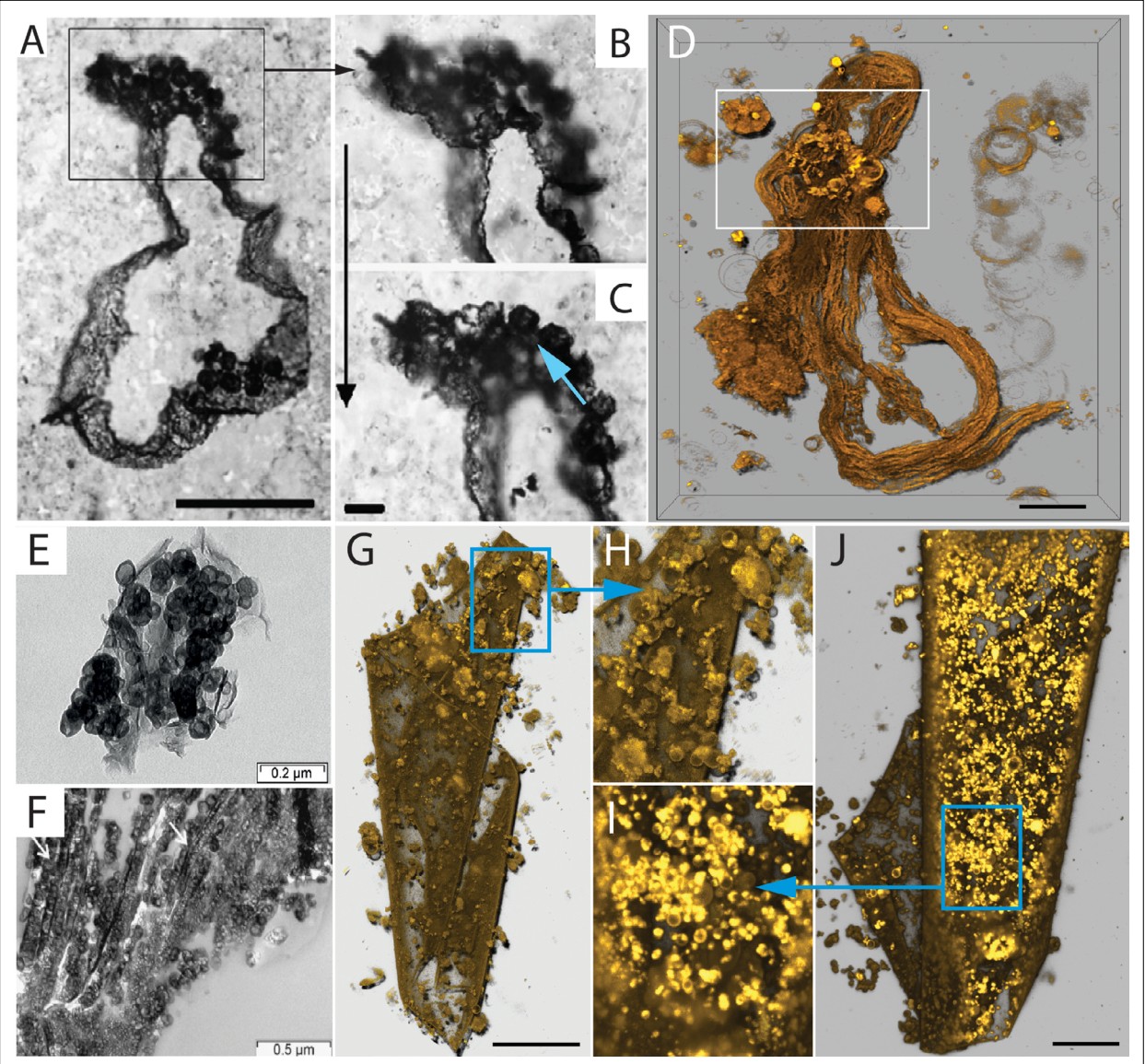

**Figure 5.** Morphological comparison of the Mt. Goldsworthy and the Sulphur Spring microfossils with *EM-P*. Image (**A-C**) are microfossils reported from the Mt. Goldsworthy Formation (*Sugitani et al., 2007*). Image (**D**) is the 3D-rendered STED microscope images of morphologically analogous membrane debris of *EM-P* cell with attached daughter cells (highlighted region) (also see *Figure 5—figure supplement 1*). Images (**E & F**) are microfossils reported from the Sulphur Spring site (*Duck et al., 2007*), showing spherical structures attached to membrane debris. Images (**G & J**) are the morphologically analogous structures observed in *EM-P* incubations. Images (**H & I**) show the magnified regions of (**G & J**) showing spherical *EM-P* daughter cells attached to membrane debris (also see *Figure 5—figure supplements 1–9*, *Video 17*). Cells and membrane debris in these images were stained with the membrane stain FM5-95 (yellow). Scale bars: A (50 μm), (**G & J**) (20 μm).

The online version of this article includes the following figure supplement(s) for figure 5:

**Figure supplement 1.** Morphological comparison between the Sulphur Spring Formation microfossils and *EM-P*.

**Figure supplement 2.** Morphological comparison between *EM-P* and the Mt. Goldsworthy microfossils.

**Figure supplement 3.** Morphological comparison between *EM-P* and the Mount Grant microfossils.

**Figure supplement 4.** Morphological comparison between *EM-P* and the SPF organic structures.

**Figure supplement 5.** Morphological comparison of the SPF microfossils with *EM-P*.

**Figure supplement 6.** Morphological comparison between *EM-P* membrane debris and SPF membrane debris.

**Figure supplement 7.** Morphological comparison between *EM-P* and the Farrel Quartzite film-like structures.

**Figure supplement 8.** Morphological comparison between *EM-P* and the Moodies Group microfossils.

**Figure supplement 9.** Morphological comparison of organic structures reported from the Dresser Formation with *EM-P*.

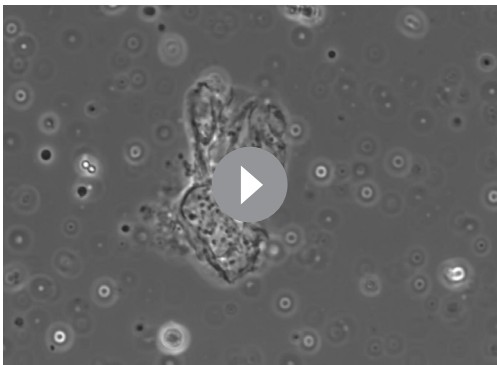

**Video 17.** The video shows cell debris of *EM-P* cells with tiny daughter cells attached to them. Scale bar: 5 µm.

https://elifesciences.org/articles/98637/figures#video17

Apart from the aggregations of hollow spherical cells, honeycomb-like structures were also reported from several microfossil sites, like the SPF, the Nuga Formation, the Buck Reef Chert, the Moodies Group, and the Turee Creek formations (*Figure 6*, *Appendix 1—figures 37–42*; *Barlow and Van Kranendonk, 2018*; *Gamper et al., 2012*; *Schopf et al., 2017*; *Tice and Lowe, 2006*). As observed in *EM-P*, these structures could have been formed by the lateral compression of cells or hollow vesicles within the biofilm (*Figure 6*). In tune with our proposition, Archaean honeycomb-like structures are often closely associated with spherical *EM-P*-like cells (*Figure 6*).

Spherical microfossils from the Pilbara and Barberton Greenstone Belts were often discovered within layers of organic carbon (*Homann, 2019*; *Homann et al., 2018*; *Tice and Lowe, 2004*; *Tice, 2009*). Over a period of 2–6 months, we observed cells in our incubations gradually being enclosed by membrane debris. These structures were formed by a multi-step process (*Appendix 1—figure 43*). First, *EM-P* grew as multiple layers of cells within a biofilm (*Appendix 1—figure 43A*). Second, the lysis of these cells led to the formation of a considerable amount of membrane debris (*Appendix 1—figures 43 and 44*, *Video 17* and *Video 18*). Subsequently, this membrane debris coalesced to form large fabric-like structures (*Appendix 1—figure 45*). These membrane fabrics were then expelled from the biofilm (*Appendix 1—figure 43D and E*, *Video 18*). Over time, these expelled membrane fabrics grew in surface area to form a continuous layer of membrane enclosing a large population of cells (*Appendix 1—figures 45–49* & *Video 19*). This fabric-like membrane debris enclosing biofilms observed in *EM-P* incubations bear close morphological resemblance to microfossils reported from Chinaman Creek in the Pilbara (*Appendix 1—figure 49*) and Mt. Goldsworthy Formation (*Figures 4 and 5*; *Sugitani et al., 2007*; *Sugitani et al., 2009*; *Brasier et al., 2005*).

Parallel layers of organic carbon termed laminations were reported from several Archaean microfossil sites (*Hickman-Lewis et al., 2019*; *Homann et al., 2018*; *Tice and Lowe, 2004*; *Homann et al., 2015*). Structures similar to these laminations were observed in our incubations. As described above, the reproduction in *EM-P* involves the lysis of cells to facilitate the release of the intracellular daughter cells, resulting in a considerable amount of cell debris (*Videos 9 and 17*). The parallel layers of organic carbon in our incubations (*Figures 7 and 8*) are formed by lysis and collapse within individual biofilm layers (*Appendix 1—figure 29*). Another way the organic carbon layers could have formed is by the lateral compression of honeycomb-like biofilms (*Appendix 1—figures 51–54*). Sequential steps resulting in the formation of such structures are shown in *Appendix 1—figure 52*. Such layers of cell debris closely resemble different types of laminated structures reported from the Barberton Greenstone Belt and Pilbara Iron Formations, like the α and β laminations (*Figures 7 and 8* & *Appendix 1—figures 49–54*; *Homann et al., 2018*; *Tice and Lowe, 2004*; *Tice, 2009*).

Also similar to the Archaean laminations, we observed layers of cell debris in our incubations have lenticular gaps (*Figure 8*, *Figure 8—figure supplements 2–4* & *Appendix 1—figures 49–55*). Within these lenticular gaps, we observed intact *EM-P* cells or honeycomb patterns, suggesting that lenticular gaps within otherwise uniformly parallel laminations were formed due to non-uniform lysis or incomplete deflation of cells within individual layers of *EM-P* cells (*Figure 7*, *Figure 8—figure supplements 2–4* & *Videos 20 and 21*). Although in the case of Archaean laminations, these lenticular gaps were thought to have been formed by the entrapment of air bubbles (*Homann et al., 2018*), based on our results, we argue that there could have been more than one way such structures could have formed. Other distinctive features of the lamination, like raised mounds or swirls (*Appendix 1—figures 57–59*; *Homann et al., 2018*; *Tice, 2009*), were also observed in batch cultures of *EM-P*. Given these morphological similarities, we propose that some of the laminated and other diaphanous filamentous structures could have been formed by the cell debris of the *EM-P*-like cells that inhabited these sites during the Archaean Eon. We will discuss these possibilities in more detail below.

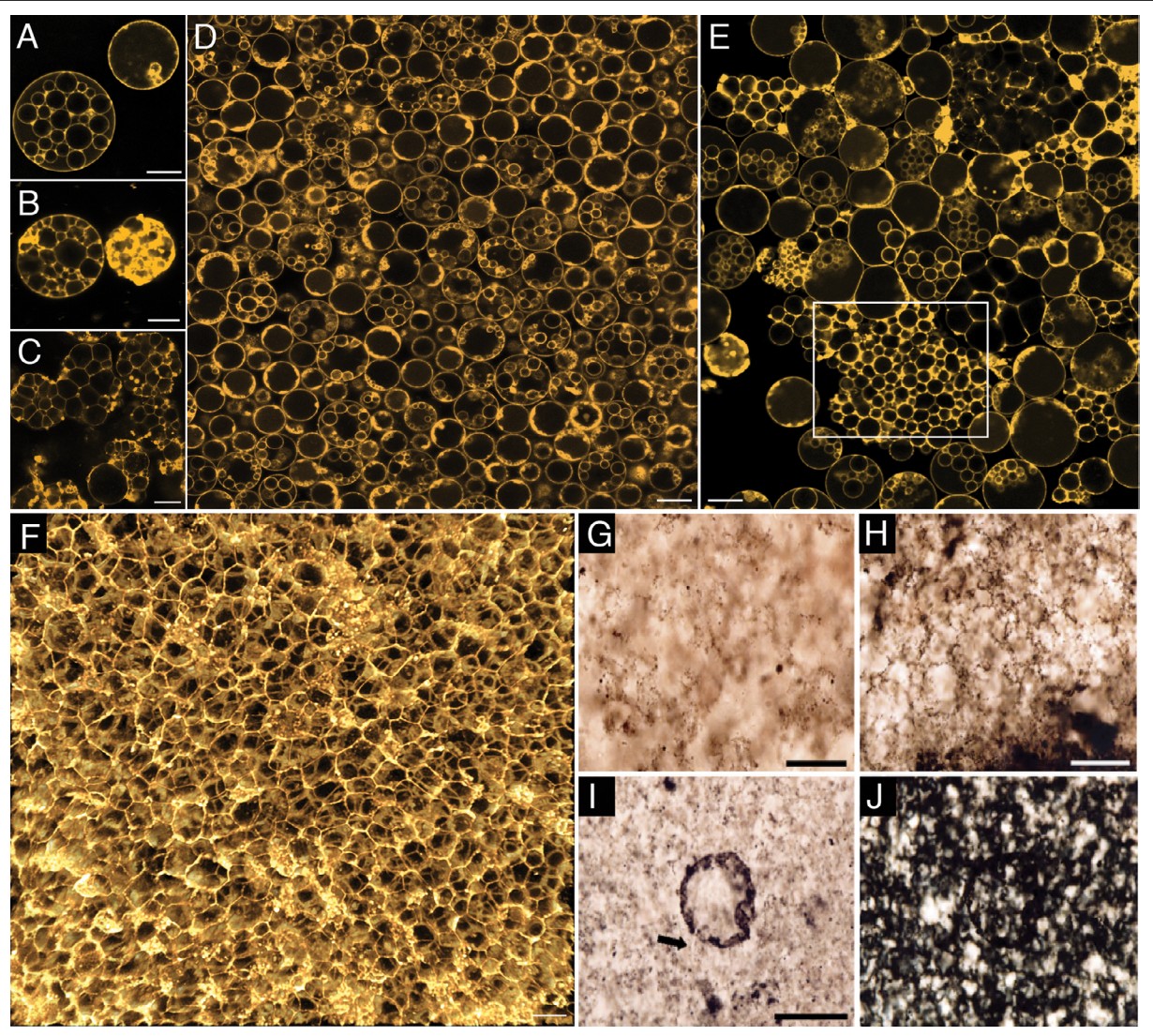

**Figure 6.** Sequential steps involved in the formation of honeycomb-shaped mats. Images (**A-C**) show single *EM-P* cells that gradually transformed from spherical cells with intracellular vesicles into honeycomb-like structures. Images (**D-E**) show a similar transformation of biofilms composed of individual spherical cells into honeycomb-like structures. Cells in these images are stained with membrane stain, FM5-95 (red), and imaged using a STED microscope. Images (**G-J**) are the microfossils reported from the SPF (originally published by *Sugitani et al., 2007*). Scale bars: (**A-F**) (10 µm), (**G & H**) (20 µm), and **I** (50 µm).

Over a period of 3–12 months, we observed the biofilms solidifying into a solid crust (*Appendix 1— figures 60 and 61*). The SEM-EDX characterization of these solidified biofilms showed the presence of potassium and magnesium minerals on the surface, suggesting that these structures were formed by the gradual adsorption of positively charged cations on the negatively charged biofilms (*Appendix 1— figure 60*). Most Archaean microfossils are restricted to coastal marine environments. Compared to open oceans, these coastal marine environments harbor higher concentrations of salt due to higher evaporation rates. Hence, these microfossils could have undergone a similar encrustation process as observed in our incubations. Moreover, solidified *EM-P* biofilms resemble the mineral-encrusted structures reported from the Kromberg Formation (*Appendix 1—figure 61*; *Westall et al., 2001*). Like the Kromberg Formations structures, solidified *EM-P* biofilms are composed of desiccation cracks and spherical cells beneath the surface (*Appendix 1—figure 61*).

When these salt-encrusted cells were transferred into fresh media, we observed a gradual increase in cell numbers (results not shown). However, given that these cells are encrusted in a layer of salts, we observed early growth-phase cells breaking out of the thick salt crust, resulting in stellar morphologies

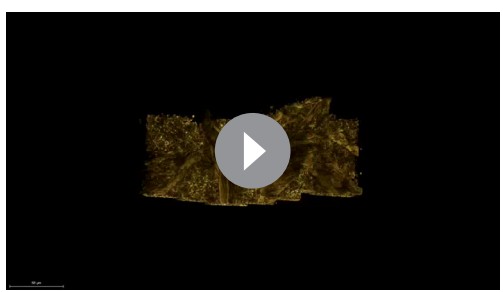

**Video 18.** The video shows cell debris being formed within the biofilm. The lateral view of the biofilm shows the membrane debris being pushed out of the biofilm. Cells in this movie are stained with membrane stain FM5-95 and imaged using a STED microscope.
https://elifesciences.org/articles/98637/figures#video18

(*Appendix 1—figures 62 and 63*). The observed morphologies of these cells closely resemble the morphologies of microfossils reported from the Strelley Pool and other North Pole cherts (*Appendix 1—figures 62 and 63*; *Sugitani et al., 2013*; *Buick, 1990*). All distinctive features of these microfossils, like the stellar-shaped cells undergoing binary fission and a string of daughter cells extending out of such stellar cells (*Video 22*), were also observed in our incubations.

## Discussion

Advances in microscopic (FIB-SEM) and analytical (NanoSIMS) techniques over the past few decades have facilitated better imaging and precise determination of chemical and isotopic compositions of microfossils (*Lepot et al., 2013*; *Brasier et al., 2015*). Nevertheless, there is considerable disagreement among researchers regarding the interpretation of this information (*Wacey et al., 2016*). Given the importance of morphology in determining the biogenicity and taxonomic affiliation of the microfossils, reconstructing the lifecycles of Archaean Eon organisms is considered crucial to understanding the nature of these microfossils (*Sugitani et al., 2009*). Our study is the first to reconstruct all known spherical microfossil morphologies and their lifecycles from extant bacteria. Furthermore, we have shown that many of the taphonomic structures observed in our study closely resemble the controversial structures observed in rocks of the Palaeo-Mesoarchaean age (3.6–3.0 Ga) and even in the Neoarchaean (3.0–2.4 Ga). These similarities help us answer long-standing questions regarding the origin and the nature of Archaean microfossils.

The nature of Archaean organic structures is currently being debated among researchers (*Schopf et al., 2018*; *Wacey et al., 2016*; *Wacey et al., 2018a*). While some studies suggest that these structures could be remnants of Archaean microorganisms, others suggest that they may have been abiotic minerals that formed due to volcanic activity (*Wacey et al., 2018a*). The argument for this proposition is based on the fact that these organic structures share more similarities with inorganic mineral structures than with extant prokaryotes. To establish the biogenicity of a microfossil, it is essential to either find a convincing morphological analog among extant bacteria or establish a biogenic process through which they are formed (*Sugitani et al., 2015a*). The biogenicity of microfossils reported from several sites like the Swartkoppoie Formation, Kitty's Gap Chert, and the Josephdal Formation is widely accepted among the scientific community due to the discovery of spherical microfossils in different stages of their lifecycle (*Appendix 1—figure 21*; *Knoll and Barghoorn, 1977*; *Westall et al., 2006*; *Westall et al., 2001*). However, such a step-by-step biological process through which Archean Eon organic structures could have formed has never been demonstrated empirically.

The biological origin of microfossils reported from several sites, like the Dresser Formation, to date, remains a matter of debate (*Wacey et al., 2018a*). Several morphological features of these organic structures, like the presence of organic carbon only at the periphery, the absence of internal cell constituents, the presence of pyrite and silicate minerals inside the cells, and the presence of a thick porous or discontinuous cell wall, were all argued as claims for their abiotic origin (*Wacey et al., 2018a*). Justifiably, these morphological features have never been observed in any living organism. Nevertheless, all spherical

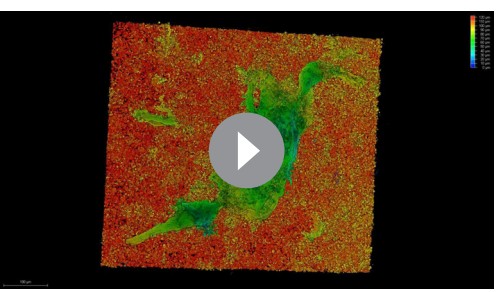

**Video 19.** The video shows a layer of *EM-P* cells and membrane debris that was formed over this layer. Several spherical cells can be seen attached to the membrane debris and in the free space between the cell layer and the wavy membrane. Membrane debris was often noticed to have lenticular gaps. Cells in this movie are stained with membrane stain FM5-95 and imaged using a STED microscope.
https://elifesciences.org/articles/98637/figures#video19

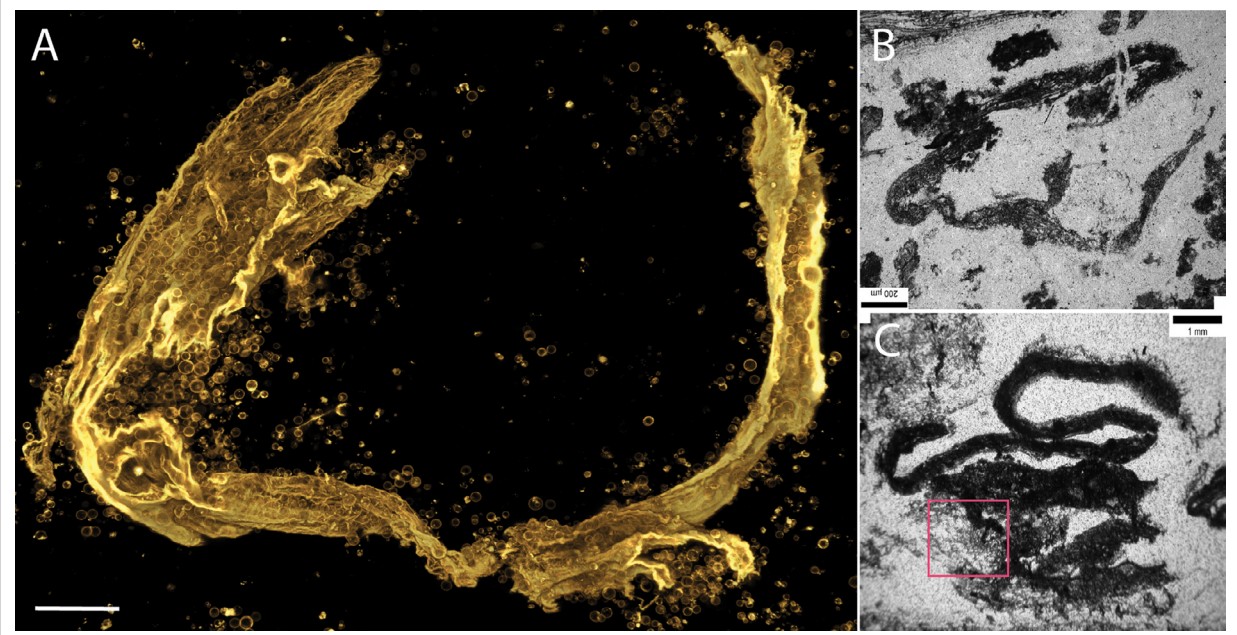

**Figure 7.** Morphological comparison of the Buck Reef Chert b-laminations with *EM-P's* membrane debris. Image (**A**) shows a 3D-rendered image of *EM-P's* membrane debris. Cells in the image are stained with membrane stain Nile red and imaged using a STED microscope. Images (**B & C**) show β-type laminations reported from Buck Reef Chert (originally published by *Tice, 2009*). The boxed region in image-a highlights the membrane-forming rolled-up structures containing spherical daughter cells, as described in the case of BRC organic structures. Scale bars: 50 μm.

microfossils reported from the Dresser Formation resemble *EM-P* cells, especially those with a single large ICV (*Appendix 1—figures 13–16*). What was thought to have been a thick, porous cell wall in the microfossils could have been the cytoplasm with tiny ICVs sandwiched between the cell and vesicle membrane (*Appendix 1—figures 13–16*). Similarly, the hollow cells with a discontinuous cell wall could either have been the ICVs released by cell lysis or the late-growth stage cells with little cytoplasm (*Appendix 1—figure 15*). In such cells, the presence of cytoplasm is restricted to discontinuous patches around the periphery of the cells (*Appendix 1—figure 15*). The sequence of steps leading to these cell morphologies that resemble the Dresser Formation microfossils is shown in *Appendix 1—figure 16*. A closer inspection of the Dresser Formation microfossils shows the ICVs membrane rupture and the daughter cell release (*Appendix 1—figure 17*). Morphologies indicating this method of reproduction among microfossils are not unique to the Dresser Formation. Microfossils with similar morphological features were reported from sites like the Strelley Pool, the Waterfall region, and Mt. Goldsworthy Formation (*Figures 1 and 2*; *Figure 2—figure supplements 1–5*; *Sugitani et al., 2015a*; *Sugitani et al., 2009*). These similarities suggest that microfossil morphologies observed in the Dresser Formation are in tune with other microfossils of similar geological time periods, suggesting their biological origin.

Spherical structures half-coated with pyrite are reported from the Dresser Formation (*Appendix 1—figure 14*; *Wacey et al., 2018a*). These structures could have been the iron-reducing *EM-P*-like cells with hollow ICV constituting half their volume. The selective co-localization of pyrite and carbon could be explained by the Fe(III) reduction happening at the cell surface. The Fe(II) produced from this metabolic reaction could have reacted with environmental sulfide, converted to insoluble pyrite, and precipitated onto the cell surface. Given the absence of this metabolic process within the hollow ICV, these structures remained pyrite-free (*Appendix 1—figure 14*). In addition to the Dresser Formation, organic structures coated with pyrite have also been reported from other microfossil sites like the Sulphur Spring Formations (*Wacey et al., 2014*). The selective presence of pyrite on these microfossils could also be explained by a similar mechanism (*Appendix 1—figures 25–27*). Apart from pyrite, minerals like anatase were reported to have been present within the cells (*Wacey et al., 2018a*). The presence of anatase within these cells could be explained by the transport of these minerals into the cells during the ICV formation (*Figure 1B&D*). ICVs are formed by a process similar to endocytosis,

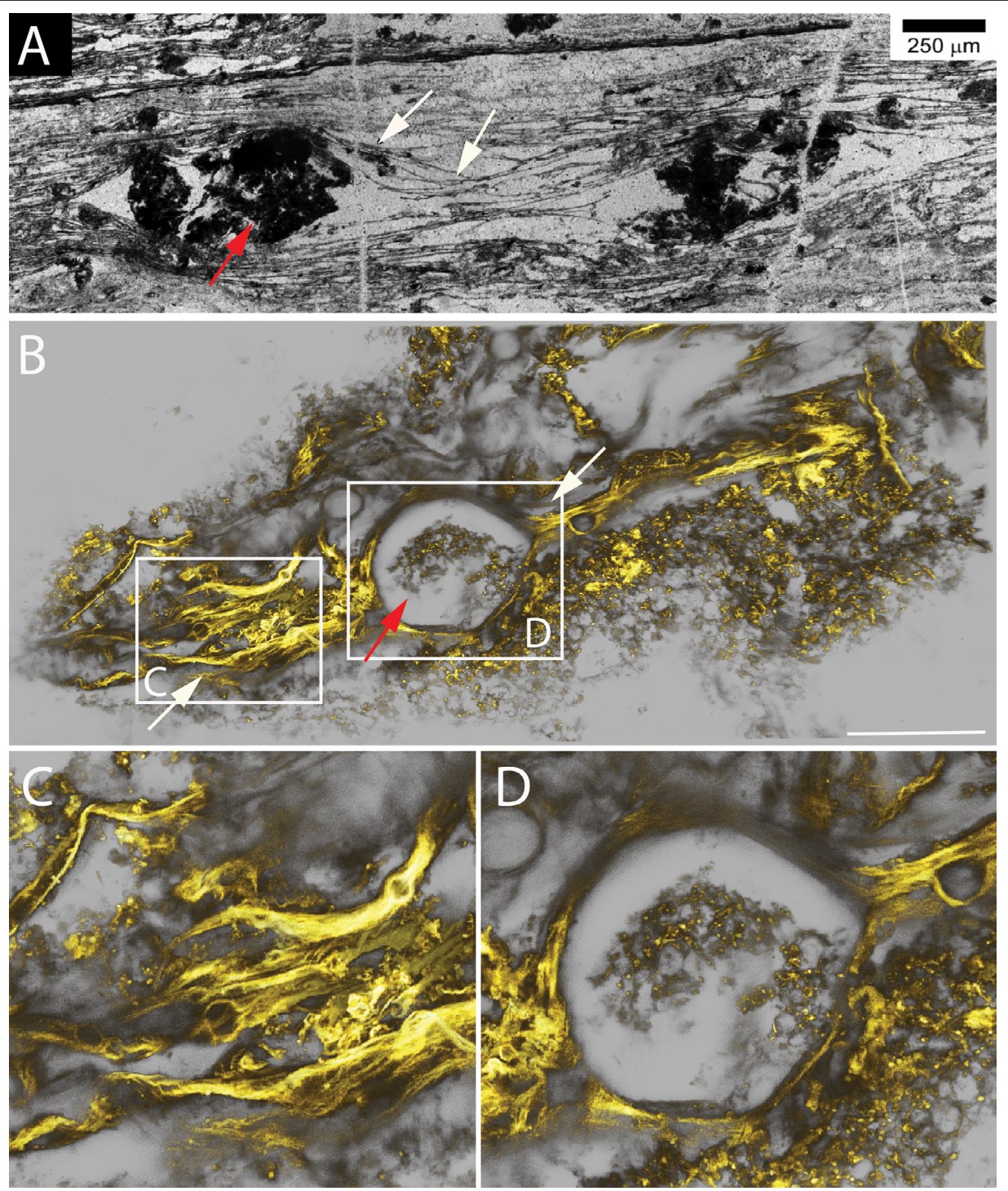

**Figure 8.** Morphological comparison between laminated structures reported from the Buck Reef Chert and structures formed by *EM-P*. Image (**A**) shows laminated structures reported from the Buck Reef Chert (originally published by *Tice, 2009*). They show parallel layers of organic carbon with lenticular gaps. Together with the quartz, these lenticular gaps consist of clumps of organic carbon. Image (**B**) is a 3D-rendered confocal image of analogous membrane debris formed by *EM-P*. Images (**C** & **D**) are the magnified regions of C. Like Buck Reef Chert formation, filamentous membrane debris bifurcating to form spherical/lenticular gaps can be seen in several regions (*Figure 8—figure supplements 1–4*). Some spherical/lenticular gaps were hollow, and some had an organic structure within them, even exhibiting a honeycomb pattern (arrow), suggesting the presence of large spherical *EM-P* cells with intracellular vesicles (**D**, & *Figure 8—figure supplement 3*). Membranes were stained with Nile red, and imaging was done using an STED microscope. The scales: 50 µm.

*Figure 8 continued on next page*

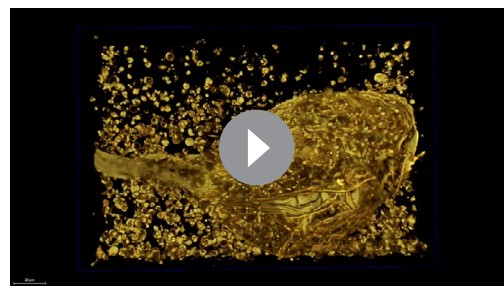

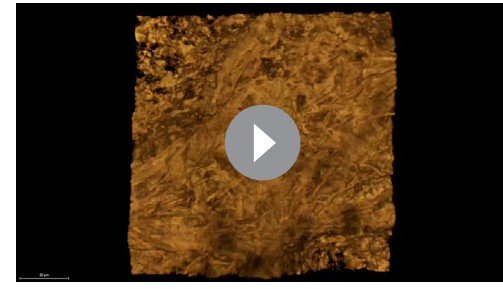

**Video 20.** The video shows layered membrane debris of *EM-P* forming hollow lenticular structures. Such structures should have formed by random folding of membrane debris, as no indication of trapped cells was observed in these structures. Scale bar: 20 μm.
https://elifesciences.org/articles/98637/figures#video20

**Video 21.** The video shows solidified *EM-P* biofilm. Hollow lenticular structures could be seen within the layers of membrane debris. Such structures should have formed honeycomb patterns within these lenticular gaps suggesting the presence of one's intact *EM-P* cells within these structures. Scale bar: 20 μm.
https://elifesciences.org/articles/98637/figures#video21

which involves the intake of salt-rich media and minerals into the cells (*Appendix 1—figure 64*). Like the Dresser Formation microfossils, we often observed the presence of salts and minerals within *EM-P*s vesicles (*Appendix 1—figure 64*). Moreover, the presence of minerals within cells is not unique to the Dresser Formation microfossils (*Wacey et al., 2018a*) and was reported previously from several bona fide microfossils from Gunflint Iron Formations (*Lepot et al., 2017*). Additionally, we observed remarkable similarities between the *EM-P* cell debris and all the organic structures closely associated with the microfossil, such as the wavy lamellar and pumice-like structures (*Appendix 1—figures 39, 41 and 65*). The step-by-step transformation of cell debris into pumice-like structures is shown in *Figure 6*. Based on these morphological similarities between the Dresser Formation organic structures and *EM-P*, we hypothesize that these organic structures are the fossil remnants of *EM-P*-like bacteria rather than mineral aggregates.

Morphological similarity between microfossils from far-flung sites like Western Australia and Southern Africa could be explained by the similarity in the environmental conditions in both sites (*Oehler et al., 2017*). This relationship between cell morphology, reproductive processes, and environmental conditions was discussed extensively in our previous work (*Kanaparthi et al., 2023*; *Kanaparthi et al., 2024*). The experimental conditions that we employed in our study are likely similar to the environmental conditions faced by Archaean organisms from both these sites at the time of their fossilization. All sites from which microfossils were reported are shallow intertidal regions. Evidence for periodic evaporation and flooding with sea water was presented from the Barberton and Pilbara Greenstone Belts (*Walsh, 1992*; *Alleon et al., 2018*), suggesting that the original microorganisms experienced high salinities. The salinities of our experiments are broadly similar to those of Archaean oceans (5–10% w/v)(*Knauth, 1998*). To our knowledge, the exact salt composition of the Archaean Ocean has not been elucidated. Hence, we used a complex mixture of salts (DSS) as a proxy to reproduce these salinities in our experiments. Salts like $Mg^{+2}$, $Ca^{+2}$, $Na^+$, and $K^+$ or their oxides were also reported to be present and constitute 1–5% by weight in both Pilbara and Barberton greenstone belt microfossil sites (*Walsh, 1992*; *Alleon et al., 2018*). Moreover, these salts were shown to be closely associated with microfossils (*Alleon et al., 2018*). The spatial distribution of these salts resembles the spatial distribution pattern of organic carbon, possibly indicating the chelation of these salts to the cell

*Figure 8 continued*

The online version of this article includes the following figure supplement(s) for figure 8:

**Figure supplement 1.** Morphological comparison of the Moodies Group laminations with *EM-P's* membrane debris.

**Figure supplement 2.** Morphological comparison between laminated structures reported from the Moodies Group and structures formed by *EM-P*.

**Figure supplement 3.** Lenticular membrane debris of *EM-P*.

**Figure supplement 4.** Lenticular membrane debris of *EM-P*.

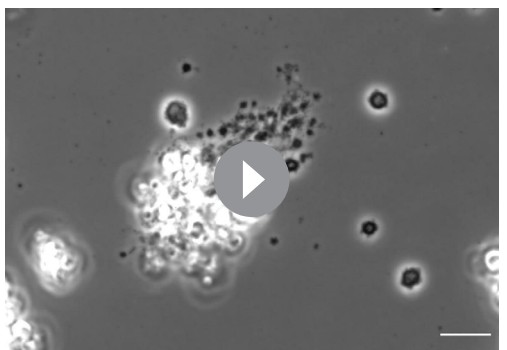

**Video 22.** The video shows a string of *EM-P* cells growing out of the stellar-shaped salt-encrusted cells. Scale bar: 10 µm.

https://elifesciences.org/articles/98637/figures#video22

membrane, which is also in agreement with our observations (*Wacey et al., 2016*). The presence of potassium phyllosilicates and NaCl crystals within the microfossils (*Alleon et al., 2018*) is also in agreement with our hypothesis that internal structures of the microfossils should have formed by invagination of cell-membrane taking in salt-rich water (*Appendix 1—figure 62*). As observed in the microfossils (*Alleon et al., 2018*), salt crystals on the cell surface, within the membrane invaginations, or cell debris were often observed in *EM-P* (*Appendix 1—figure 65*).

The above-presented results suggest that Archaean Eon cells are likely primitive lipid-vesicle-like protocells that lack a cell wall. From a physiological perspective, it would have been unlikely for primitive cells to possess a cell wall given the substantial number of genes required to synthesize individual building blocks, to mediate its assembly, and its constant modification to facilitate cell growth and reproduction (*Adams and Errington, 2009*; *Vollmer et al., 2008*; *Egan et al., 2017*). Furthermore, a cell wall could impede the transport of physiologically relevant compounds in and out of the cells. To overcome this limitation, present-day microorganisms (with a cell wall) had to develop extensive molecular biological processes for transporting nutrients and metabolic end products across the cell wall (*Dijkstra and Keck, 1996*; *Prajapati et al., 2021*). This could not have been the case for primitive Archaean life forms. Hence, rather than drawing parallels between the microfossils and life as we know it today, we propose that these microfossils could have been liposome-like protocells, as proposed by the theory of chemical evolution (*Oparin, 1969*). Indeed, it has been recently shown that liposome-like molecules could be produced in some of the hydrothermal settings proposed for the emergence of life (*Purvis et al., 2024*). To the best of our knowledge, this is the first study to provide a link between theoretical propositions and geological evidence for the existence of protocells on early Earth.

According to the theory of chemical evolution, biological organic compounds are formed by abiotic processes (*Powner et al., 2009*). These compounds then self-assembled to form lipid vesicles, which grew in complexity and eventually evolved into self-replicating protocells (*Zhu and Szostak, 2009*; *Chang et al., 2023*). These protocells are believed to have undergone Darwinian evolution, resulting in the emergence of bacteria, archaea, and eukaryotes (*Woese et al., 1990*). It was previously thought that the fragility of protocells made it unlikely for them to be preserved in rock formations. However, later studies showed the preservation of cellular features by a rapid encrustation of cells with cationic minerals (*Orange et al., 2009*; *Orange et al., 2011*). The rapid encrustation and preservation of cells observed in our study (*Appendix 1—figures 60 and 61*) is in accordance with the proposition that environmental conditions influence the extent of cellular preservation. Our study aligns with the interpretations from these studies that environmental conditions play a pivotal role in determining the extent of cellular preservation.

## Conclusion

For the first time, our investigations have been able to reproduce morphologies of most Archaean microfossils from wall-less, extant cells. Apart from reproducing the morphologies, we also presented a step-by-step biological process by which Archaean organic structures could have formed. Based on these results, we propose that Archaean microfossils were likely liposome-like cells, which had evolved mechanisms for energy conservation but not for regulating cell morphology and replication. In an earlier study, we have shown that the morphologies of such primitive cells are determined by environmental conditions (*Kanaparthi et al., 2023*; *Kanaparthi et al., 2024*) rather than the information encoded in their genome. Given this lack of intrinsic ability to regulate their morphology, we argue that morphological features such as cell size, shape, or cytological complexity are reliable factors in interpreting either the phylogeny or the physiology of microfossils (at least from the Archaean Eon).

Rather than attempting to assign present-day taxonomies to these microfossils, we suggest that these microfossils represent primitive protocells proposed by the theory of chemical evolution. To the best of our knowledge, ours is the first study to provide paleontological evidence of the possible existence of protocells on the Palaeoarchaean Earth.

## Methods

### Isolation of cells and their transformation to protoplasts

*Exiguobacterium* strain-Molly (*EM*) was isolated from submerged freshwater springs within the Dead Sea (*Häusler et al., 2014*). The taxonomic identification of the isolate to the genus *Exiguobacterium* was determined by 16 S rRNA gene sequencing (*Kanaparthi et al., 2017*; *Kanaparthi and Conrad, 2015*). *EM* cells were transformed into protoplasts following a previously documented protocol (*Kanaparthi et al., 2023*). The resulting *EM-P* cells were cultured in half-strength TSB with 7% Dead Sea Salt (7% DSS-TSB).

### Microscopic observation of *EM-P* cells

Morphology *EM-P* was routinely assessed using an Axioskop 2plus microscope (Carl Zeiss, Germany) with a Plan-NEOFLUAR ×100 /1.3 objective. Images were captured using a Leica DSF9000 camera (Leica Microsystems, Mannheim, Germany). STED microscopy was performed with an inverted TCS SP8 STED ×3 microscope (Leica Microsystems, Mannheim, Germany) using an ×86 /1.2 NA water immersion objective (Leica HC PL APO CS2 - STED White). Fluorophores were excited with 488, 561 nm, 594 nm, or 633 m laser light derived from an 80 MHz pulsed White Light Laser (Leica Microsystems, Mannheim, Germany). For stimulated emission, either a pulsed 775 nm laser or a 592 nm CW laser (Leica Microsystems, Mannheim, Germany) was used depending on the fluorophore. Photon counting mode and line accumulation were used for image recording, and Huygens Professional (SVI, Hilversum, The Netherlands) was used for image deconvolution on selected images and videos.

Spinning Disk Microscopy was performed using an Olympus SpinSR10 spinning disk confocal microscope (Olympus, Tokyo, Japan) equipped with a ×100 /NA1.35 silicone oil immersion objective (Olympus UPLSAPO100XS, Tokyo, Japan), a CSU-W1-Spinning Disk-Unit (Yokogawa, Tokyo, Japan) and ORCA-Flash 4.0 V3 Digital CMOS Camera (Hamamatsu, Hamamatsu City, Japan).

Transmission electron microscopy was conducted utilizing a Zeiss EM 912 (Zeiss, Oberkochen, Germany) equipped with an integrated OMEGA filter, operating at 80 kilovolts (kV). Image acquisition was carried out using a 2k × 2k pixel slow-scan CCD camera (TRS, TrÖndle Restlichtverstrkersysteme, Moorenweis, Germany) with ImageSP software (SysProg, Minsk, Belarus).

## Acknowledgements

We want to thank Gabriella Berthal for her excellent technical support and Christian Sibert for providing the Dead Sea samples from which *EM* was isolated. We thank the Advanced Light Microscopy Facility at EMBL, Heidelberg, Ulf Schwartz from Leica Microsystems, and colleagues at the departments of Ecological Microbiology (Bayreuth University) and of Cellular and Molecular Biophysics (Max Planck Institute for Biochemistry) for their support throughout the work. This research was funded by the European Research Council (ERC) grant agreement 616644 (POLLOX) and by the Deutsche Forschungsgemeinschaft (DFG) grant agreements DFG-TRR174 and Seed funding from Excellence Cluster ORIGINS EXC2094 – 390783311.

## Additional information

### Funding

| Funder | Grant reference number | Author |
| --- | --- | --- |
| European Research Council | 616644 | Tillmann Lueders |

| Funder | Grant reference number | Author |
|---|---|---|
| Deutsche Forschungsgemeinschaft | DFG-TRR174 | Petra Schwille |
| Seed funding from Excellence Cluster | ORIGINS EXC2094 - 390783311 | Dheeraj Kanaparthi |

The funders had no role in study design, data collection and interpretation, or the decision to submit the work for publication. Open access funding provided by Max Planck Society.

## Author contributions

Dheeraj Kanaparthi, Conceptualization, Formal analysis, Validation, Visualization, Methodology, Writing – original draft, Writing – review and editing; Frances Westall, Supervision, Writing – original draft, Writing – review and editing; Marko Lampe, Resources, Software, Supervision, Methodology, Writing – original draft, Writing – review and editing; Baoli Zhu, Data curation, Formal analysis, Methodology, Writing – review and editing; Thomas Boesen, Methodology; Bettina Scheu, Supervision, Funding acquisition, Writing – review and editing; Andreas Klingl, Methodology, Writing – review and editing; Petra Schwille, Funding acquisition, Writing – review and editing; Tillmann Lueders, Supervision, Funding acquisition, Project administration, Writing – review and editing

## Author ORCIDs

Dheeraj Kanaparthi https://orcid.org/0000-0003-1009-4103
Frances Westall https://orcid.org/0000-0002-1938-5823
Marko Lampe https://orcid.org/0000-0002-4510-9048
Petra Schwille https://orcid.org/0000-0002-6106-4847
Tillmann Lueders http://orcid.org/0000-0002-9361-5009

Joint Public Review: https://doi.org/10.7554/eLife.98637.3.sa1
Author response https://doi.org/10.7554/eLife.98637.3.sa2

## Data availability

The raw data related to this study has been uploaded to Zenodo at https://doi.org/10.5281/zenodo.18086634.

The following dataset was generated:

| Author(s) | Year | Dataset title | Dataset URL | Database and Identifier |
|---|---|---|---|---|
| Kanaparthi D, Westall F, Lampe M, Zhu B, Boesen T, Scheu B, Klingl A, Schwille P, Luederes T | 2025 | On the nature of the earliest known lifeforms | https://doi.org/10.5281/zenodo.18086634 | Zenodo, 10.5281/zenodo.18086634 |

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

## Appendix 1

## On the nature of the earliest known lifeforms

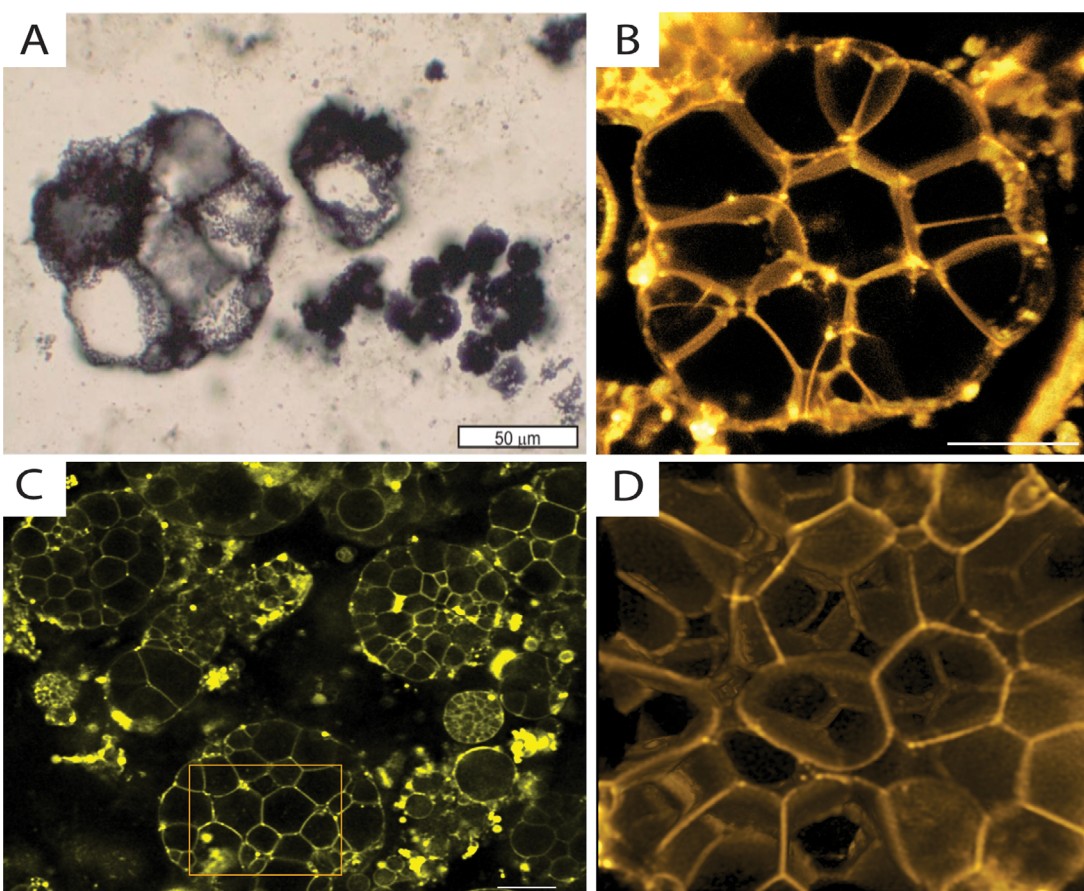

**Appendix 1—figure 1.** Morphological comparison between *EM-P* and the Farrel Quartzite microfossils. Image (**A**) shows microfossils with hollow polygonal vacuoles from the Farrel Quartzite formation (originally published by *Retallack et al., 2016*; *Kapteijn et al., 2022*). Images (**B-D**) are images of morphological analogous *EM-P* cells with polygonal vacuoles. Cells in images (**B-D**) were stained with membrane stain, FM5-95. Polygonal vacuoles in these images could have formed by the confinement of many vacuoles within a cell, as shown in *Figure 1—figure supplement 4F*. The cytoplasm in these cells was restricted to narrow spaces between the vacuoles, as shown in *Figure 1E and F*. Scale bar: 10 μm (**B**) and 20 μm (**C**).

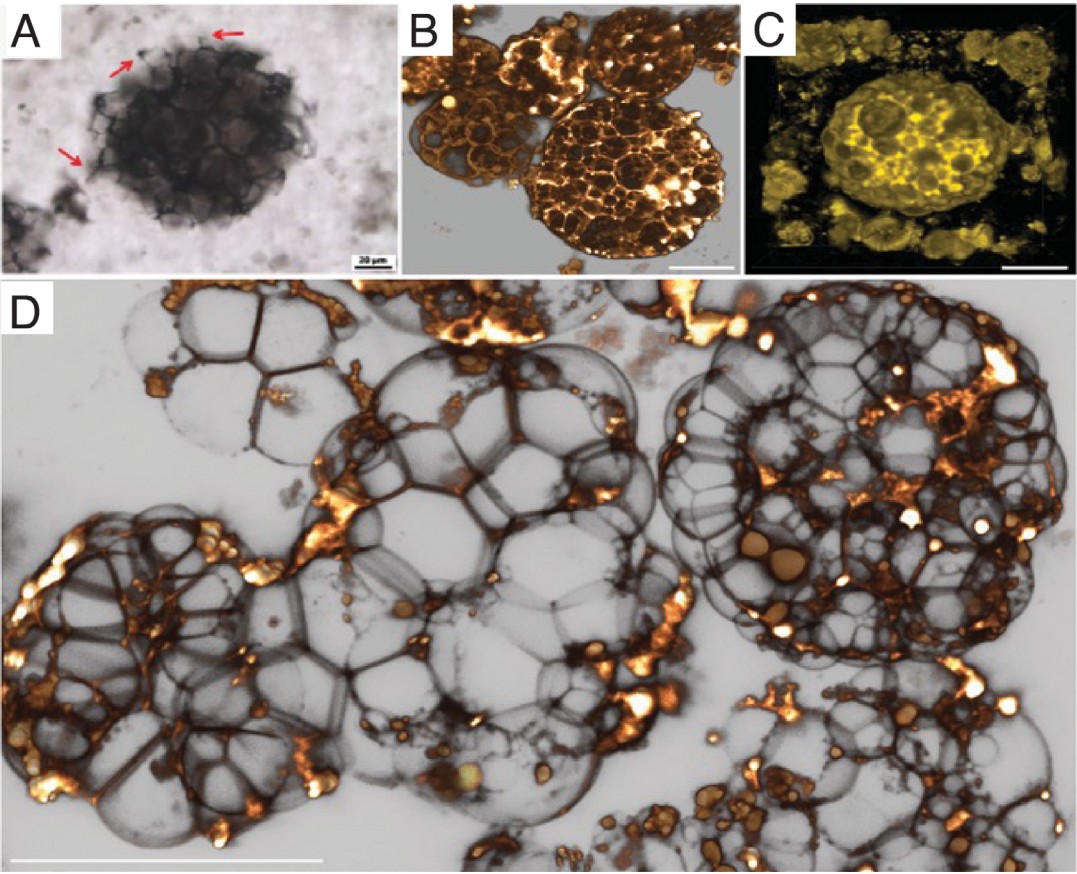

**Appendix 1—figure 2.** Morphological comparison between honeycomb structures reported from the Turee Creek Formations and *EM-P*. Image (**A**) shows a spherical microfossil reported from the Turee Creek Formation (originally published by *Barlow and Van Kranendonk, 2018*; *Sugitani et al., 2009*). Images (**B-D**) are morphologically analogous to *EM-P* cells. Cells in these images were stained with membrane stain, FM5-95. Scale bar: 20 µm.

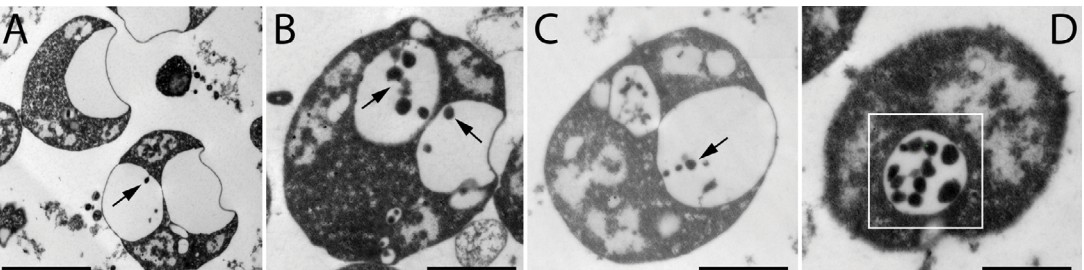

**Appendix 1—figure 3.** Morphological comparison between *EM-P* and the Fig Tree Formation microfossils. Images (**A-F**) show spherical microfossils reported from Fig Tree Formation (originally published by *Schopf and Barghoorn, 1967*; *Sugitani et al., 2007*). Images (**G-I**) are morphologically analogous phase-contrast images of *EM-P*. The presence of regions with and without organic carbon can be seen in microfossil images A-E and *EM-P* cell shown in images (**G & H**). Distinctive spherical vacuoles devoid of organic carbon can be seen in images (**B & C**), (**G & H**). Image (**F**) shows hollow structures morphologically similar to hollow ICVs released by the lysis of *EM-P* cells (**J & K**). Scale bar: 10 µm (**G–K**).

**Appendix 1—figure 4.** Sequential stages of intracellular daughter cell formation. Images (**a-d**) are TEM images of *EM-P* cells showing the formation of daughter cells into hollow ICVs. Arrows in image (**A**) point to the first step
*Appendix 1—figure 4 continued on next page*

in daughter cell formation by a process resembling budding. Images (**B-D**) show the gradual growth of these buds into a string of daughter cells. Scale bar: 500 nm.

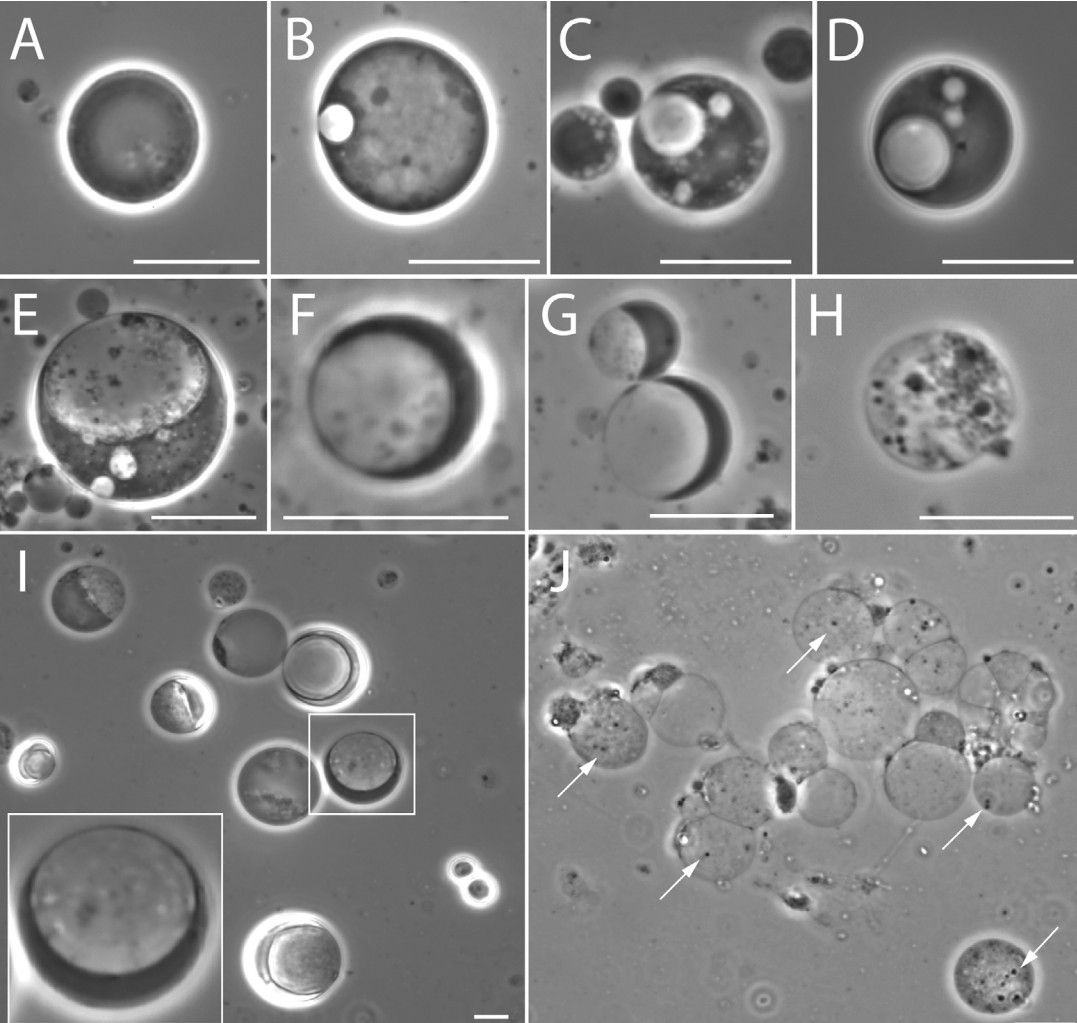

**Appendix 1—figure 5.** Lifecycle of *EM-P* cells reproducing via the formation of internal daughter cells. Images (**A-H**) show phase-contrast images of *EM-P* reproducing by forming internal daughter cells. These images show a gradual increase in the volume of the ICV and a proportional decrease in the cytoplasmic volume of the cells. Images (**I & J**) show *EM-P* cells in their mid and late stationary growth phase. Insert in the image I show a magnified view of the highlighted cell. White arrows in image-J point to internal daughter cells. Scale bars: 10 μm.

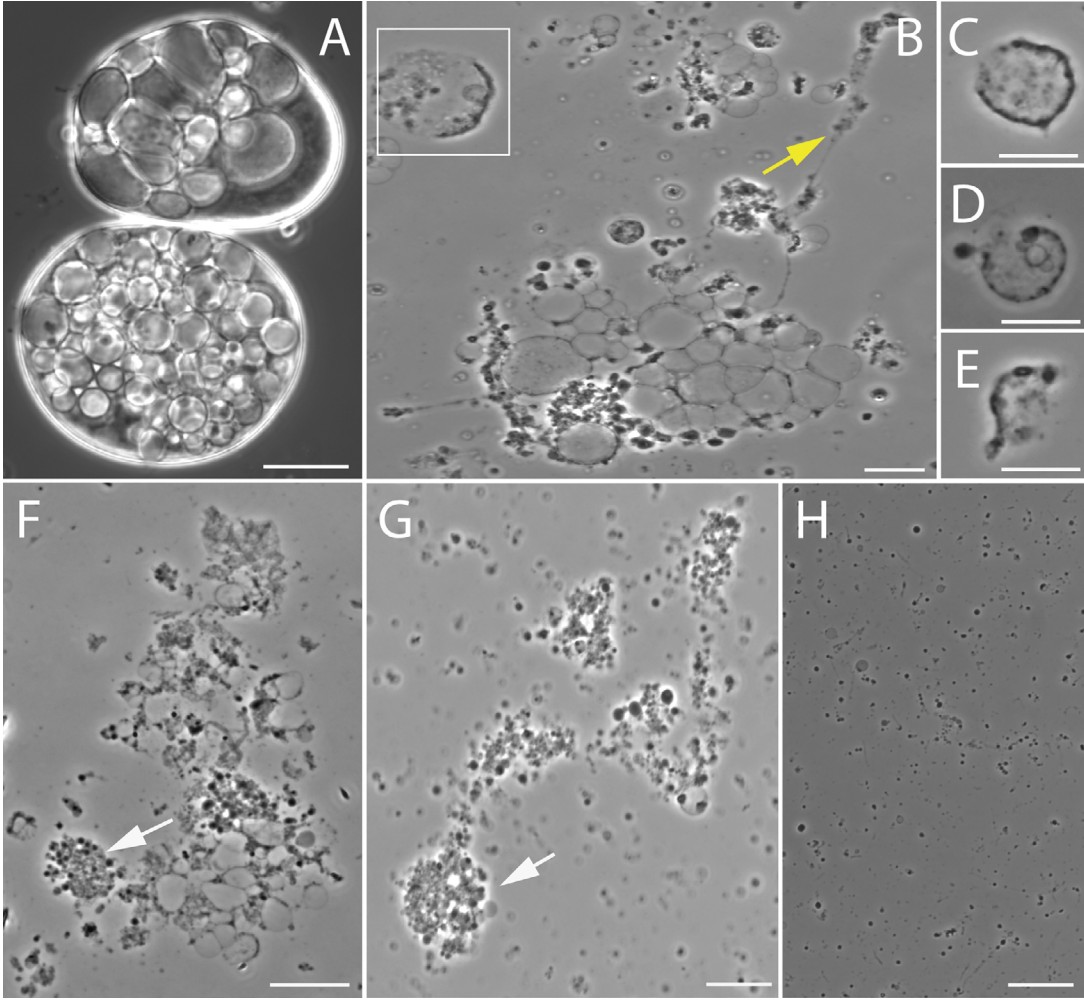

**Appendix 1—figure 6.** Lifecycle of *EM-P* cells reproducing via the formation of internal daughter cells. Images (**A-H**) show phase-contrast images of *EM-P* cells undergoing lysis and release of internal daughter cells. The highlighted region in image (**B**) and images (**C-E**) shows a cell undergoing lysis and cells in different stages of lysis. Arrows in image (**B**) point to thin strands of membrane debris formed during cell lysis. Arrows in images (**F & G**) point to clusters of daughter cells released by the lysis of *EM-P* cells. These cell clusters gradually dispersed, leading to the formation of individual daughter cells (**H**). TEM images of such clusters of interconnected daughter cells and individual daughter cells with membrane overhangs are shown in *Appendix 1—figure 7F*. Scale bar: 10 µm.

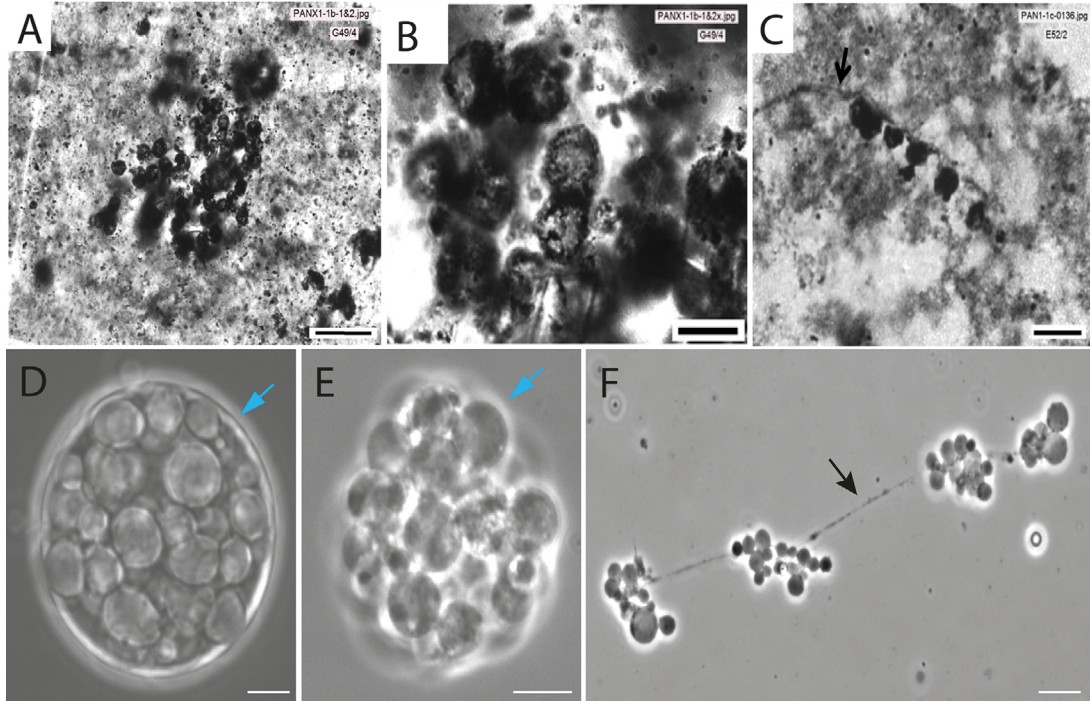

**Appendix 1—figure 7.** Morphological comparison between the SPF microfossils and *EM-P*. Images (**A-D**) are hollow spherical SPF microfossils with internal inclusions (originally published by *Sugitani et al., 2013*; *Schopf and Barghoorn, 1967*). Images (**D-F**) are images of *EM-P* cells undergoing lysis (**E**) and dispersion (**F**) of intracellular vacuoles. Cyan arrows in images (**D and E**) point to the presence and absence of the outer cell membrane. Black arrows in images (**C & F**) point to filamentous structures interlinking spherical structures. Scale bar: 50 µm (**A**), 20 µm (**B & C**), 2 µm (**D & E**), and 5 µm (**F**).

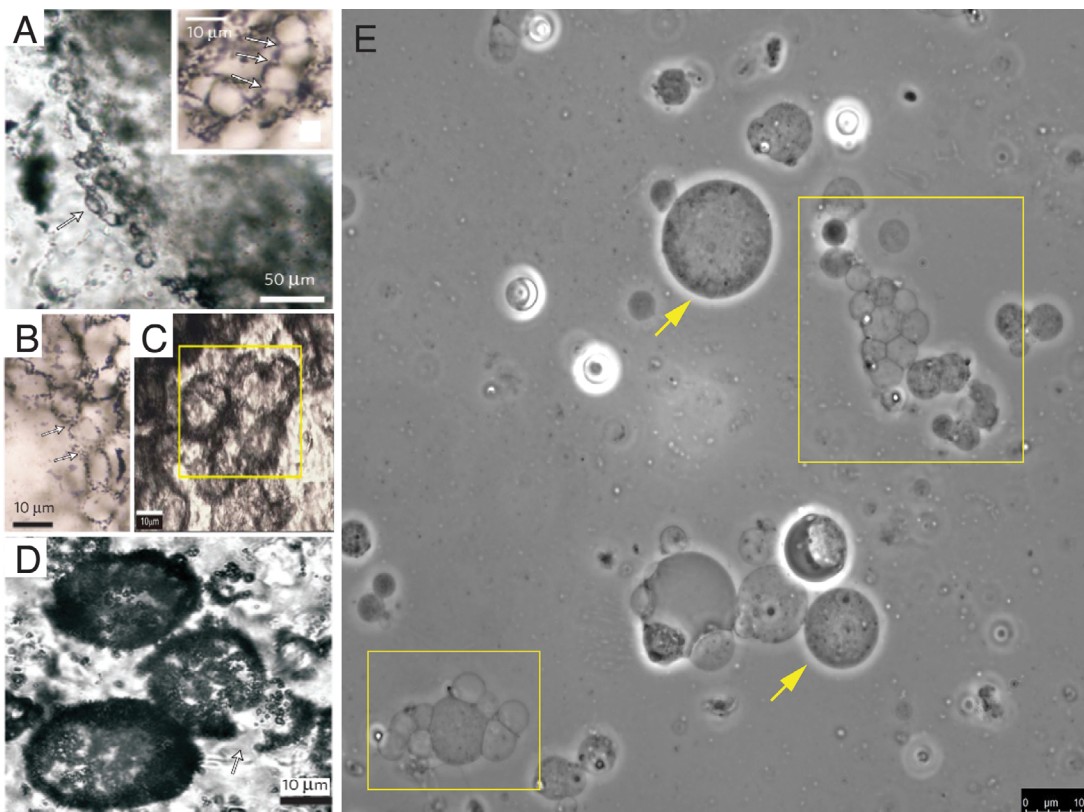

**Appendix 1—figure 8.** Morphological comparison between *EM-P* and Strelley Pool Formation (SPF) microfossils. Images (**A-D**) depict SPF microfossils with intracellular organic inclusions (originally published by *Wacey et al., 2011*). Image (**E**) shows morphologically analogous *EM-P* cells. Images (**A-C**) show chains and clumps of cells. A similar cluster of cells can be seen in image (**E**) (boxed regions). Image (**D**) shows spherical microfossils with intracellular spherical inclusions. Similar *EM-P* cells with intracellular daughter cells can be seen in image (**E**) (yellow arrows). Lysis and release of daughter cells can be seen in *Appendix 1—figure 6* and *Video 4*.

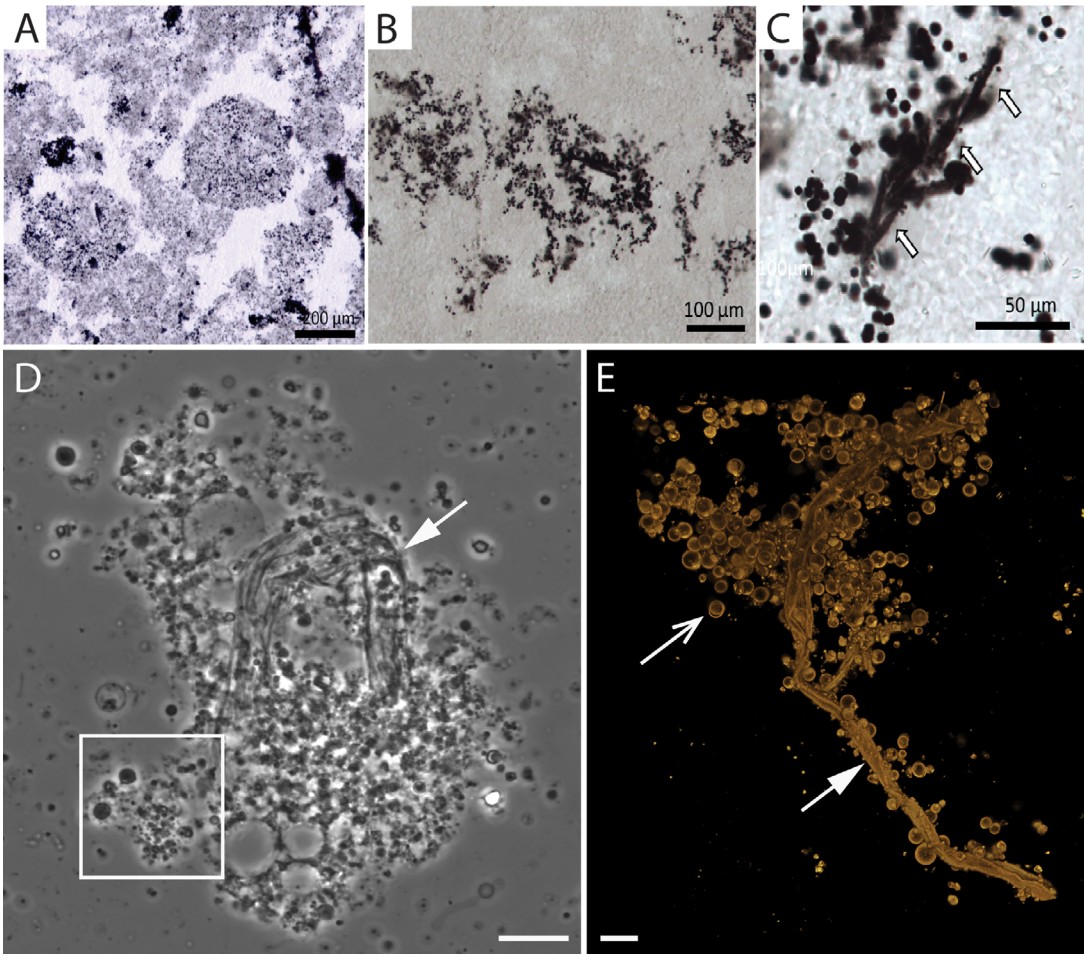

**Appendix 1—figure 9.** Morphological comparison between *EM-P* and the Strelley Pool Formation (SPF) microfossils. Images (**A-C**) show clusters of faint and dark spherical microfossils reported from SPF (originally published by *Sugitani et al., 2015b*; *Barlow and Van Kranendonk, 2018*). Images (**D & E**) show morphologically similar structures observed in *EM-P* incubations. Image A shows spherical clusters of spherical granules. Based on their morphological similarity, they likely represent the tiny intracellular daughter cells, as shown in D (highlighted region) and *Appendix 1—figure 8*. The thread-like structures in C (arrows) likely are the membrane debris often found associated with the release of daughter cells, as shown in (**D & E**) (arrows). Scale bars in images (**D & E**) are 10 μm.

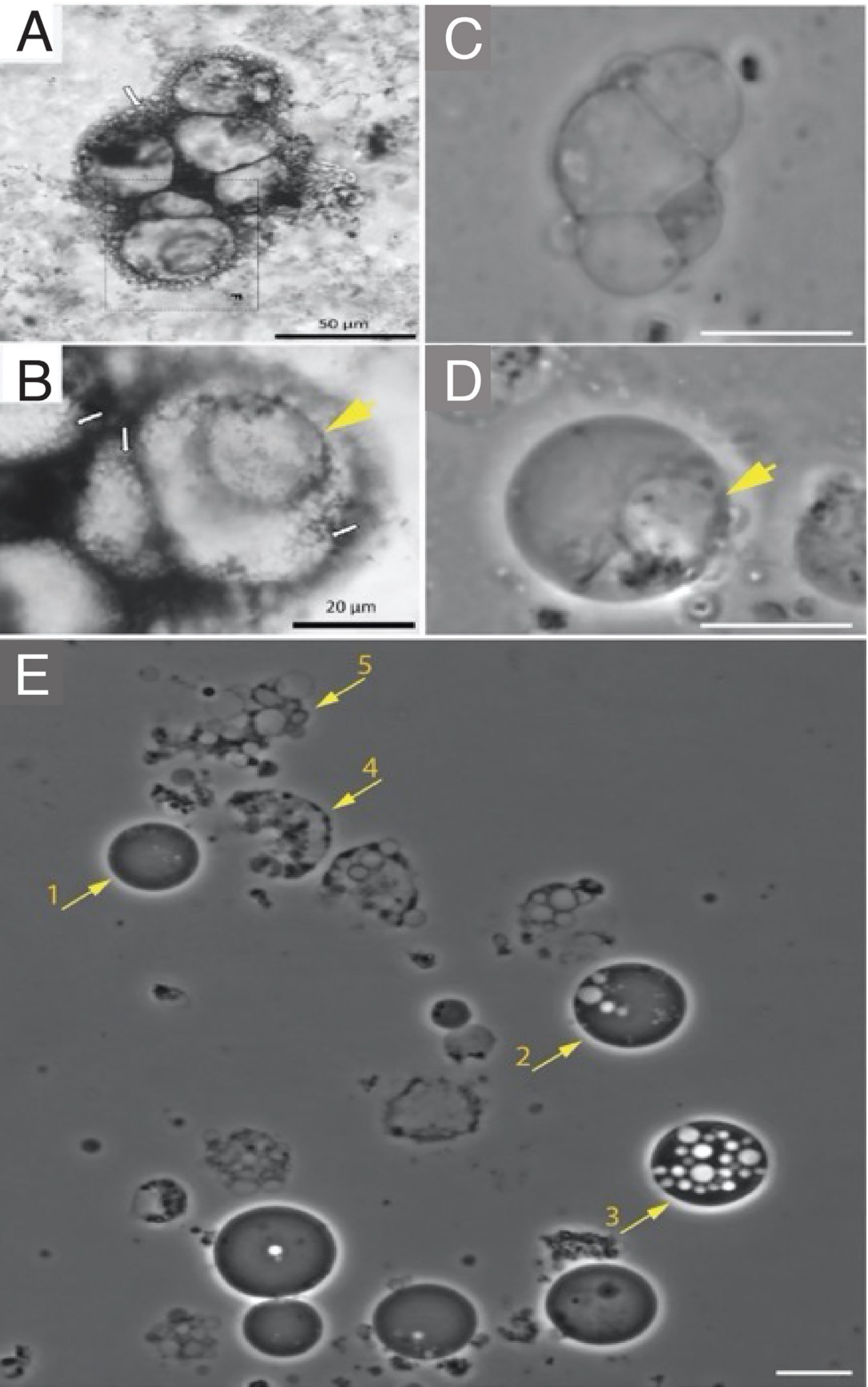

*Appendix 1—figure 10 continued on next page*

*Appendix 1—figure 10 continued*

**Appendix 1—figure 10.** Morphological comparison between *EM-P* and the Waterfall microfossils. Images (**A & B**) are microfossils reported from the Waterfall locality within the Strelley Pool Formation (SPF) (originally published by *Sugitani et al., 2015b*; *Barlow and Van Kranendonk, 2018*). Images (**C & D**) show morphologically analogous *EM-P* cells. Arrows in images (**C & D**) point to spherical intracellular vacuoles within the cell. Image E shows sequential stages involved in the formation of structures shown in images (**A & B**). Sequential steps are indicated by numbers next to the arrows. Scale bars: 10 µm (**C–E**).

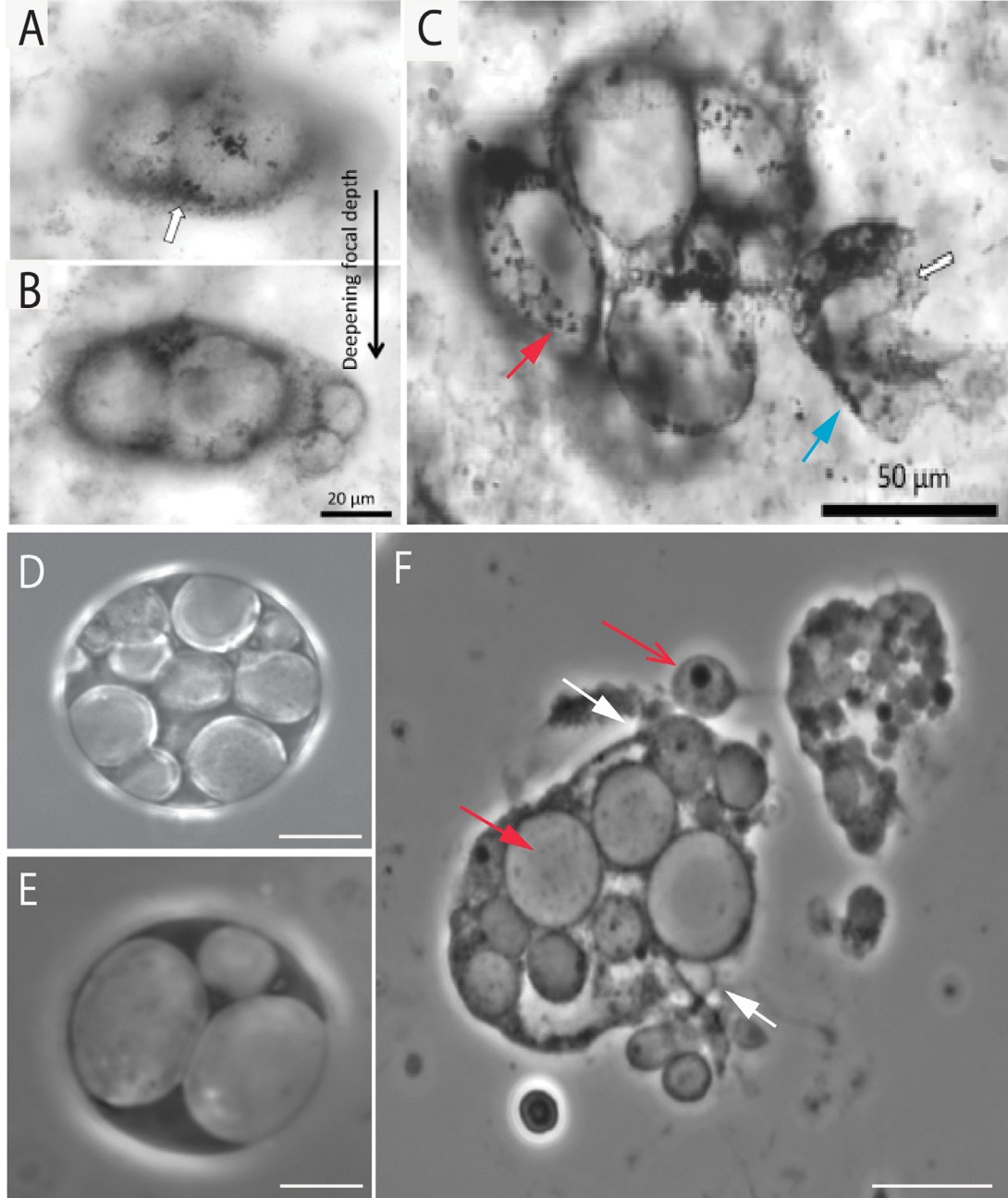

**Appendix 1—figure 11.** Morphological comparison between *EM-P* and Waterfall microfossils. Images (**A, B, & C**) are microfossils reported from the waterfall locality within SPF (originally published by *Sugitani et al., 2015b*; *Barlow and Van Kranendonk, 2018*). Images (**D-F**) are phase-contrast images of morphologically analogous structures formed by *EM-P*. Red arrows in these images point to intracellular vacuoles with daughter cells of different sizes. Open red arrows point to a vacuole with a relatively large daughter cell. The white arrows in image (**F**) point to the ruptured region of the cell membrane and the release of the intracellular vesicles (ICVs). Scale bars: 10 µm (**D–F**).

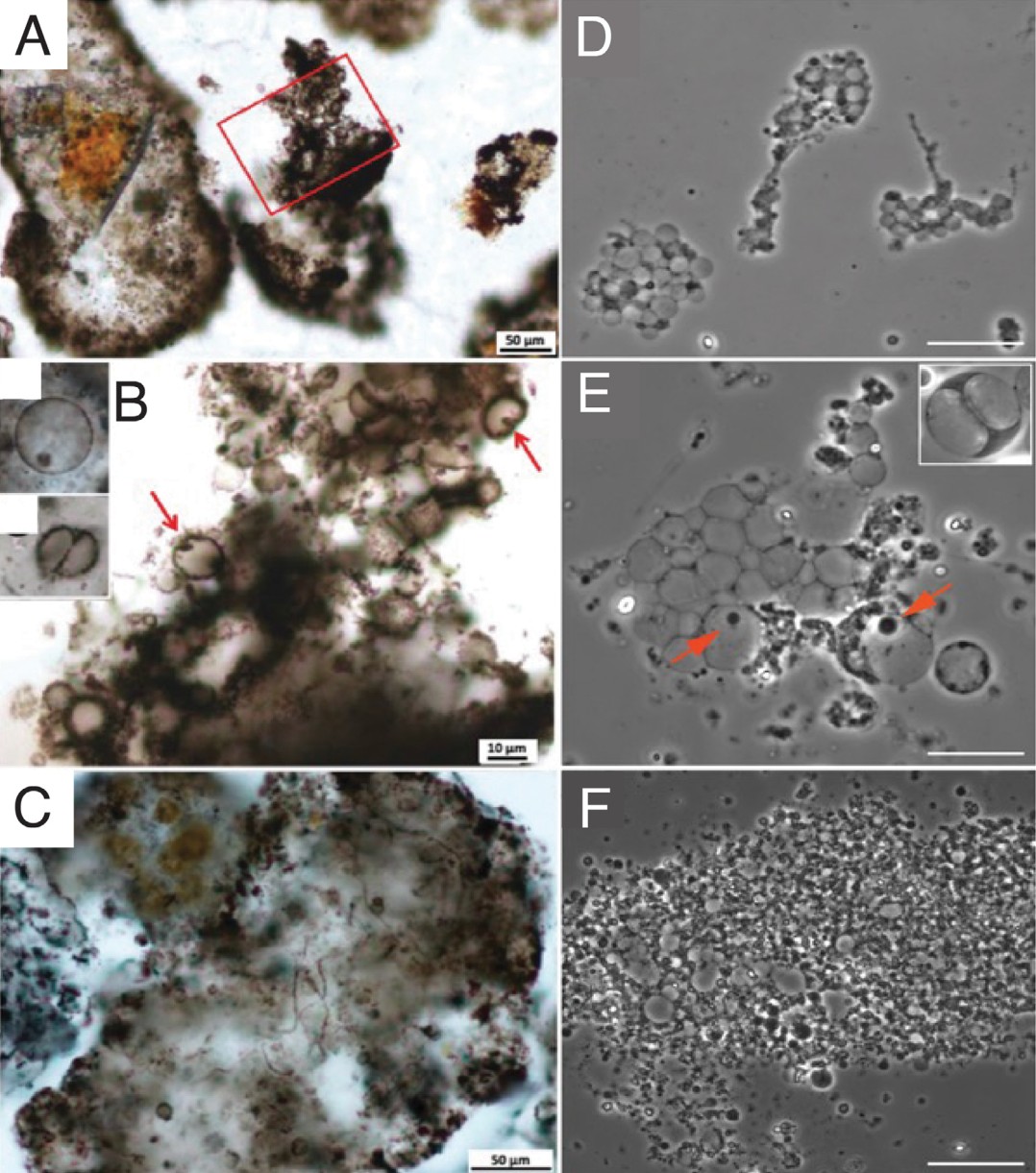

**Appendix 1—figure 12.** Morphological comparison between honeycomb structures reported from the Turee Creek Formations and *EM-P*. Images (**A, B, & C**) are microfossils reported from the Turee Creek locality within the Strelley Pool Formation (SPF) (originally published by *Barlow and Van Kranendonk, 2018*; *Sugitani et al., 2009*). Images (**D-F**) are phase-contrast images of morphologically analogous structures formed by *EM-P*. The boxed region in image (**A**) resembles the lysed cells of *EM-P*, shown in image (**D**). Arrows in image (**B & E**) point to the similar organic carbon inclusions (**B**) and morphologically analogous daughter cells (**E**) within a hollow spherical *EM-P* cell. Insert in images (**B & E**) show spherical cells with two intracellular vesicles or nearly equal volume. Scale bars: 10 μm (**D–F**).

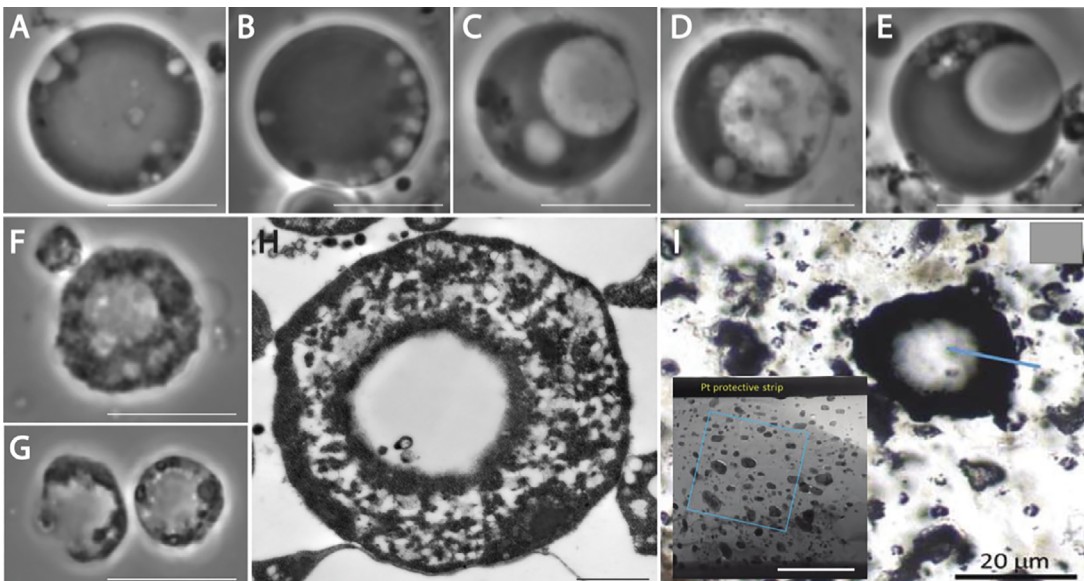

**Appendix 1—figure 13.** The sequence of morphological transformations of *EM-P* leading to the formation of cells having the appearance of a thick cell wall. Images (**A-G**) show the sequential morphological transformations of *EM-P* cells with one large central vacuole and multiple smaller vacuoles. Over the course of its growth, smaller vacuoles were squeezed between the cell membrane and the vacuole membrane (**F & H**), leading to the appearance of a porous cell wall. Images (**F-G and H**) are phase-contrast and TEM images of such *EM-P* cells. The image (**I**) shows a spherical microfossil reported from Dresser formation (originally published by *Wacey et al., 2018a*). Insert in the image-I shows the TEM image of the region indicated by the blue line. The porous nature of this region is similar to the *EM-P* cell shown in image (**H**). Scale bars: 10 μm (**A–G**) and 200 nm (**H**).

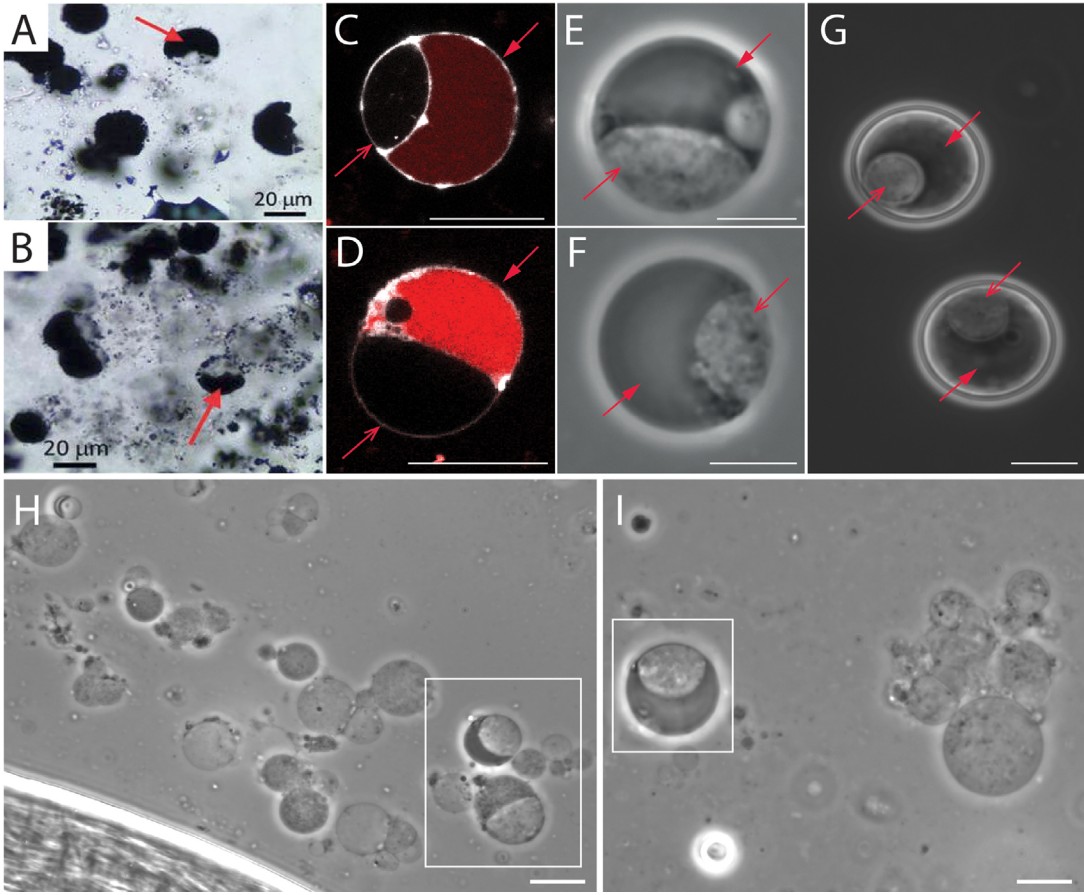

**Appendix 1—figure 14.** Morphological comparison of organic structures reported from the Dresser Formation with *EM-P*. Images (**A & B**) are spherical microfossils reported from the Dresser formation (originally published by *Wacey et al., 2018a*). Images (**C & D**) are STED microscope images of morphologically analogous *EM-P* cells. Cells in these images were stained with FM5-95 (membrane, white) and PicoGreen (DNA, red). Images (**G-I**) are phase-contrast images of morphologically analogous *EM-P* cells. Closed arrows in these images point to the regions of the cell with the cytoplasm (organic carbon), and open arrows point to the hollow space created by the intracellular vesicles (ICVs). Scale bars 10 µm (**C–I**).

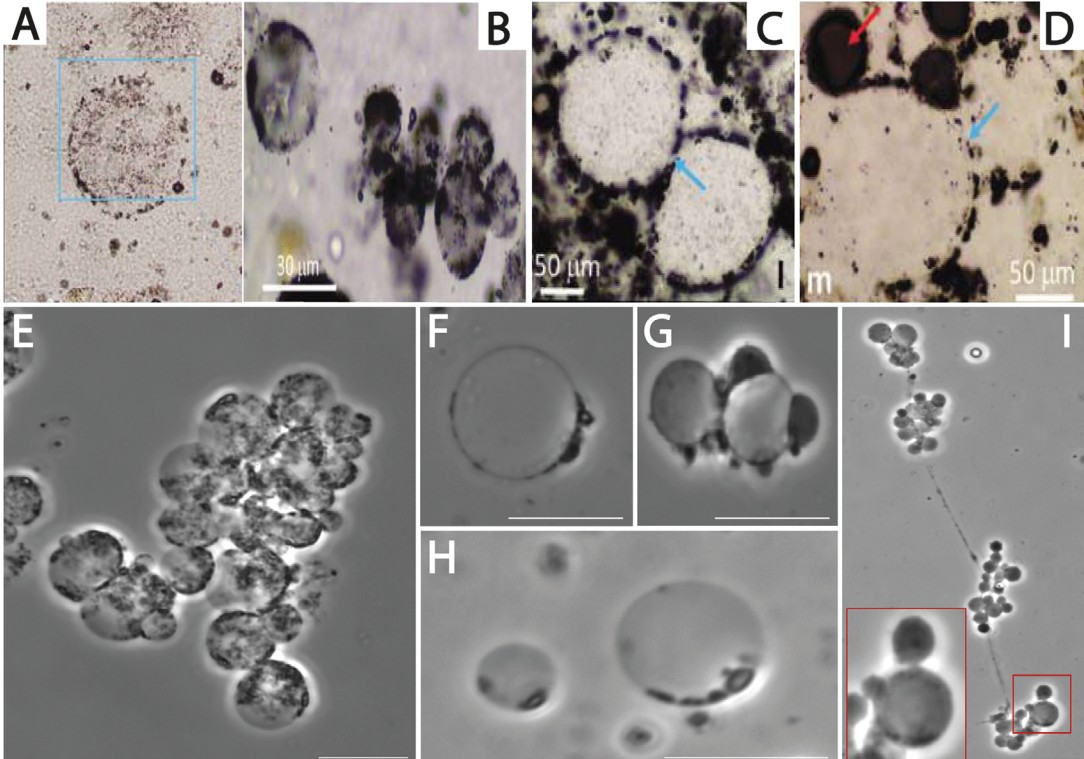

**Appendix 1—figure 15.** Morphological comparison of Dresser formation spherical microfossils with *EM-P*. Images (**A-D**) were aggregations of spherical microfossils reported from Dresser formation (originally published by *Wacey et al., 2018a*). (**E-I**) are images of morphologically analogous structures observed from *EM-P*. Blue arrows in images (**C & D**) point to cells with discontinuous cell walls. Similar cell boundary features can be seen in *EM-P* (**E–H**). Sequential stages involved in forming such *EM-P* cells are shown in *Appendix 1—figure 16*. The red arrow in image (**D**) points to structures with organic carbon attached to the hollow spherical cells. Image-g showed morphologically analogous *EM-P* cells. Scale bar: 10 µm (**E–I**).

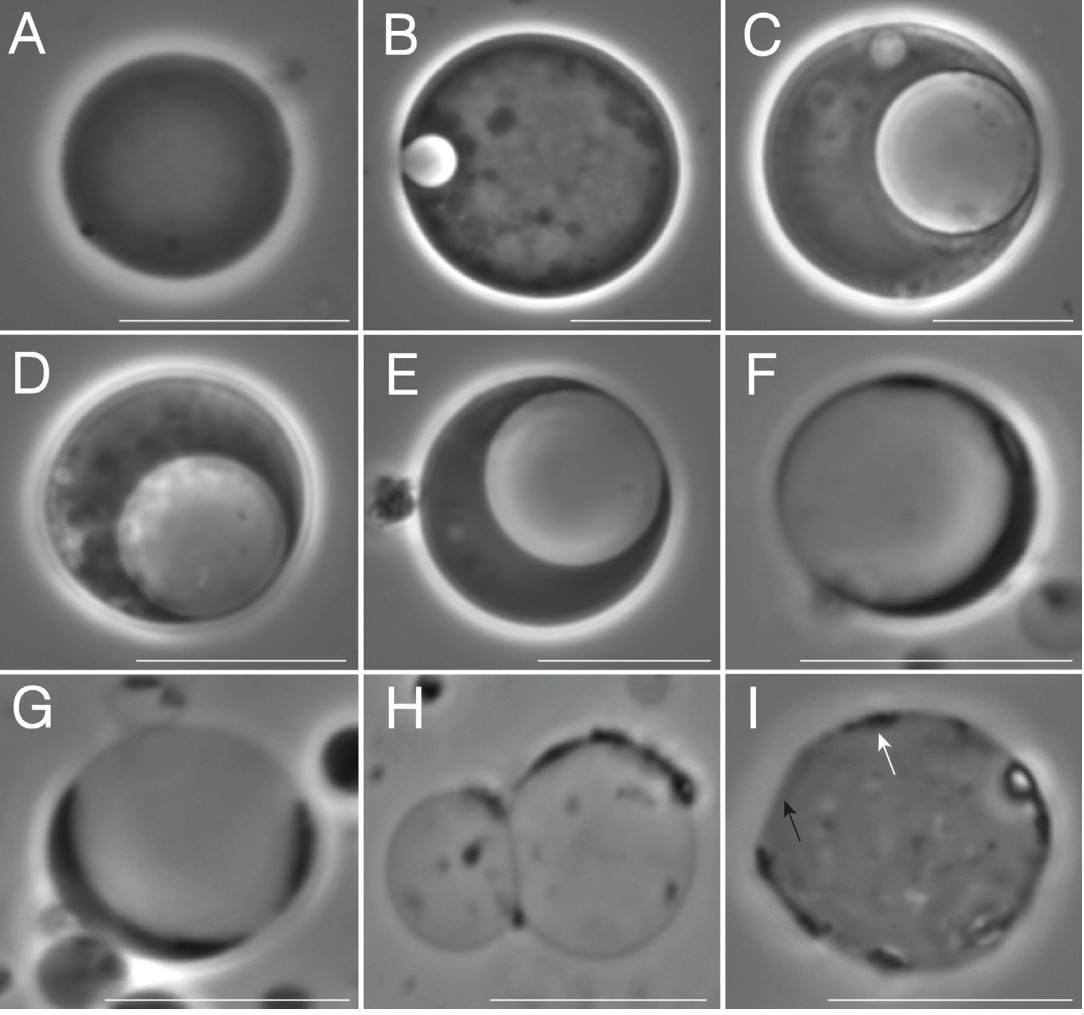

**Appendix 1—figure 16.** Sequential morphological transformation of *EM-P* cells with cytoplasm to hollow vacuoles. Images (**A-I**) are phase-contrast images of *EM-P*. In sequence, they show formation (**B**) and an increase in the size of the intracellular vacuole (**B–G**). Over the course of its growth, cytoplasm from the parent cell is transferred into the daughter cells in the vacuole (barely visible tiny spherical structures in the vacuole), which leads to the gradual depletion of cytoplasmic volume and a gradual increase in vesicle volume. In their late growth stages, the presence of cytoplasm is restricted to the periphery of the cell as discontinuous patches (**H & I**). Scale bars: 10 μm.

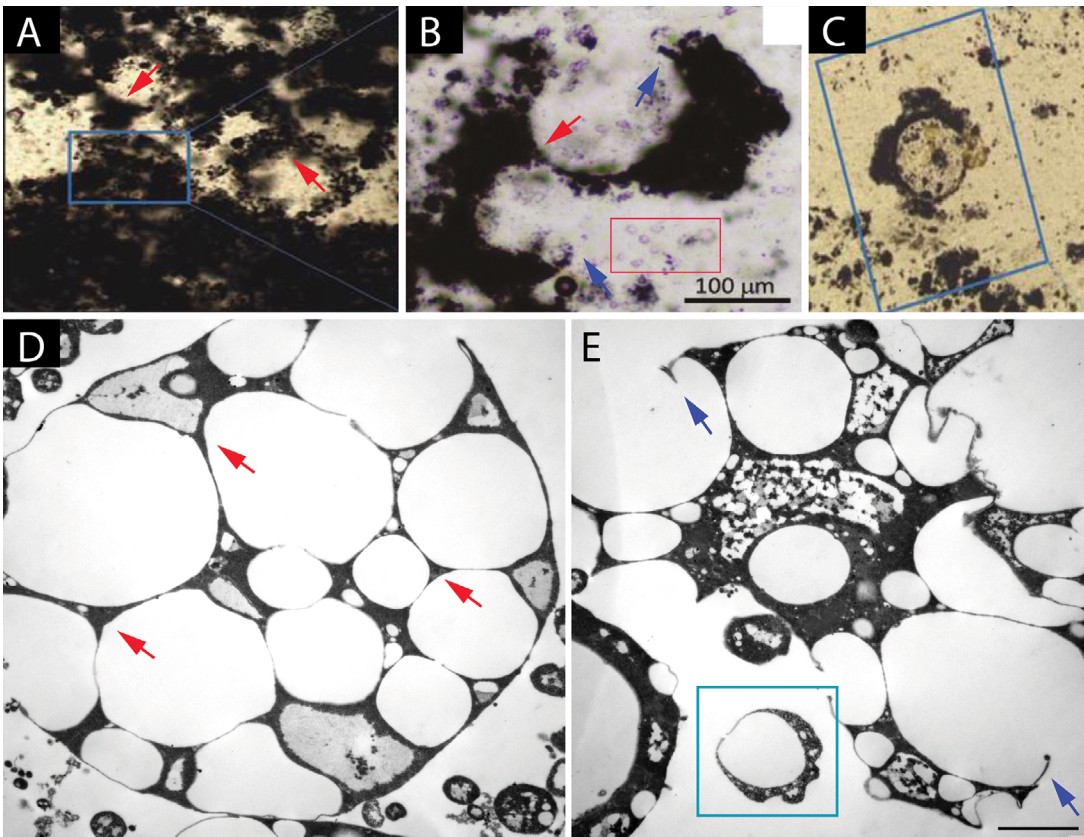

**Appendix 1—figure 17.** Morphological comparison of the Dresser Formation spherical microfossils with *EM-P*. Images (**A-C**) were aggregations of spherical microfossils reported from the Dresser Formation (originally published by *Wacey et al., 2018a*). Images (**D &E**) are TEM images of morphologically analogous *EM-P* cells. Image (**A**) shows hollow spherical aggregations that were devoid of organic carbon. Similar *EM-P* cells were shown in image (**D**). Red arrows in this image point to Y-shaped junctions with organic carbon. Such *EM-P* cells, over the course of their growth, underwent membrane rupture to release daughter cells (image (**E**), purple arrows). The process of lysis and dispersion of these cells is shown in Image (**E**). The morphology of these *EM-P* cells was very similar to the organic structures reported from the Dresser Formation (Image **B**). Like *EM-P* cells, daughter cells can also be seen nearby (red box). Image (**C**) shows an isolated spherical structure, with internal spherical inclusions with the presumed thick and uneven cell wall. Morphologically analogous *EM-P* cells can be seen in Image E (Cyan box). Scale bar: 250 nm (**E**).

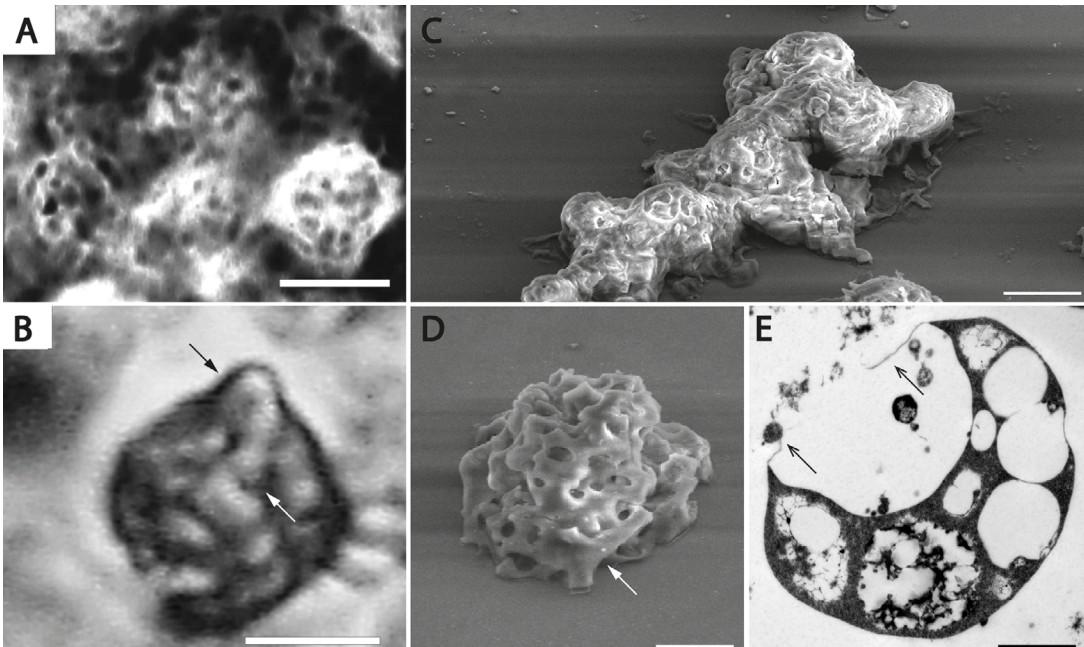

**Appendix 1—figure 18.** Morphological comparison between *EM-P* and the Kromberg Formation microfossils. Images (**A & B**) are microfossils reported from the Kromberg Formation (originally published by *Kaźmierczak and Kremer, 2019*; *Wacey et al., 2011*). Images (**C, D, & E**) are SEM and TEM images of morphologically analogous *EM-P* cells. Such *EM-P* cells were formed by deflation caused by lysis and release of daughter cells. SEM images show similar surface depressions in both the Kromberg microfossils and *EM-P* cells (arrows in **A & C**). Different stages of surface depression formation are shown in *Figure 1—figure supplement 1G–K*. Arrows in images (**B & D**) point to similar surface textures of microfossils and *EM-P* cells. Scale bars: 10 μm (**A & B**), 0.5 μm (**C & D**), and 250 nm (**E**).

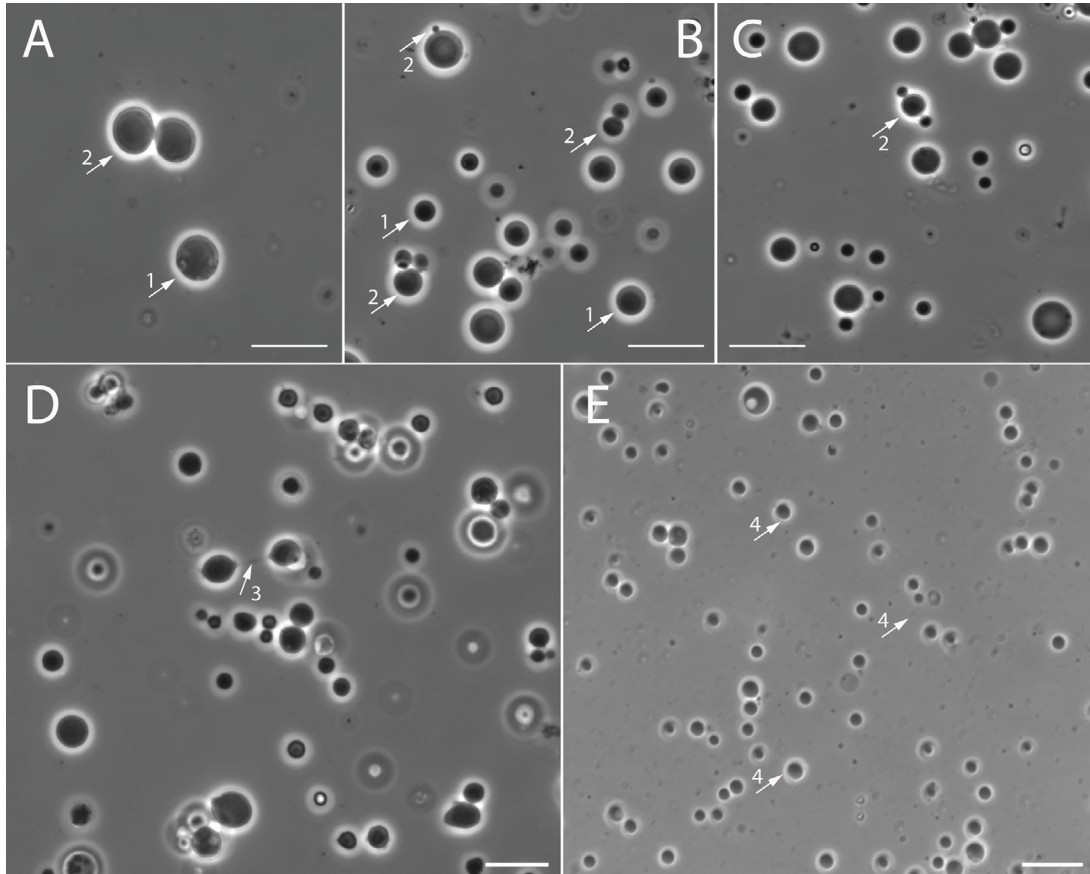

**Appendix 1—figure 19.** Reproduction by budding and binary fission. Images (**A-E**) are the phase-contrast images of *EM-P* reproducing by budding or binary fission. The arrows in the images point to different stages of *EM-P* reproduction. The arrows in images (**B & C**) point to the cells undergoing budding. Scale bars: 10 μm.

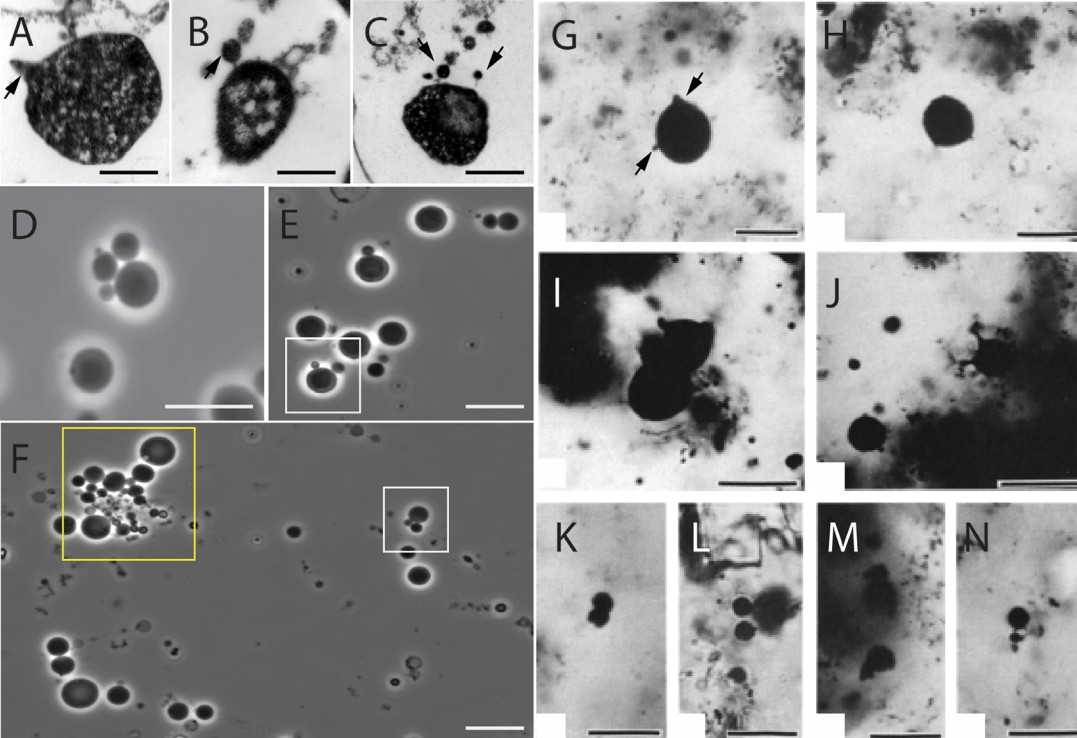

**Appendix 1—figure 20.** Morphological comparison between the North-Pole locality microfossils and *EM-P*. (**A-C**) are TEM images of *EM-P* showing sequential stages of bud formation. Images (**D-E**) are phase-contrast images of *EM-P* cells that appear to be reproducing by budding or binary fission, depending on the size of the daughter cells. Images G-N show spherical microfossils reported from the North Pole locality (originally published by *Buick, 1990*; *Kaźmierczak and Kremer, 2019*). These microfossils resemble *EM-P* cells reproducing by budding. Arrows in images (**A & G**) point to pustular protuberances in *EM-P* and the North-Pole locality microfossils. Images (**E, F, I & K**) show *EM-P* cells and microfossils reproducing by binary fission. Image (**D**) and highlighted regions in images (**E & F**) (in white) show *EM-P* cells reproducing by the formation of multiple buds (also *Figure 1B*). Highlighted region in image (**F**) (in yellow) shows cells forming Y-junctions, similar to the microfossils shown in N. Scale bars: 500 nm (**A & B**), 10 µm (**D–F**), and 100 µm (**G–N**).

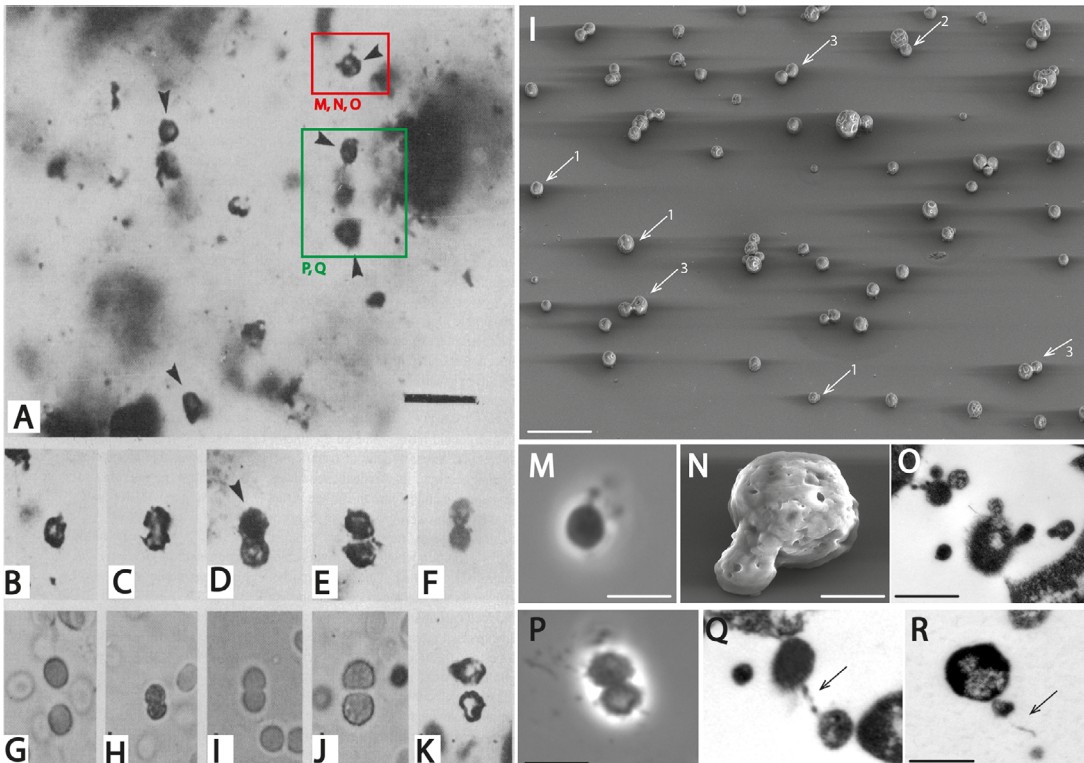

**Appendix 1—figure 21.** Comparison of *EM-P* with the Swartkoppie Formation microfossils. Images (**A-K**) are the images of the Swartkoppie microfossils (originally published by *Knoll and Barghoorn, 1977*; *Knoll and Barghoorn, 1977*). Images (**I-R**) are images of *EM-P* exhibiting morphological similarities with the Swartkoppie microfossils. Image (**A**) shows cells with membrane overhangs (red box), cells in dyads (green box), and individual spherical cells (arrows). Morphologically analogous *EM-P* cells can be seen in the images mentioned below in the red and green boxes. Images (**M, N, & O**) are phase-contrast, SEM, and TEM images of cells with membrane extensions, like the microfossil cell shown within the red box. Images (**P-R**) show a sequence of stages involved in *EM-P*'s cell division. Images (**B-F**) and (**G-K**) show the Swartkoppie microfossils reproducing by binary division. Morphologically similar *EM-P* cells were shown in image (**L**) (white arrows). Numbers next to the arrows indicate different stages of cell division. Scale Bar: 10 µm (**A**), 0.5 µm (**L & N**), 0.25 µm (**O, Q, & R**) and 1 µm (**M & P**).

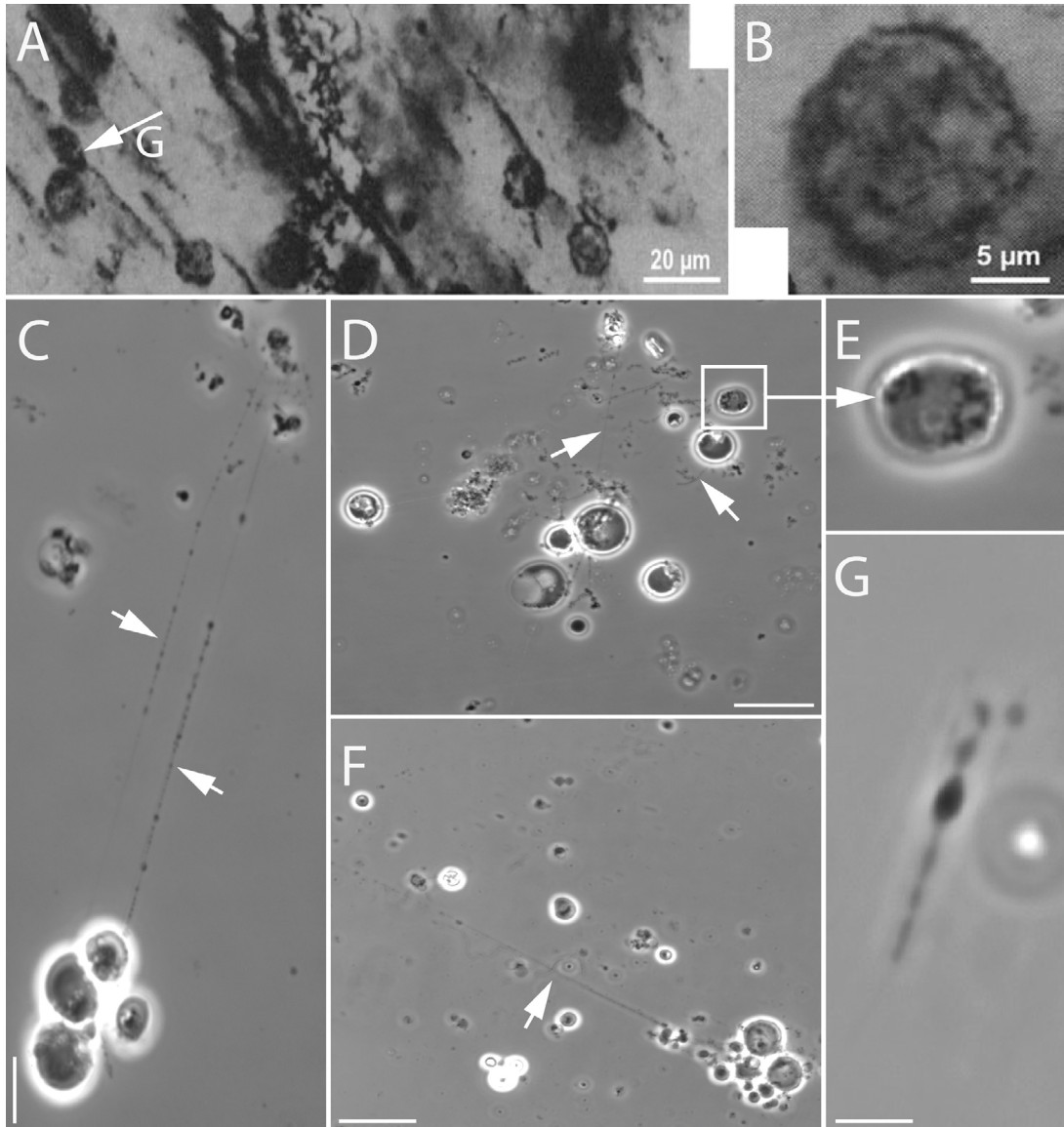

**Appendix 1—figure 22.** Comparison of *EM-P* with the Sheba Formation microfossils. Images (**A & B**) show spherical microfossils reported from the Sheba Formation (originally published by *Hickman-Lewis et al., 2018*; *Hickman-Lewis et al., 2019*). Image (**A**) shows the aggregation of spherical microfossils within organic carbon clasts. Image (**B**) shows a magnified image of spherical cells with unevenly distributed cytoplasm. Images (**C & G**) are morphologically analogous *EM-P* cells. Like the Sheba Formation, microfossils' spherical *EM-P* cells formed filamentous overhangs (arrows in images (**C-F**), *Video 16*). *EM-P* cells were also noticed to have uneven cytoplasm (**E**) (*Figure 2*). The arrow in image A points to spindle-like cells with filamentous extensions that are morphologically similar to *EM-P* cells, as shown in image (**C**). Scale bars: 10 μm (**C**).

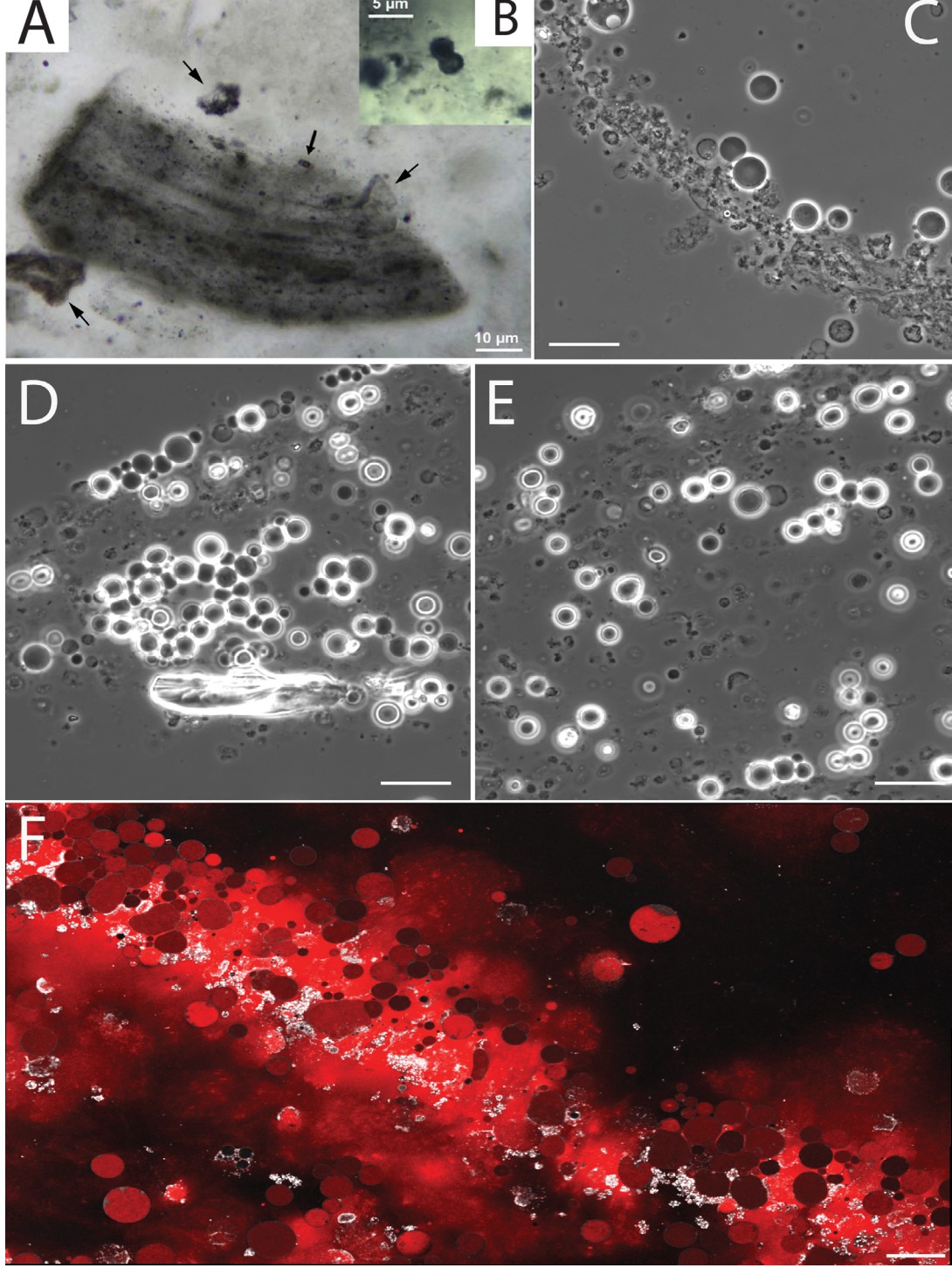

**Appendix 1—figure 23.** Comparison of *EM-P* with the Sheba Formation microfossils. Images (**A** & **B**) show spherical microfossils reported from the Sheba Formation (originally published by *Homann, 2019*; *Buick, 1990*). Image (**B**) shows microfossils reproducing by what appears to be binary fission. Images (**C-F**) are the phase-contrast and STED microscope images of morphologically similar *EM-P* cells. Images (**C-E**) show spherical *EM-P* cells that are surrounded by membrane debris. Image (**F**) shows such debris composed of extracellular DNA (red). Scale bar: 10 mm.

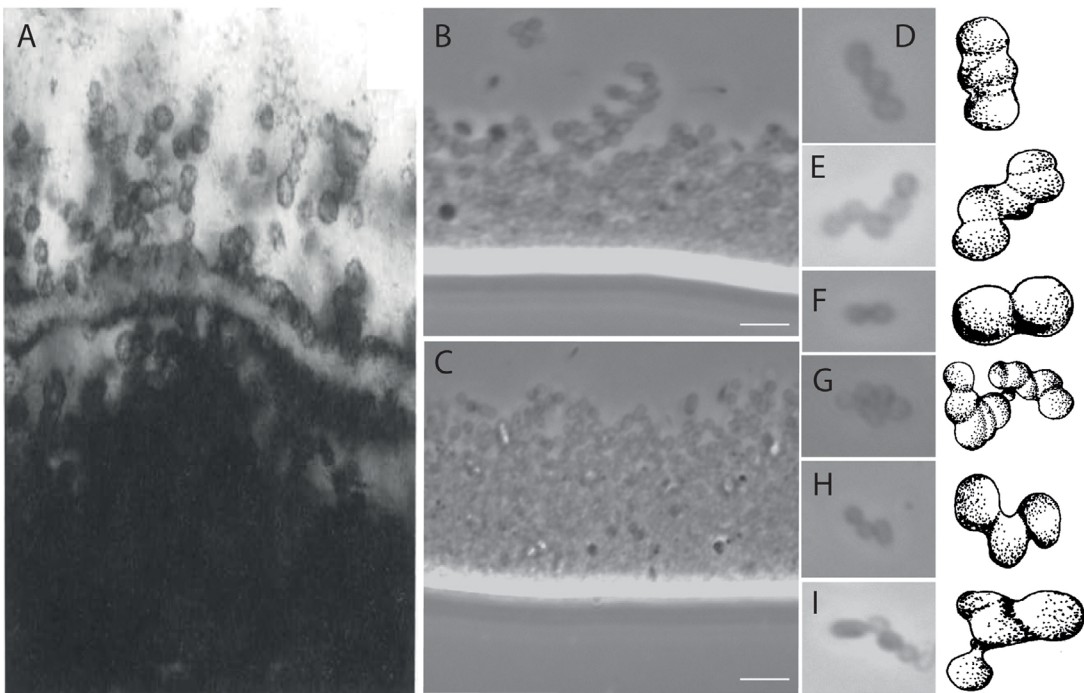

**Appendix 1—figure 24.** Comparison of the Onverwacht group microfossils with *EM-P*. Image (**A**) shows the Onverwacht microfossils (originally published by *Walsh, 1992*). Images (**C & D**) show sub-micrometer size cells *EM-P*, exhibiting morphological similarities with the Onvewacht microfossils. Images (**D-I**) show a cluster of *EM-P* cells organized in different configurations and interpretive drawings of morphologically similar Onverwacht microfossils (originally published by *Walsh, 1992*). Scale bars: (**B & C**) (5 µm).

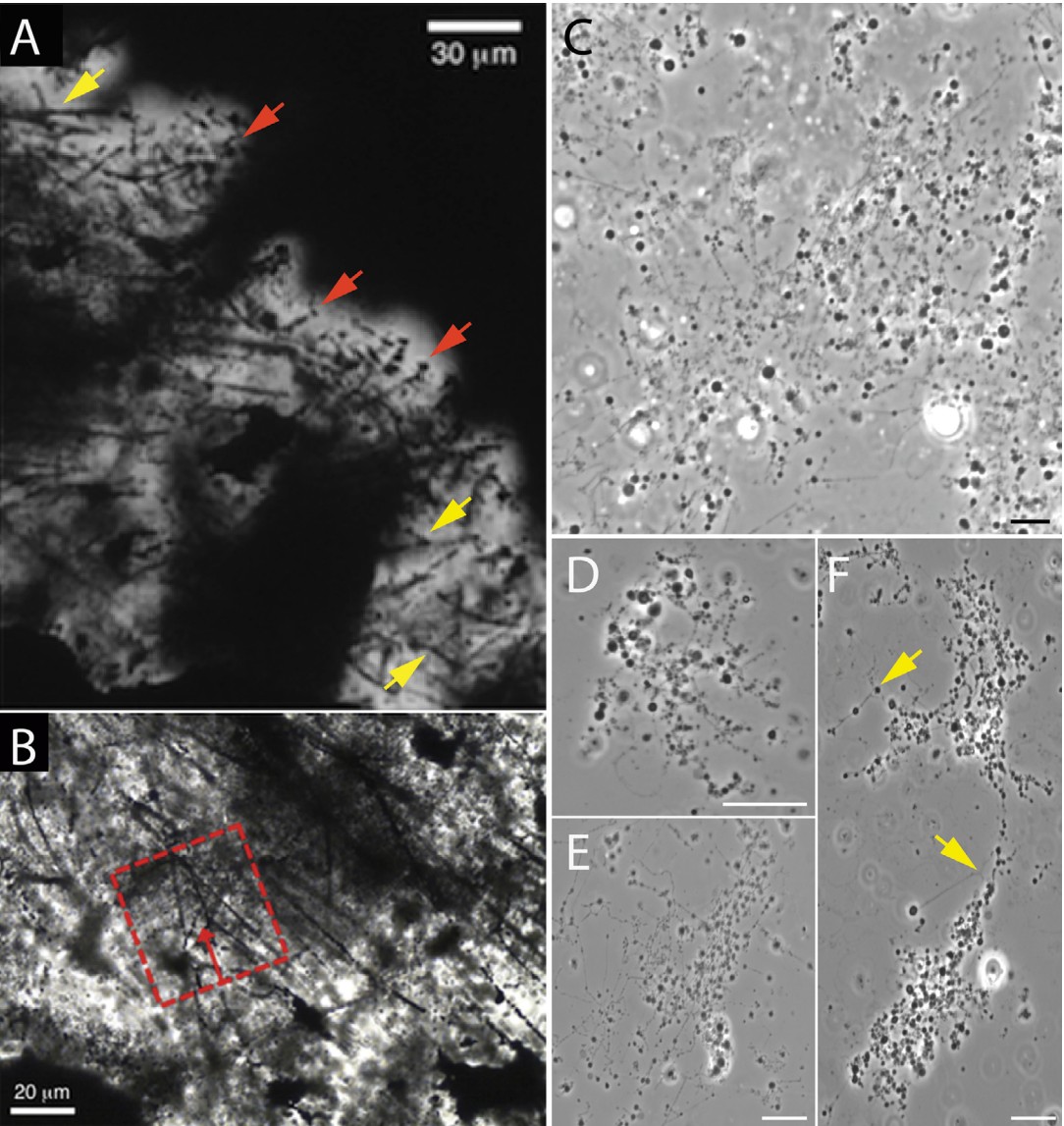

**Appendix 1—figure 25.** Morphological comparison between *EM-P* and the Sulphur Spring Formation microfossils. Images (**A & B**) are microfossils reported from the Sulphur Spring Formations (originally published by *Wacey et al., 2014*). Red arrows in these images point to filamentous structures with spherical inclusions. Images (**C-F**) are morphologically analogous to *EM-P* cells. Similar to images **A & B**, *EM-P* exhibited strings of spherical daughter cells. Arrows in these images point to the branching of the filaments. Scale bars: 10 μm (**C–F**).

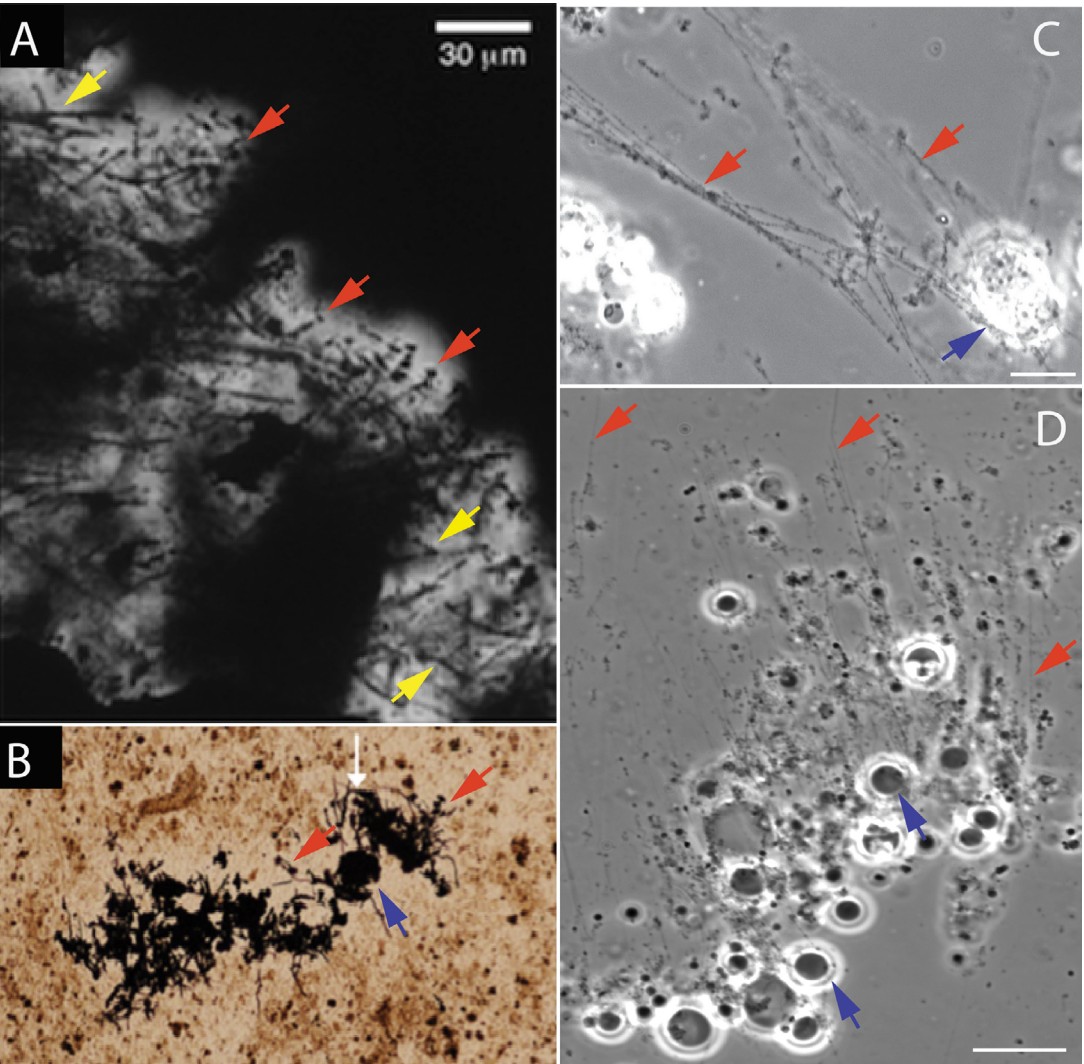

**Appendix 1—figure 26.** Morphological comparison between *EM-P* and the Sulphur Spring Formation microfossils. Images (**A & B**) are microfossils reported from Sulphur Spring Formations (originally published by *Wacey et al., 2014*). Red arrows in these images point to filamentous structures with spherical inclusions. The purple arrow in image (**B**) points to clusters of spherical organic structures from which filamentous structures appear to have originated. Yellow arrows point to the branching within the filamentous structures. Images (**C & D**) show morphologically similar *EM-P* cells. Images show spherical *EM-P* cells with filamentous extensions. Most filamentous extensions contain spherical daughter cells. Scale bars: 10 µm (**B & C**).

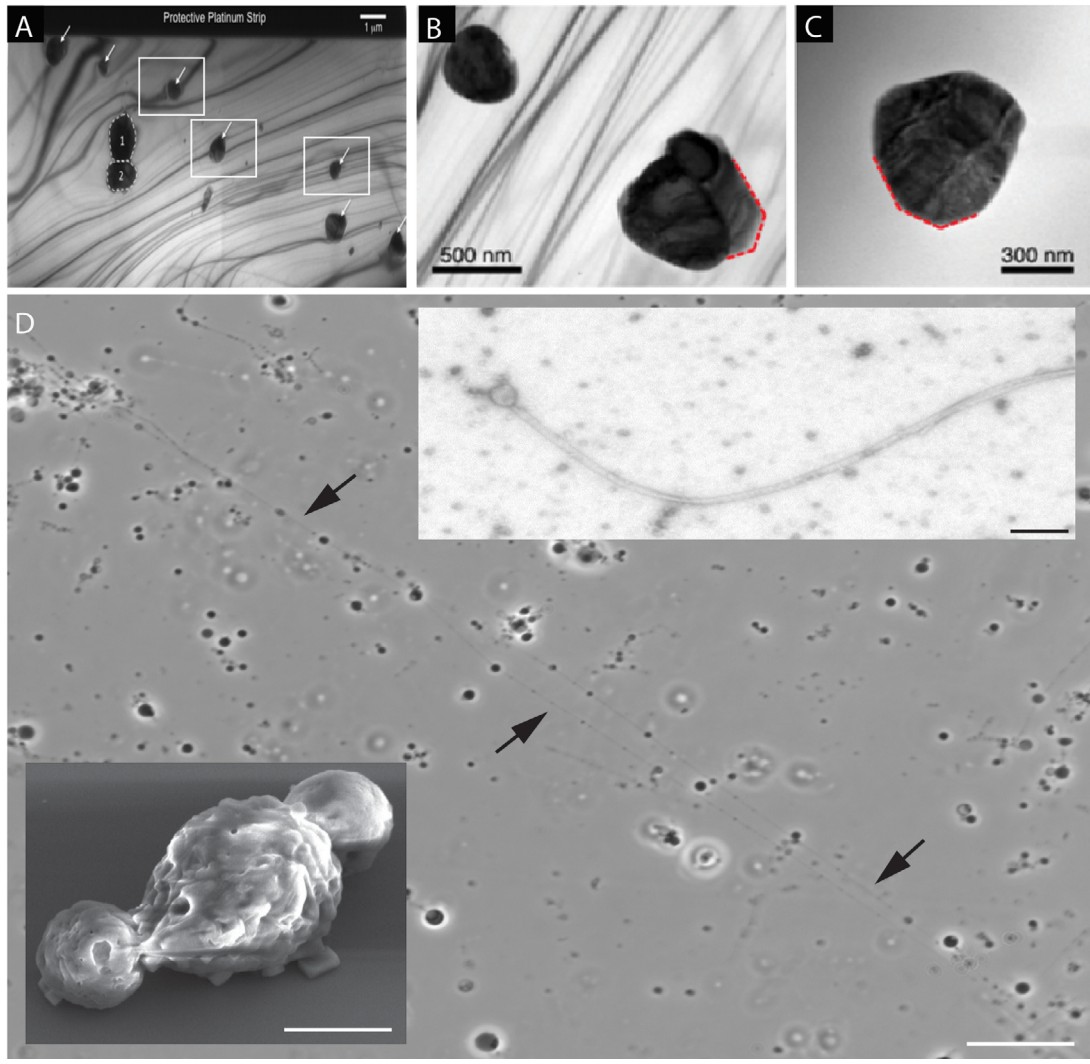

**Appendix 1—figure 27.** Morphological comparison between *EM-P* and the Sulphur Spring Formation microfossils. Images (**A-C**) are microfossils reported from the Sulphur Spring Formations (originally published by *Wacey et al., 2014*). The boxed regions in image A show spherical pyrite-encrusted microfossils within hollow filaments. Images (**B & C**) show close-ups of spherical structures with wrinkly surfaces. Image (**D**) shows hollow filaments with spherical inclusions morphologically analogous to the Sulphur Spring Formation microfossils (Scale bar: 10 μm). Arrows in the image point to spherical daughter cells within the filaments. Insert at the bottom of the image shows an SEM image of such a cell with a wrinkled surface (Scale bar: 1 μm). The Insert image in the top right corner of image (**D**) shows a TEM image of such hollow filaments (Scale bar: 500 nm).

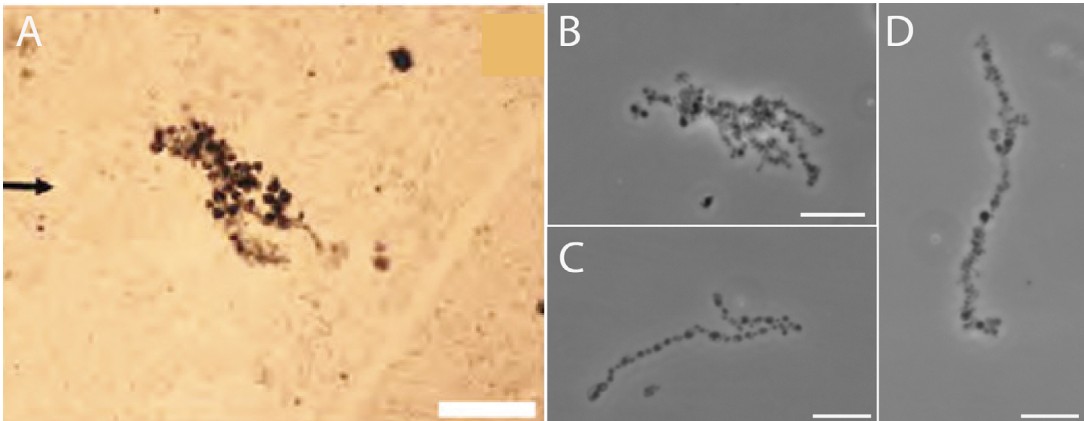

**Appendix 1—figure 28.** Morphological comparison between the Mt. Grant microfossils with *EM-P*. Images (**A & B**) show microfossils reported from the Mt. Grant Formation (originally published by *Sugitani et al., 2007*; *Retallack et al., 2016*). Images (**B-D**) are morphologically analogous to *EM-P* daughter cells. Morphologically analogous *EM-P* cells shown in image (**A**) can be seen in motion in *Video 14*. Scale bars: 100 μm (**A**) & 10 μm (**B–D**).

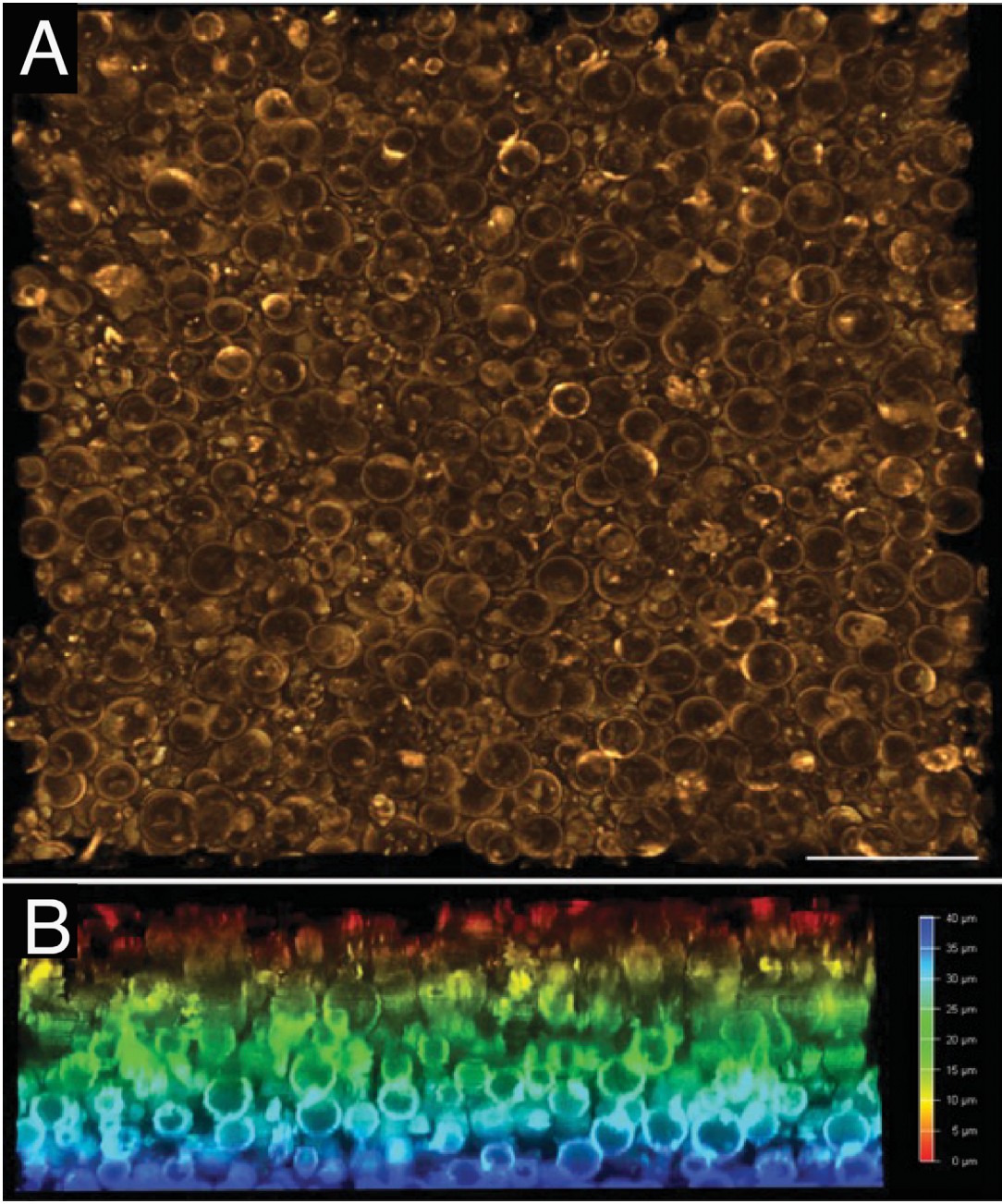

**Appendix 1—figure 29.** Multilayered *EM-P* cells. Images (**A & B**) show a top and lateral view of *EM-P* cells growing in multiple layers at the bottom of the chamber slide. Cells in the above images were stained with membrane stain, FM5-95. Scale bar: 20 μm (**A**).

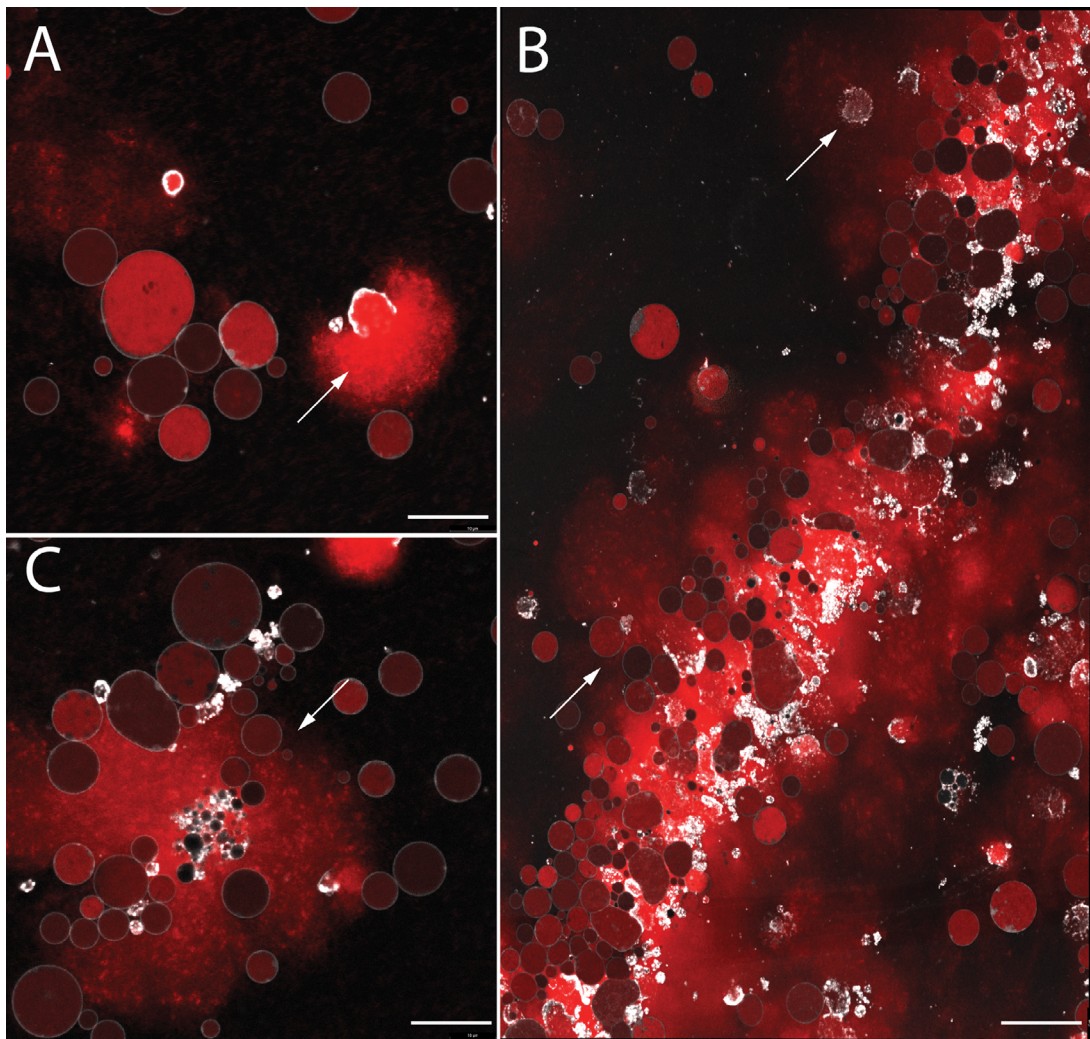

**Appendix 1—figure 30.** Cell lysis and release of DNA observed in *EM-P*. Images (**A & B**) show the lysis and release of cell constituents like DNA (red) into their surroundings. Image (**B**) shows the aggregation of cells within the biofilms with a considerable amount of extracellular DNA. Cells in these images were stained with FM5-95 (membrane, white) and PicoGreen (DNA, red). Scale bars: 10 µm.

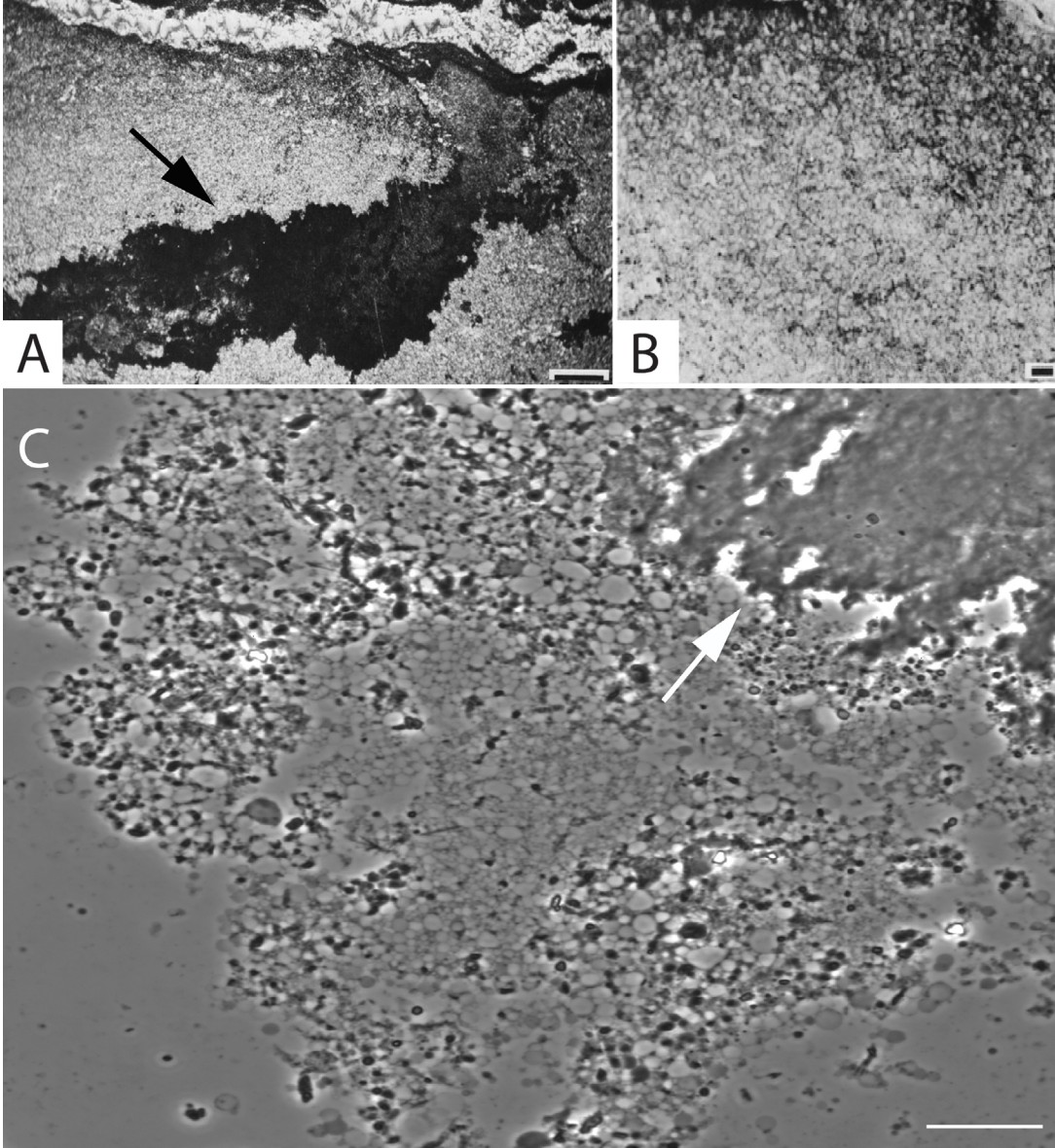

**Appendix 1—figure 31.** Morphological comparison of the North Pole locality microfossils and *EM-P*. Images (**A** & **B**) are the organic structures reported from the North-pole locality (originally published by *Buick, 1990*; *Kaźmierczak and Kremer, 2019*). Image (**D**) shows a similar aggregation of spherical *EM-P* cells observed in our study. Arrows in images (**A** & **C**) point to the membrane debris between the cell aggregates. Further comparison of North Pole locality microfossils and *EM-P* cells is shown in *Appendix 1—figures 32 and 33*. Scale bars: (**A**) (1 mm), (**B**) (100 μm), and (**C**) (20 μm).

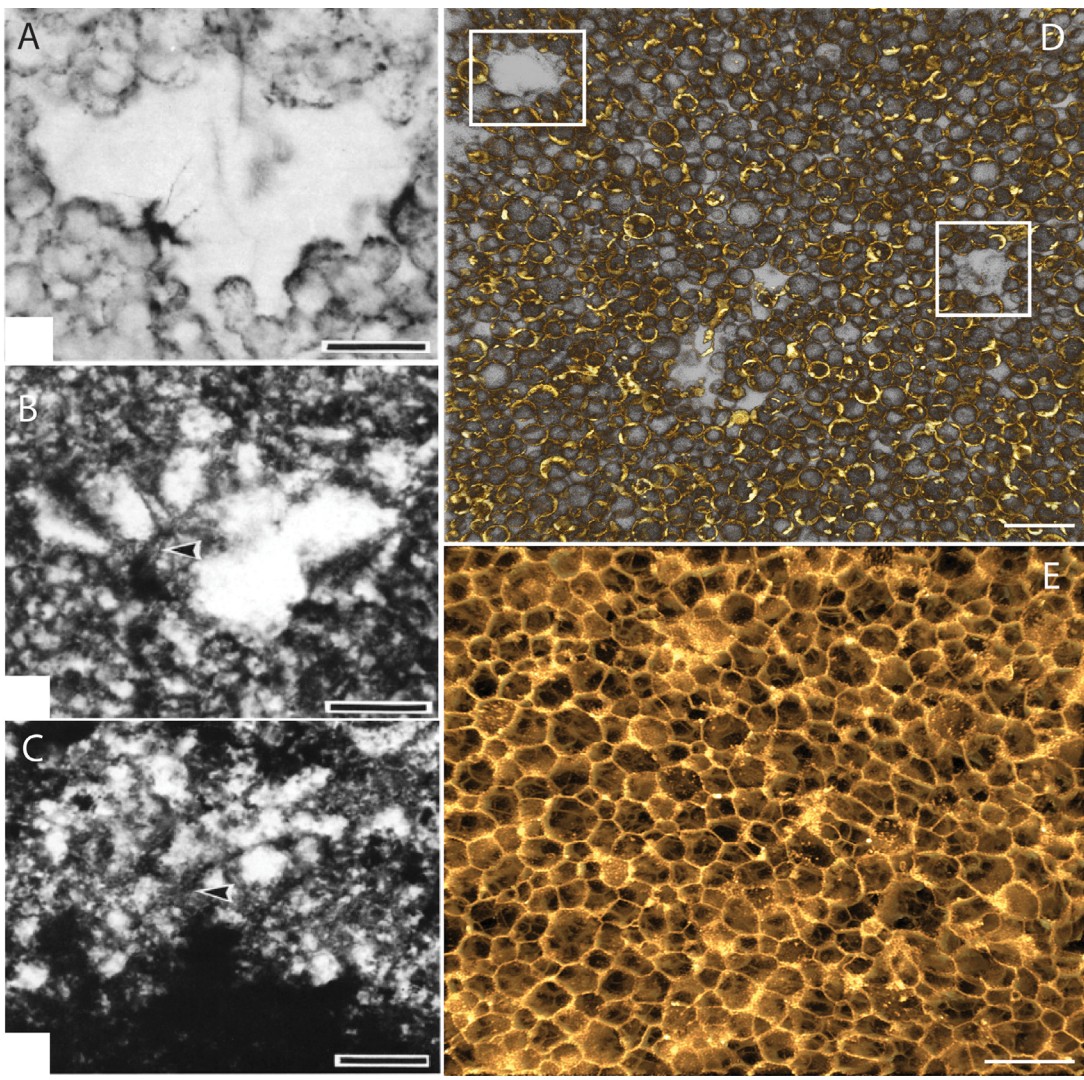

**Appendix 1—figure 32.** Morphological comparison of the North Pole locality microfossils and *EM-P*. Images (**A-C**) are the organic structures reported from the North Pole locality (originally published by *Buick, 1990*; *Kaźmierczak and Kremer, 2019*). They show organic mats composed of individual hollow spherical cells and large gaps within the mats (**A & B**). Images (**D & E**) are similar mat-like organic structures formed by *EM-P*. The highlighted regions in (**D**) show morphologically similar gaps observed in the North Pole locality and *EM-P* biofilms. Scale bars: (**D & E**) (10 μm).

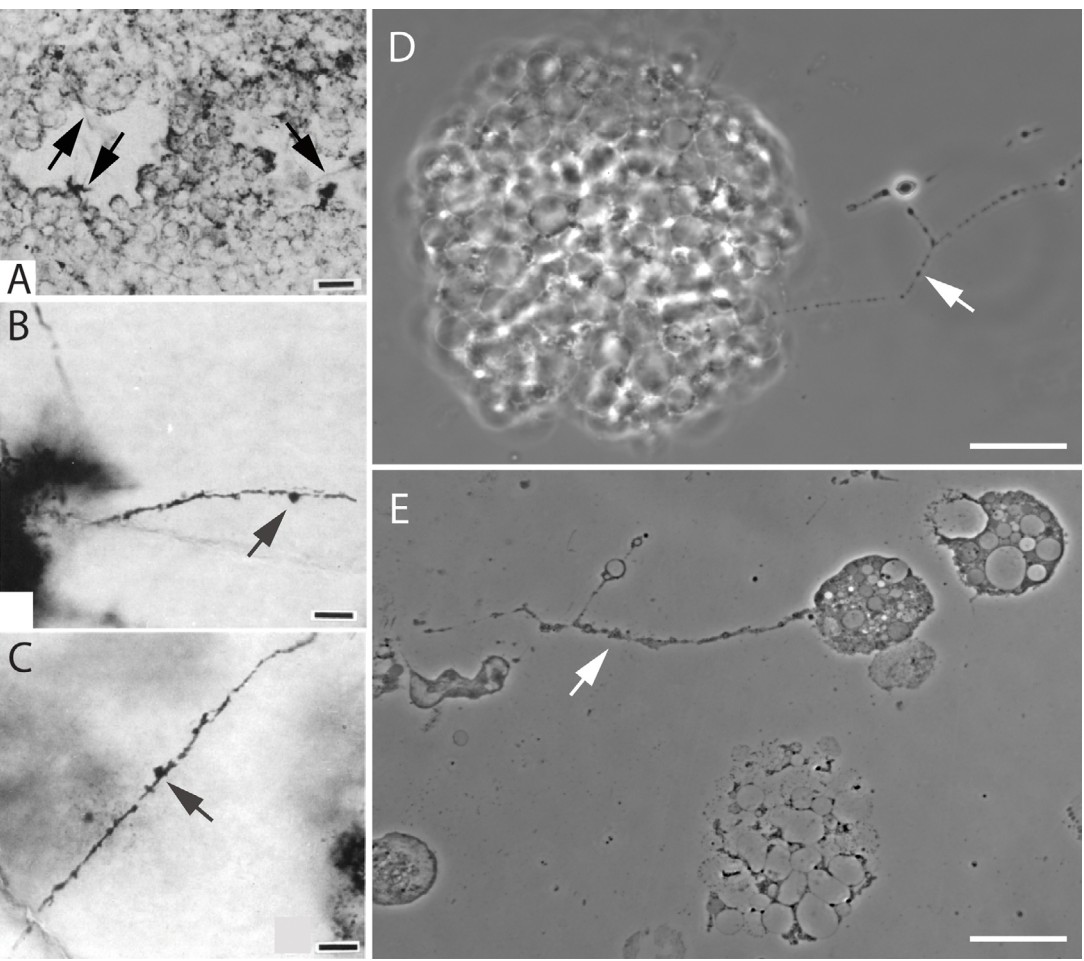

**Appendix 1—figure 33.** Morphological comparison of the North Pole locality microfossils and *EM-P*. Images (**A-C**) are the organic structures reported from the North-pole locality (originally published by *Buick, 1990*; *Kaźmierczak and Kremer, 2019*). Images (**D & E**) show phase-contrast images of *EM-P* cells. Image (**D**) shows an aggregation of hollow *EM-P* vesicles with filamentous structures. The method of their formation is shown in *Videos 5–9*. Scale bars: (**D & E**) (10 μm).

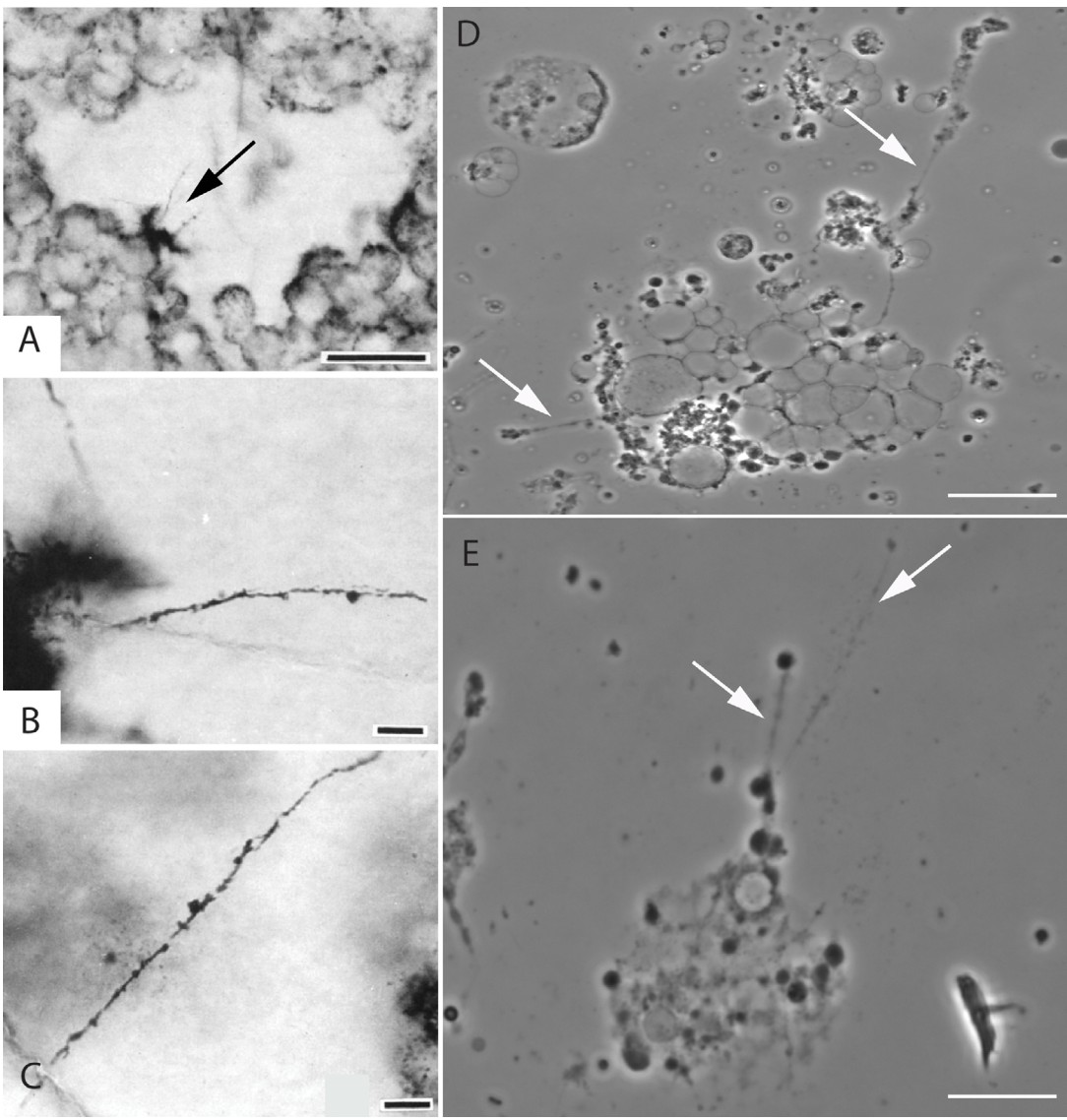

**Appendix 1—figure 34.** Morphological comparison of North Pole locality microfossils and *EM-P*. Images (**A-C**) are the organic structures reported from the North-Pole locality (originally published by *Buick, 1990*; *Kaźmierczak and Kremer, 2019*). Images (**D & E**) show phase-contrast images of *EM-P* cells. Image (**D**) shows an aggregation of hollow *EM-P* vesicles with filamentous structures. The method of their formation is shown in *Videos 5–9*. Scale bars: (**D & E**) (10 μm).

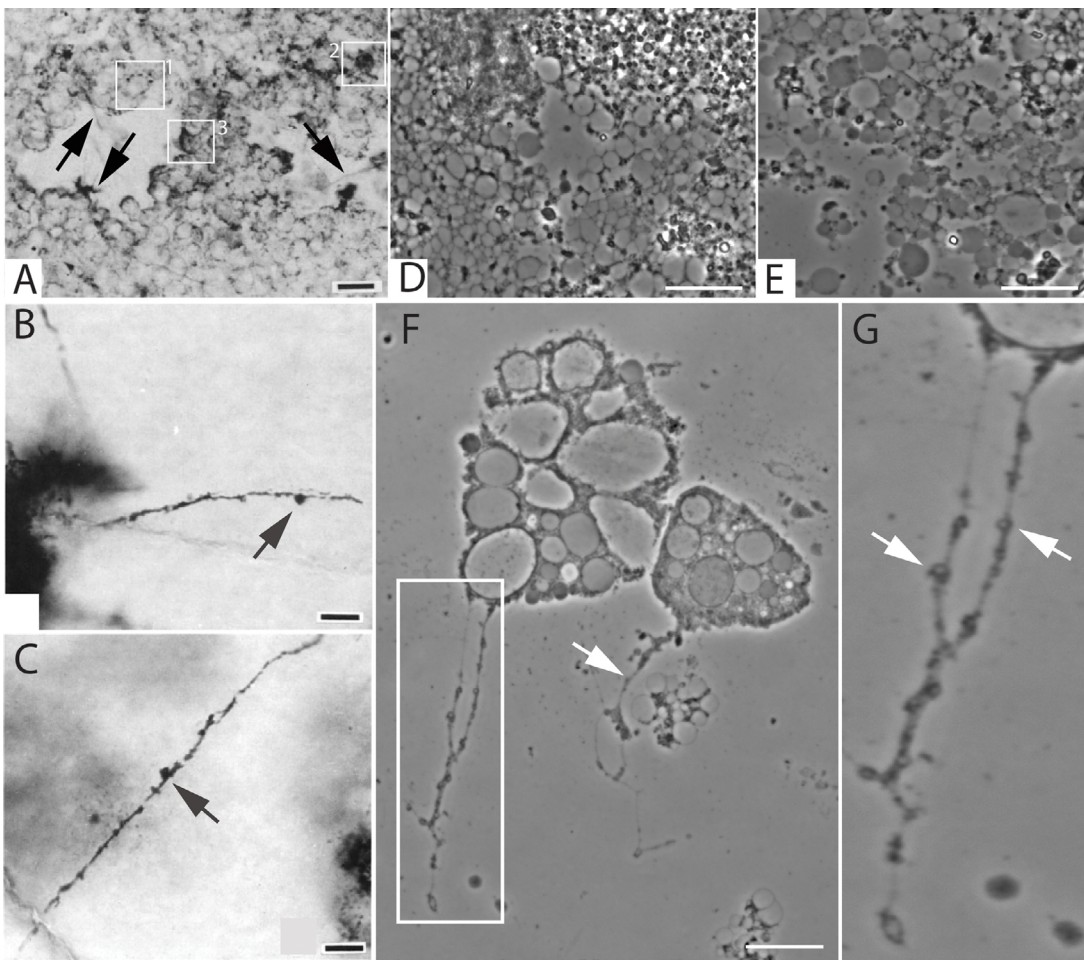

**Appendix 1—figure 35.** Morphological comparison of the North Pole locality microfossils and *EM-P*. Images (**A-C**) are the organic structures reported from the North Pole locality (originally published by *Buick, 1990*; *Kaźmierczak and Kremer, 2019*). Images (**D-G**) show phase-contrast images of *EM-P* cells. Image (**D**) shows an aggregation of hollow *EM-P* vesicles with filamentous structures. The method of their formation is shown in *Videos 5–9*. Arrows in images point to the similarities in the filamentous structure from both north-pole formation and *EM-P*. Scale bars: (**D & E**) (10 μm).

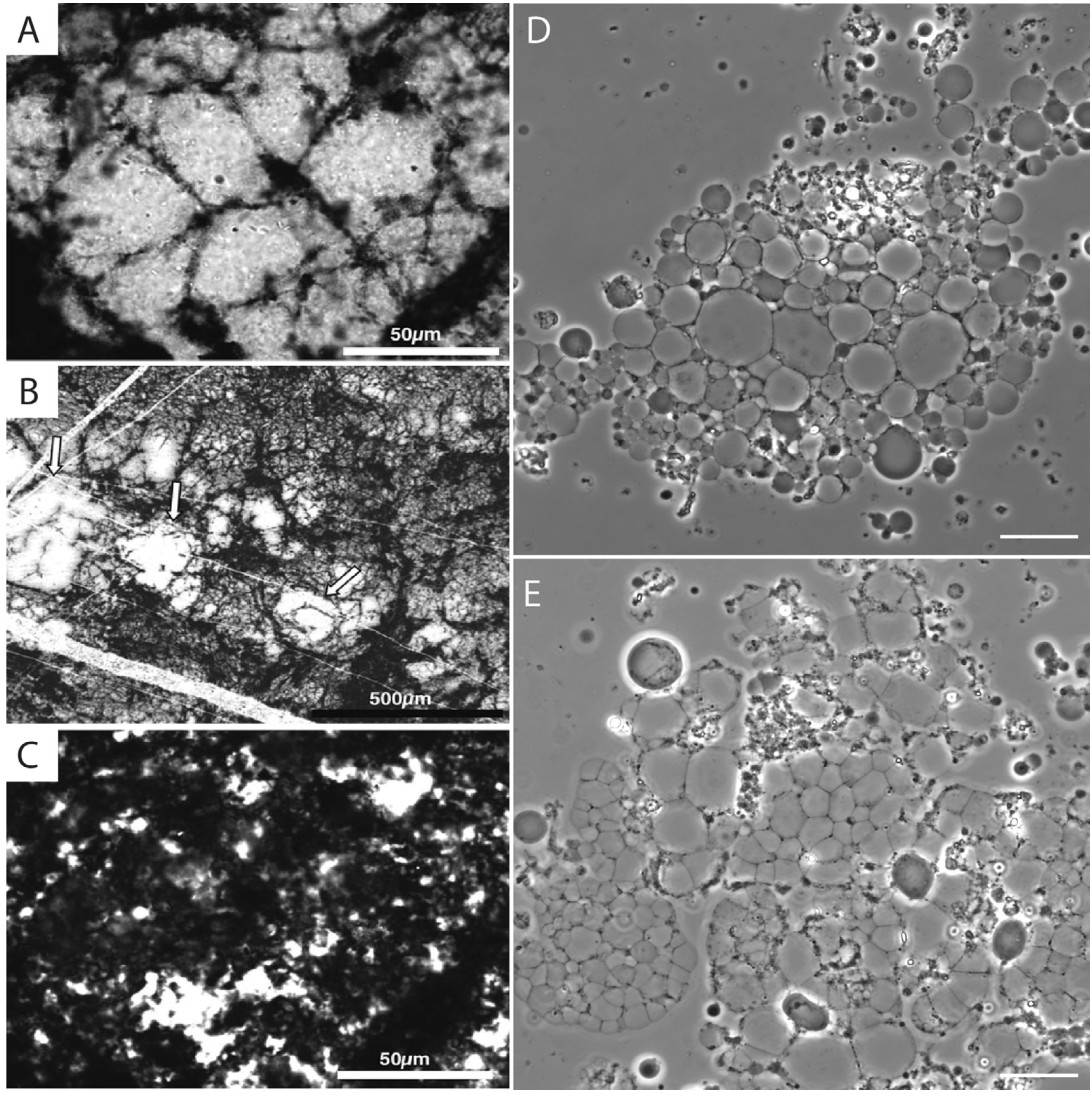

**Appendix 1—figure 36.** Morphological comparison of Strelley Pool Formation (SPF) microfossils with *EM-P*. Images (**A-C**) are the organic structures reported from the SPF (originally published by *Sugitani et al., 2010*);. Images (**D & E**) show phase-contrast images of *EM-P* cells. Highlighted regions in images (**D & E**) show *EM-P* cells forming hexagonal structures, similar to SPF microfossils. Scale bar: (**D & E**) (20 μm).

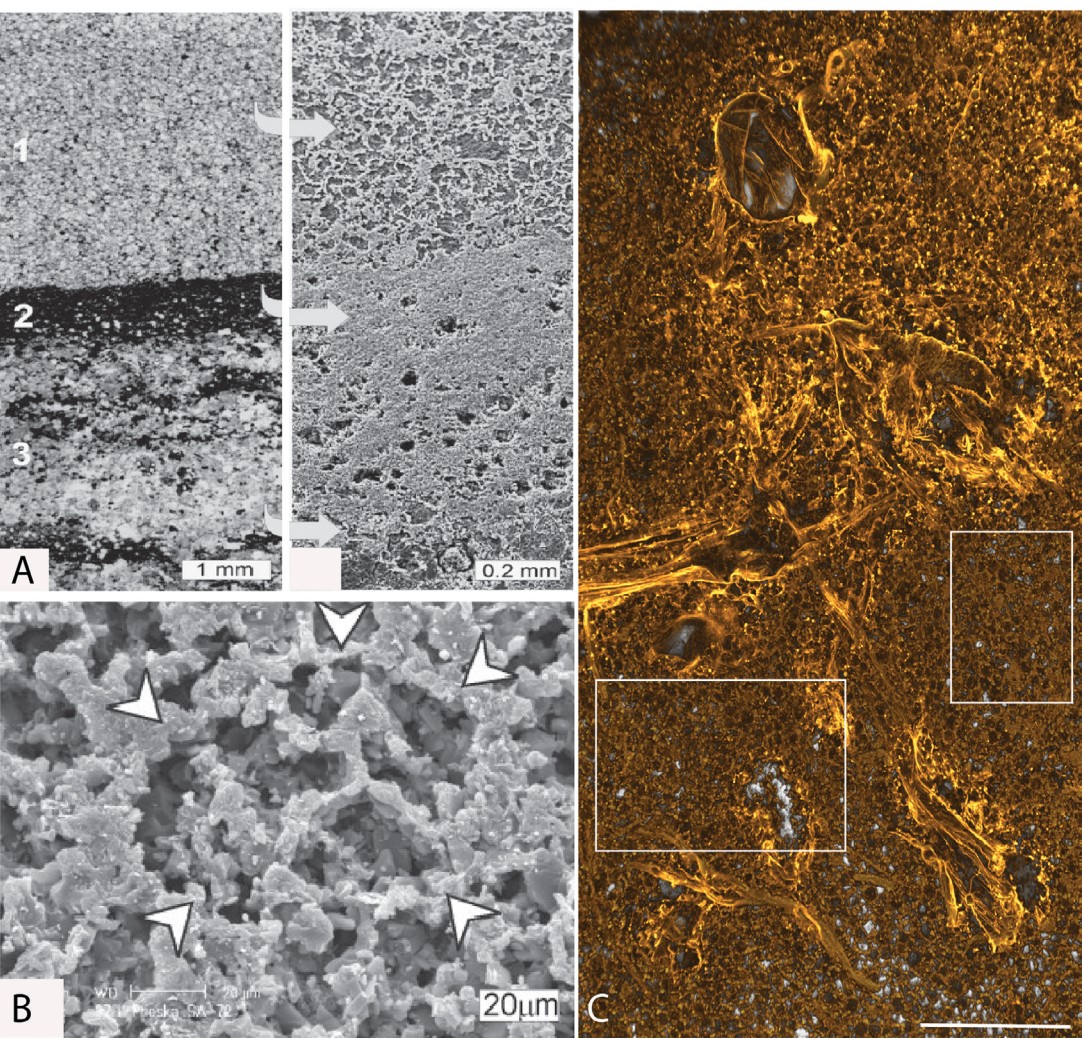

**Appendix 1—figure 37.** A morphological comparison between organic structures was reported from the Nuga Formation and honeycomb structures of *EM-P*. Images (**A & B**) are honeycomb-shaped organic structures reported from the Nauga Formation (originally published by *Kazmierczak et al., 2009*; *Ueno et al., 2006*). Arrows in (**A**) point to the magnified image. Arrows in (**B**) point to the honeycomb structures within the organic structures. Image (**D**) shows morphologically analogous structures formed by *EM-P*s. Cells in images (**D-F**) were stained with membrane stain, FM5-95 (yellow). Boxed regions within Image (**C**) show polygonal structures similar in their morphology to organic structures shown in Image (**B**). The scale bar: 20 µm (**C**).

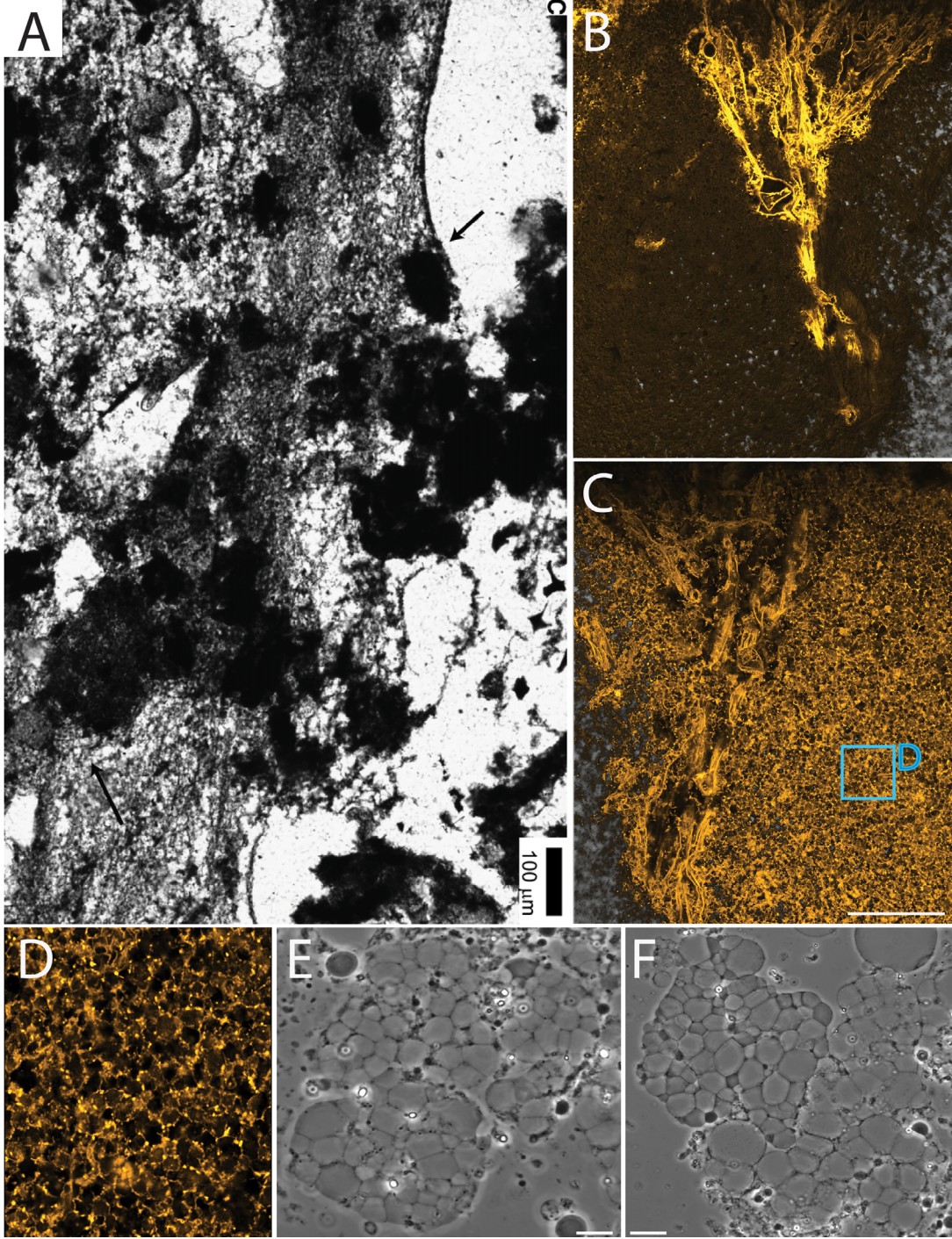

**Appendix 1—figure 38.** Morphological comparison between organic structures reported from the Buck Reef Chert (BRC) Formation and honeycomb structures of *EM-P*. Image (**A**) is a honeycomb-shaped organic structure reported from the BRC (originally published by *Tice, 2009*; *Tice and Lowe, 2004*). Images (**B & C**) are anterior and posterior views of morphologically analogous *EM-P's* membrane debris. Cells in images (**B-D**) are stained with membrane stain, FM5-95 (yellow). Image (**D**) is the magnified region of the biofilm showing honeycomb structures. Images (**E & F**) are phase-contrast images of large spherical *EM-P* cells with polygonal vacuoles undergoing clumping to form large honeycomb-shaped mats. Scale bars: (**B & C**) (100 μm) and (**E & F**) (10 μm).

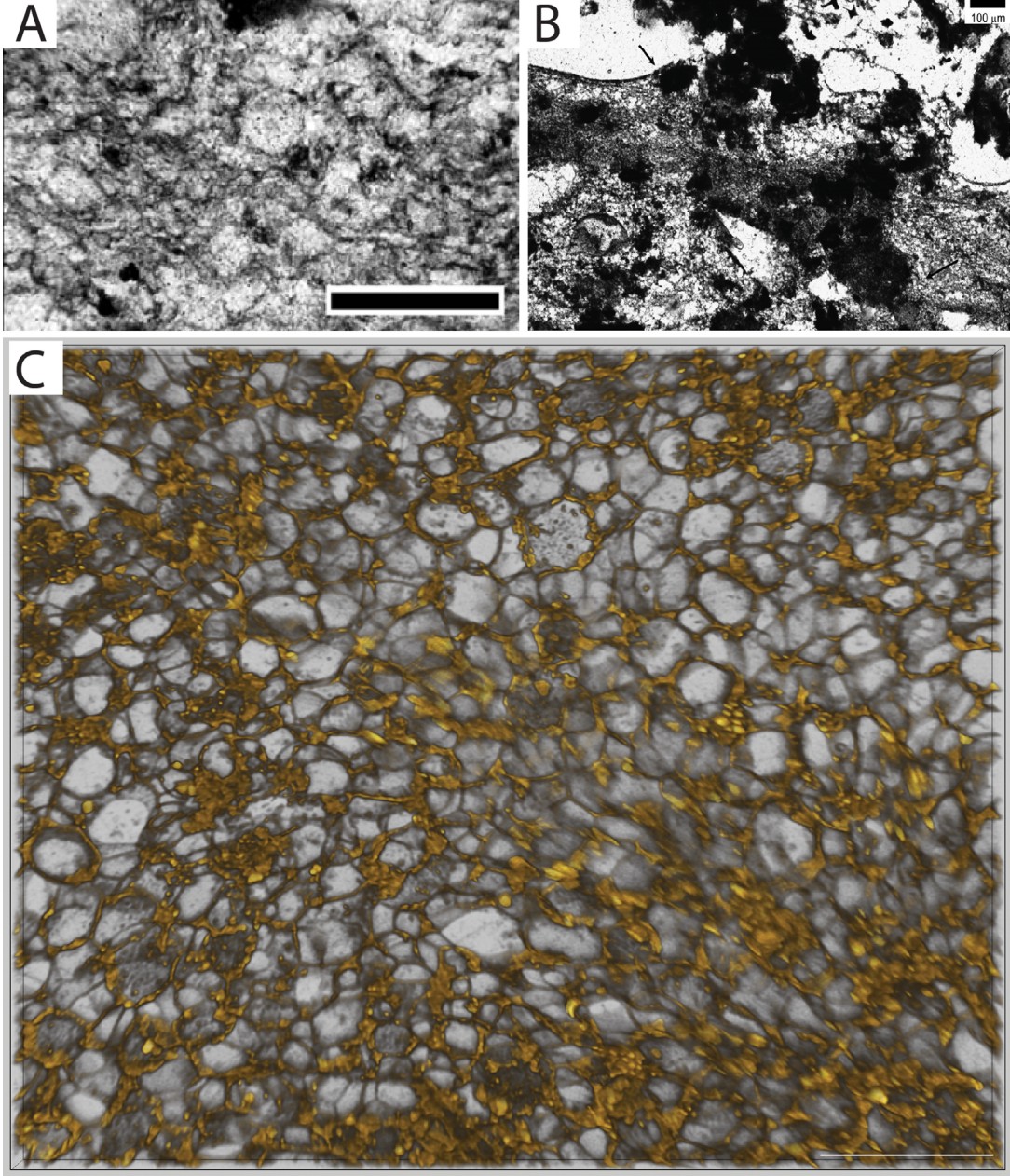

**Appendix 1—figure 39.** Morphological comparison between organic structures reported from the BRC Formation and honeycomb structures of *EM-P*. Images (**A & B**) show the bifurcating or very fine strands of carbonaceous matter within the BRC formation (originally reported by *Tice and Lowe, 2006*; *Schopf et al., 2017*). Image (**C**) is the morphologically similar structure formed by *EM-P*. The subsequent disintegration of these honeycomb structures into strands of filamentous membrane is shown in *Appendix 1—figure 40B and C*. Scale bars: (**C**) (20 µm).

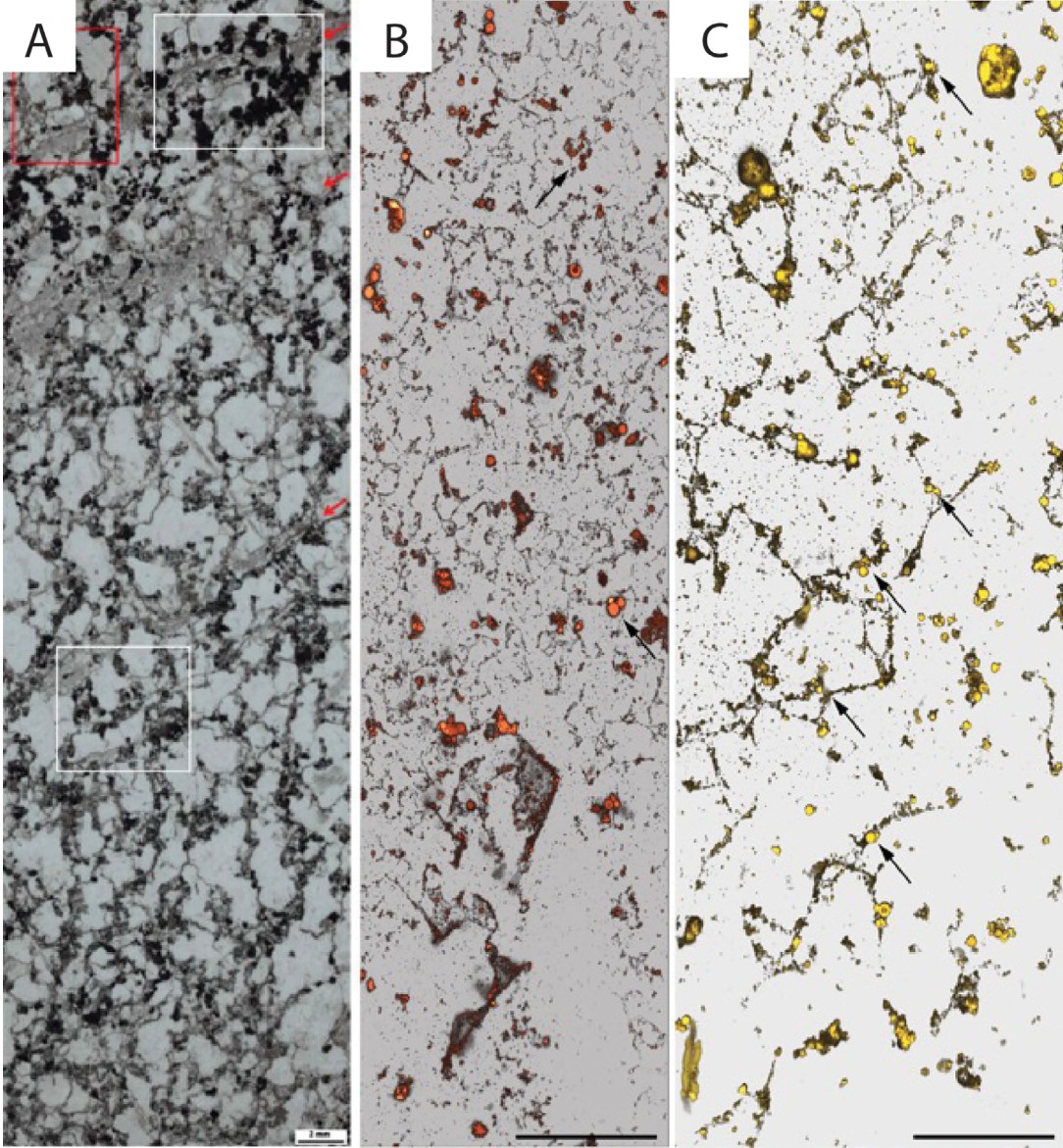

**Appendix 1—figure 40.** Morphological comparison of honeycomb structures reported from the Turee Creek Formations and *EM-P*. Image (**A**) shows a tangled network of microfossil structures reported from the Turee Creek Formation (originally published by *Barlow and Van Kranendonk, 2018*; *Sugitani et al., 2009*). Images (**B & C**) are morphologically analogous structures formed by *EM-P*. Cells in these images are stained with membrane stain, FM5-95, and imaged using a STED microscope. Boxed regions and arrows in image (**A**) highlight the spherical cells closely associated with honeycomb-shaped organic structures (*Appendix 1—figures 36 and 39C*). Arrows in (**B & C**) point to spherical daughter cells attached to membrane debris. Based on the morphological resemblance, we propose that the Turee Creek organic structures were leftover membrane debris of *EM-P*-like cells rather than an entangled network of filamentous microfossils. Scale bars: (**A**) (2 mm), (**B & C**) (50 µm).

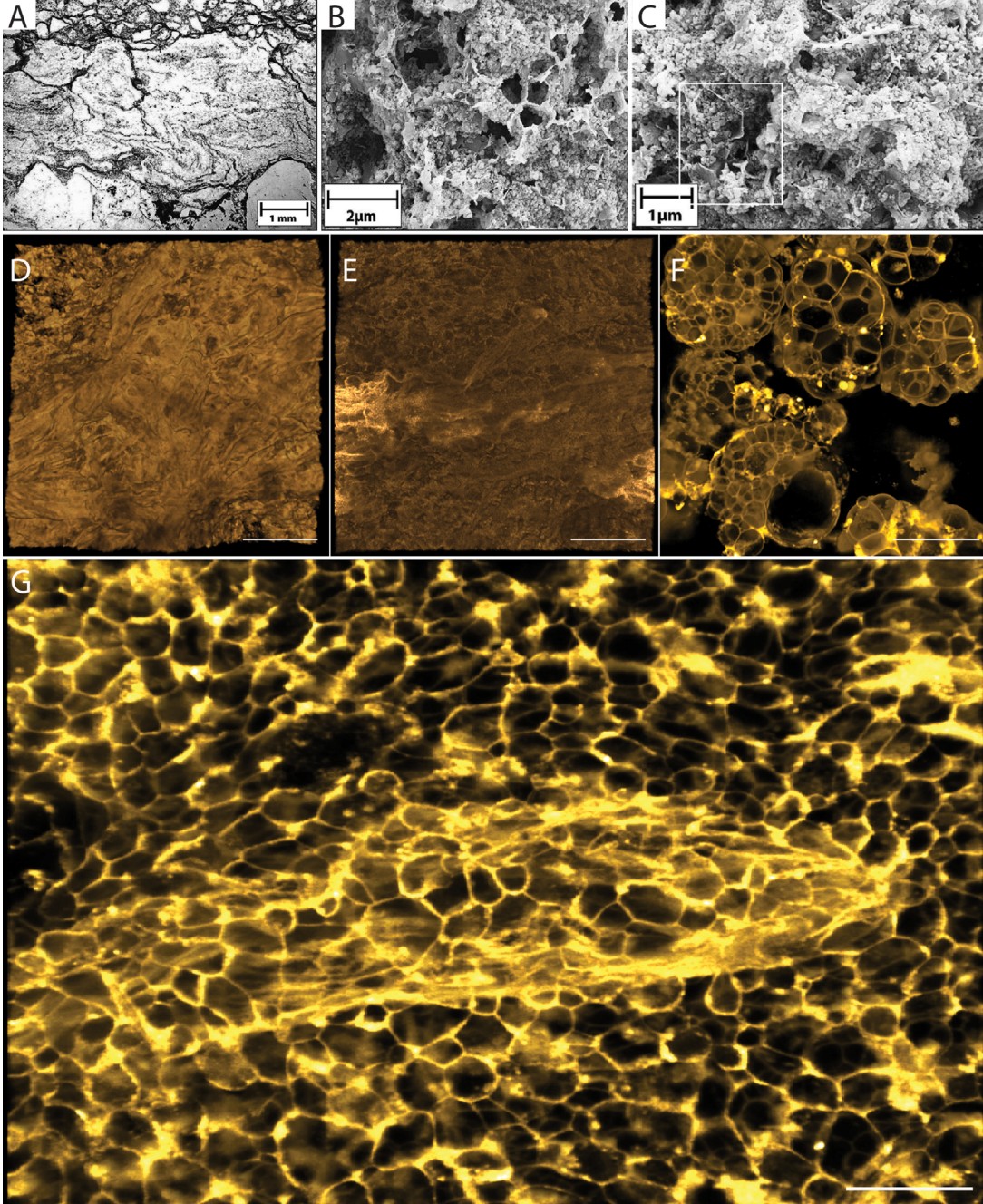

**Appendix 1—figure 41.** Morphological comparison of organic structures reported from the Moodies Group with *EM-P*. Images (**A-C**) are the microbial mats reported from the Moodies Group. Honeycomb-like polygonal structures (*Gamper et al., 2012*; *Westall et al., 2006*). Images (**D-G**) are morphologically analogous structures observed in *EM-P* incubations. Images (**D & E**) show the encrusted surface and interior of the biofilm. The hexagonal structures underneath the surface can be seen in image (**E**). Image (**F**) shows the individual *EM-P* cells that constitute the biofilm. Image (**G**) shows the deeper layers of the biofilm with a distinctive honeycomb pattern. Scale bars: (**D-G**) (20 µm).

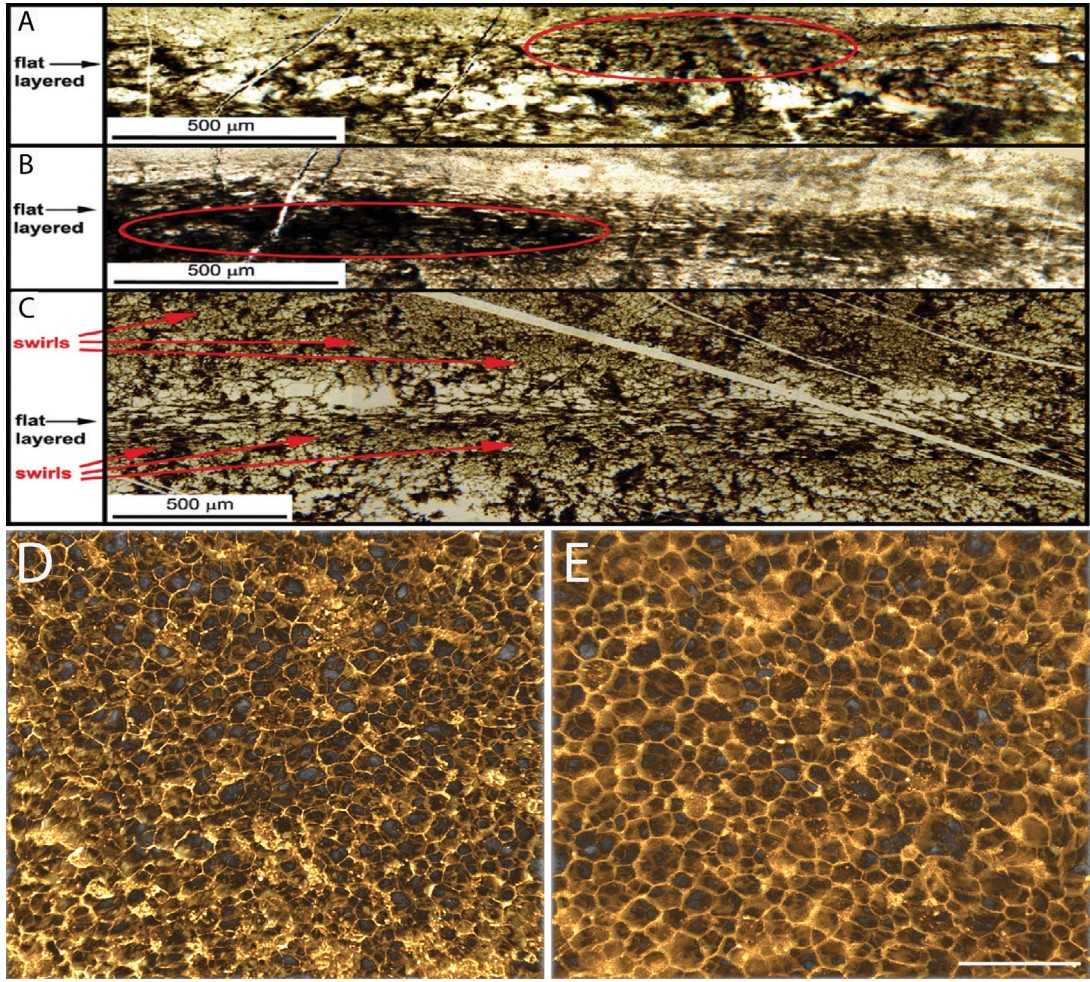

**Appendix 1—figure 42.** Morphological comparison between honeycomb structures of *EM-P* organic structures reported from Strelley Pool Formation (SPF). Images (**A-C**) are honeycomb-shaped organic structures reported from SPF (originally published by *Schopf et al., 2017*; *Gamper et al., 2012*). The circled region in Image (**B**) points to flattened honeycomb structures. Images (**D & E**) show morphologically analogous honeycomb-like structures observed in *EM-Ps* incubations. Cells in images (**D-F**) were stained with membrane stain, FM5-95. The scale bar: 20 μm (**D**).

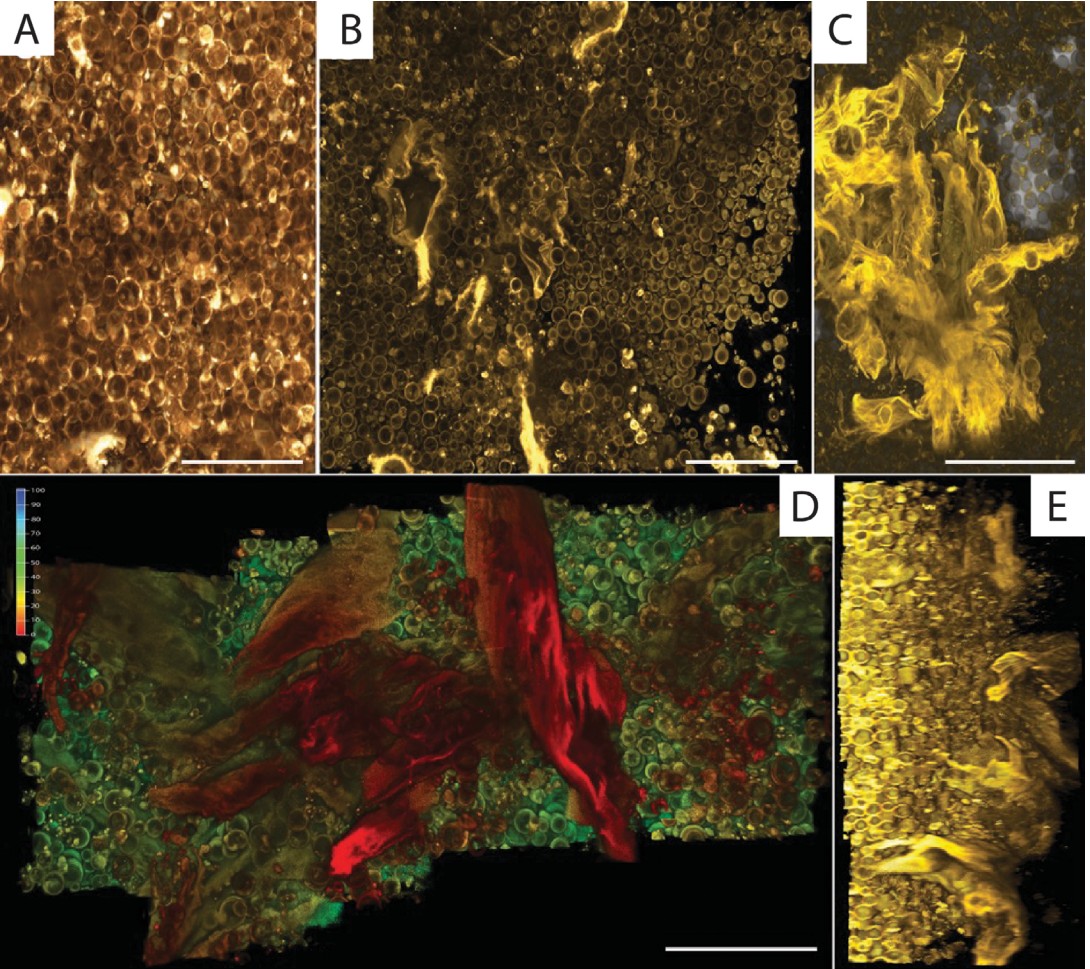

**Appendix 1—figure 43.** Membrane debris formation. Images (**A-E**) show different stages involved in forming fabric-like membrane debris formation. Image (**A**) shows a biofilm with stacks of spherical *EM-P* cells. Image (**B**) shows the membrane debris formed during the cell lysis. Image (**C**) shows the gradual accumulation and increase in the surface area of membrane debris. Image (**D**) shows the top view of the biofilm with sheets of membrane debris growing out of the biofilm surface. A later stage of the biofilm largely engulfed in the membrane debris is shown in *Appendix 1—figure 45*. Image (**E**) is the lateral view of the *EM-P* biofilm shown in (**D**). Cells and membrane debris in these images were stained with membrane stain, FM5-95, and imaged using a STED microscope. Scale bars: 20 μm.

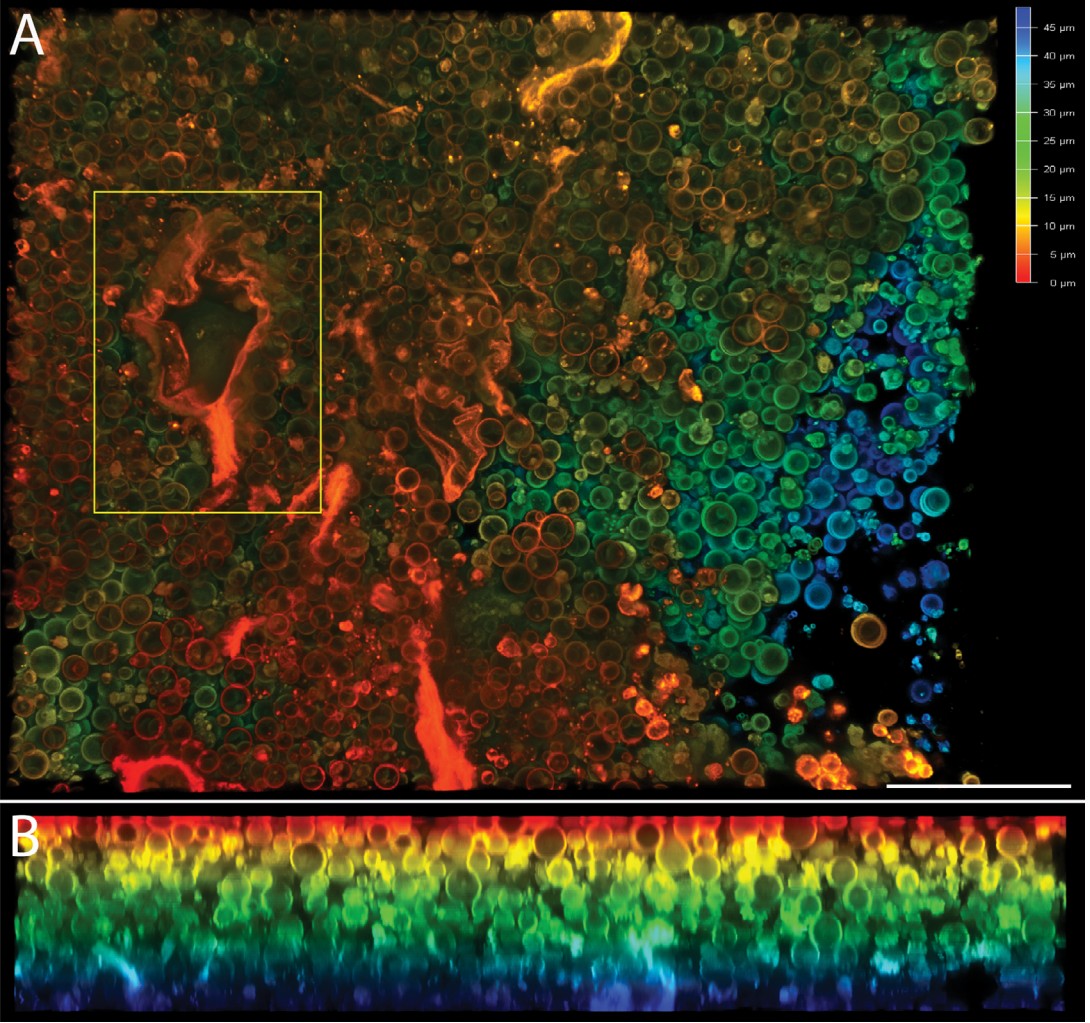

**Appendix 1—figure 44.** Cell lysis and formation of membrane debris within multilayered *EM-P* cells. Images (**A & B**) show a top and lateral view of *EM-P* cells growing in multiple layers at the bottom of the chamber slide. Cells in the above images were stained with membrane stain, FM5-95. The boxed region in image (**A**) shows the membrane debris formed from the lysis of cells. Scale bar: 20 µm (**A**).

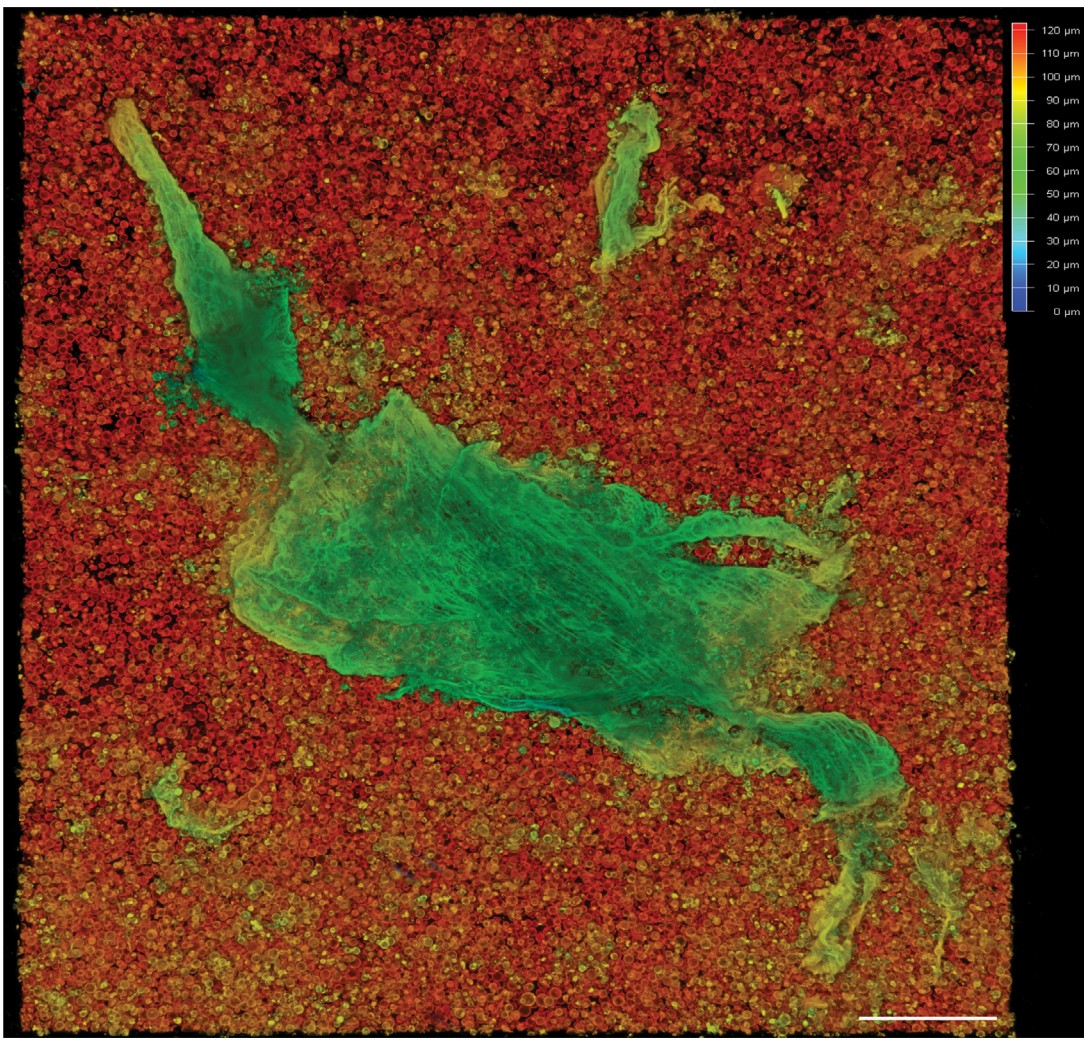

**Appendix 1—figure 45.** Membrane debris of *EM-P*. The image shows membrane debris of *EM-P* forming a layer over a mat of multilayered cells. Based on the morphological similarities with laminated structures, we presume such structures were formed by a similar process. The lateral view of the image, along with its morphological comparison with α-type laminations reported from BRC, was shown in ***Appendix 1—figure 46***. Membranes were stained with Nile red and were imaged using a confocal microscope. Scale bar: 100 µm.

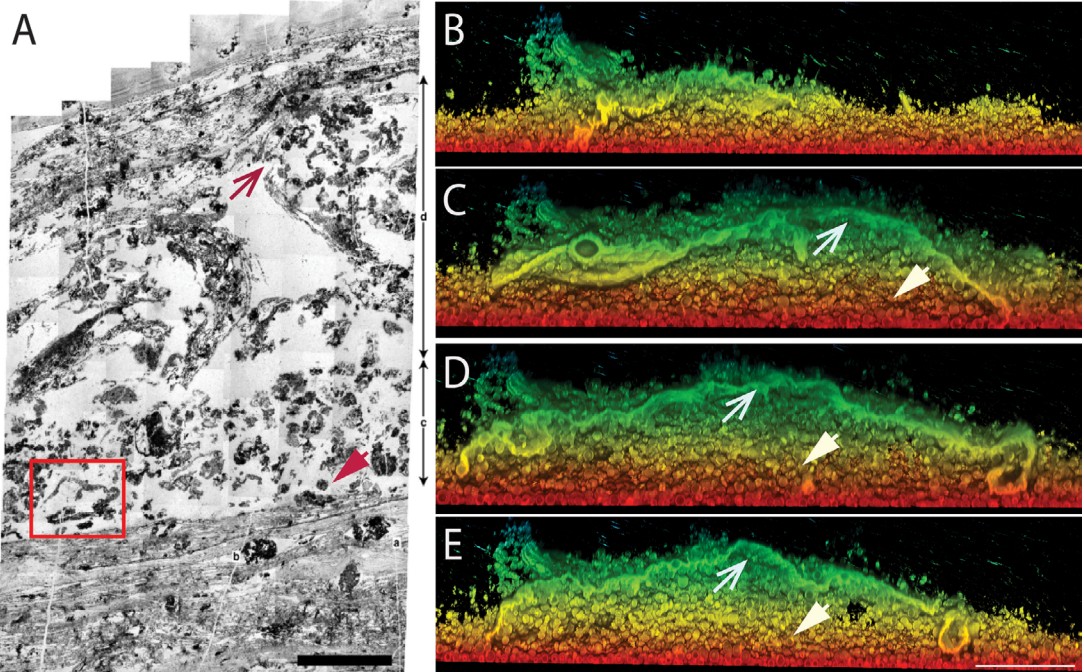

**Appendix 1—figure 46.** Morphological comparison between the BRC laminations and membrane debris of *EM-P*. Image (**A**) shows α-type laminations reported from the BRC (originally published by *Tice, 2009*; *Tice and Lowe, 2004*). Two filamentous layers with hollow spaces between them can be seen in image (**A**). The hollow space in between is filled with filamentous membrane debris and spherical inclusions. Images (**B-E**) show lateral sections of *EM-P* membrane debris, as shown in *Appendix 1—figure 45*. Open and closed arrows in all the images point to the top layer of the membrane enclosure and the bottom cell layer, respectively. The hollow space in between is filled with membrane debris and spherical cells. We presume that the lysis of these cells could have produced membrane debris like the ones observed in image (**A**). In support of this presumption, membrane debris similar to β-type laminations (indicated by the red box) was observed in *EM-P* batch cultures (*Appendix 1—figures 51 and 54*). Also, see *Video 19*. Scale bars: 5 mm (**A**) and 100 μm (**E**).

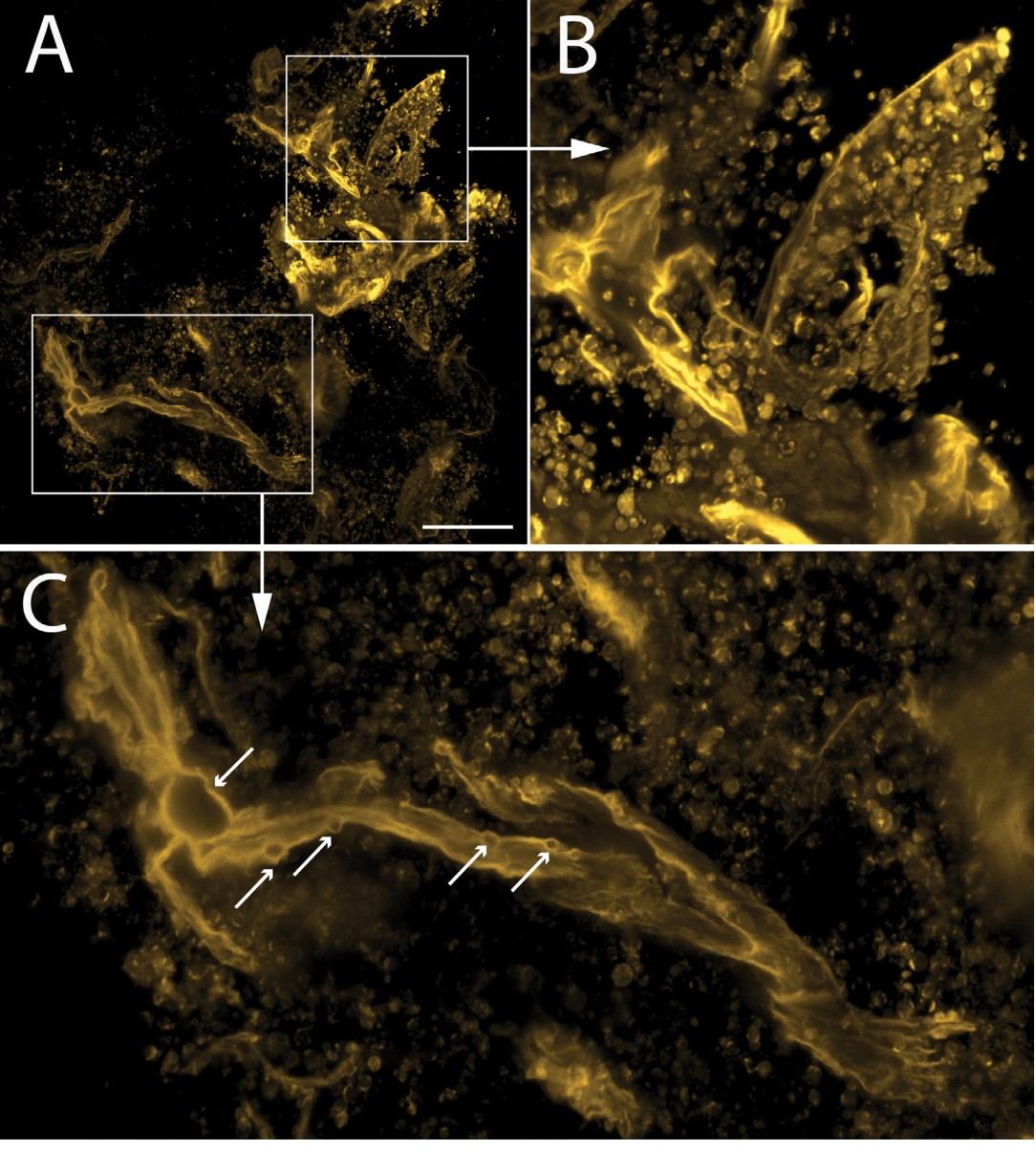

**Appendix 1—figure 47.** Membrane debris of *EM-P*. The image shows *EM-P* cells covered in membrane debris. Images (**B & C**) are the magnified regions of image (**A**). Arrows in (**C**) point to the naturally formed lenticular gaps within the membrane debris. Scale bar: 100 μm (**A**).

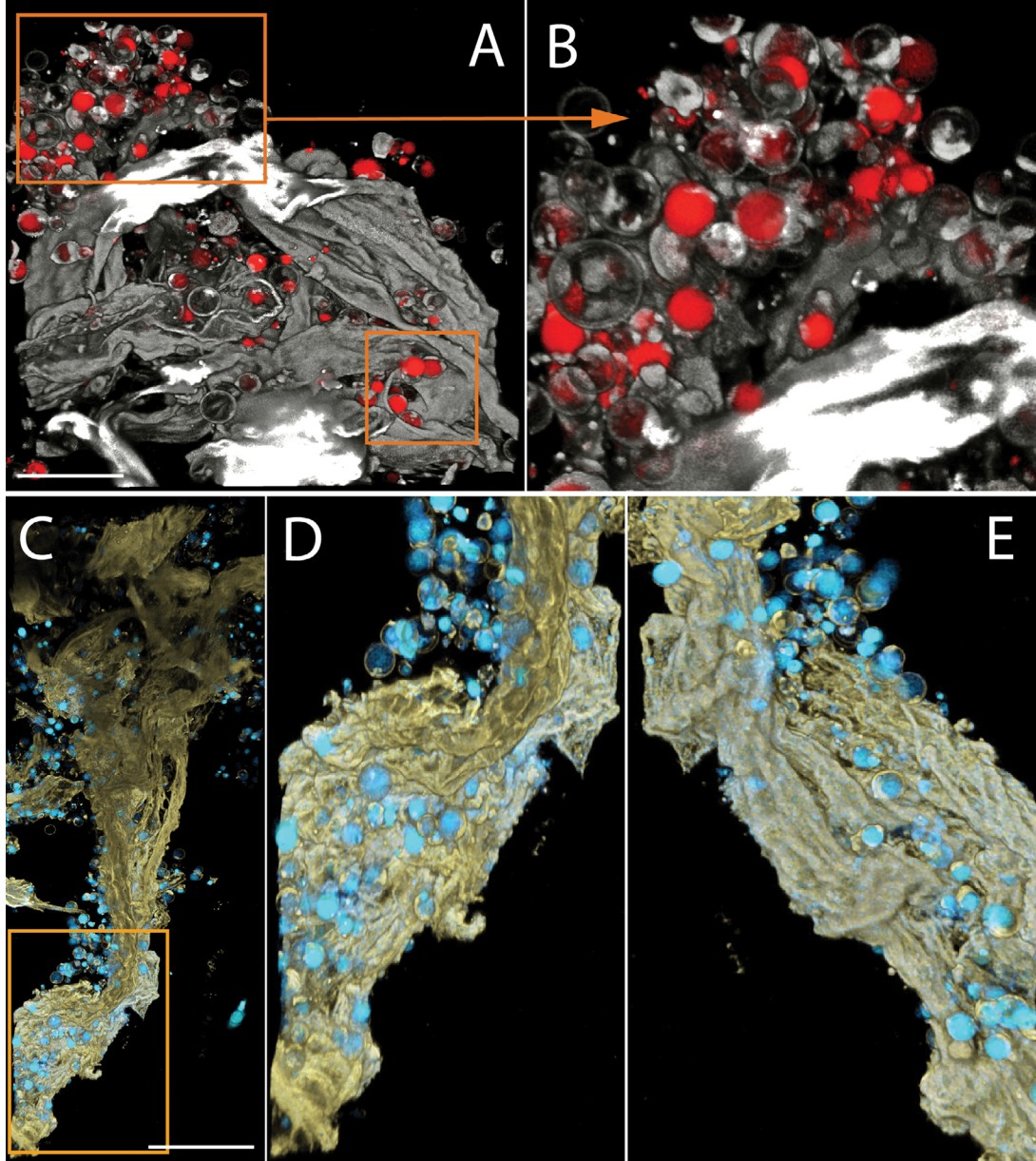

**Appendix 1—figure 48.** Membrane debris of *EM-P*. The image shows *EM-P* covered in membrane debris. Image (**B**) shows the magnified regions of image (**A**). Images (**D & E**) are the anterior and the posterior view of the membrane debris with cells. These images highlight the fabric-like texture of the membrane debris. Cells in these images were stained with FM5-95 membrane, white in (**A & B**) and gray in (**C-E**) and PicoGreen (DNA, red in **A & B** and cyan in **C-E**). Scale bar: 50 µm (**A & C**).

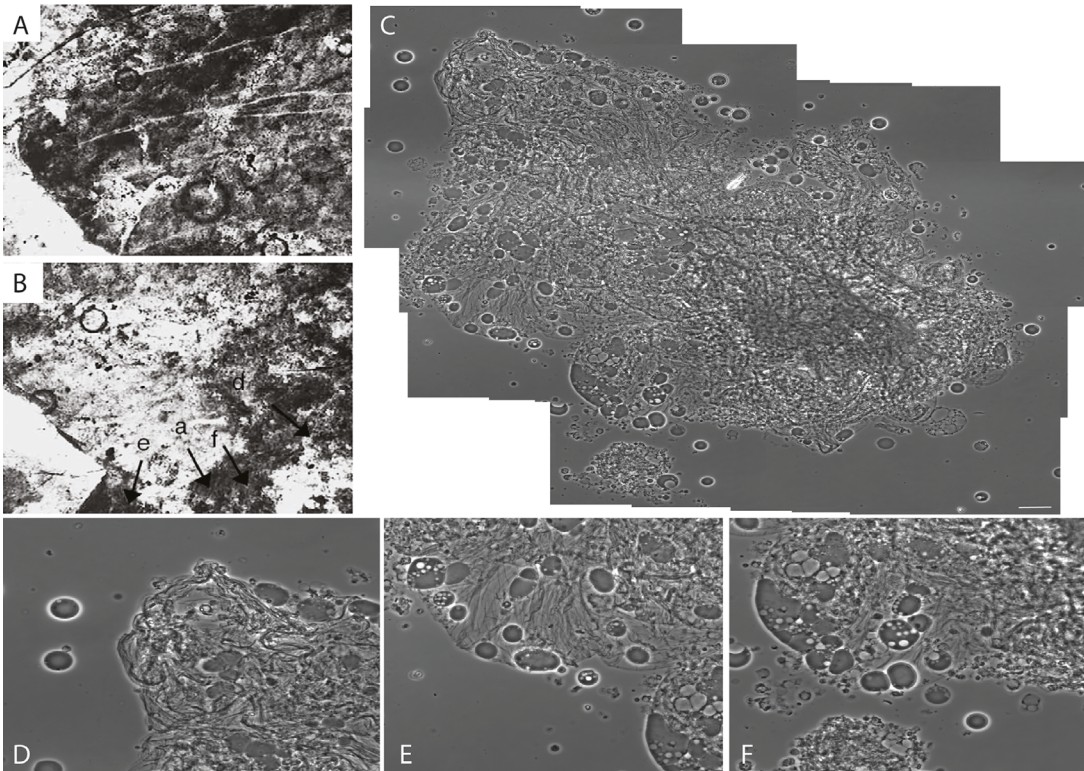

**Appendix 1—figure 49.** Morphological comparison between *EM-P* and the Chinaman Creek microfossils. Images (**A & B**) are organic structures reported from the Chinaman Creek (*Brasier et al., 2005*; *Tice, 2009*). Image (**C**) is a composite image of morphologically analogous *EM-P* structures. Images (**D-F**) are magnified regions of image (**C**). All the images show spherical structures with or devoid of organic carbon enclosed in a film-like structure. Scale bar: 10 μm (**C**).

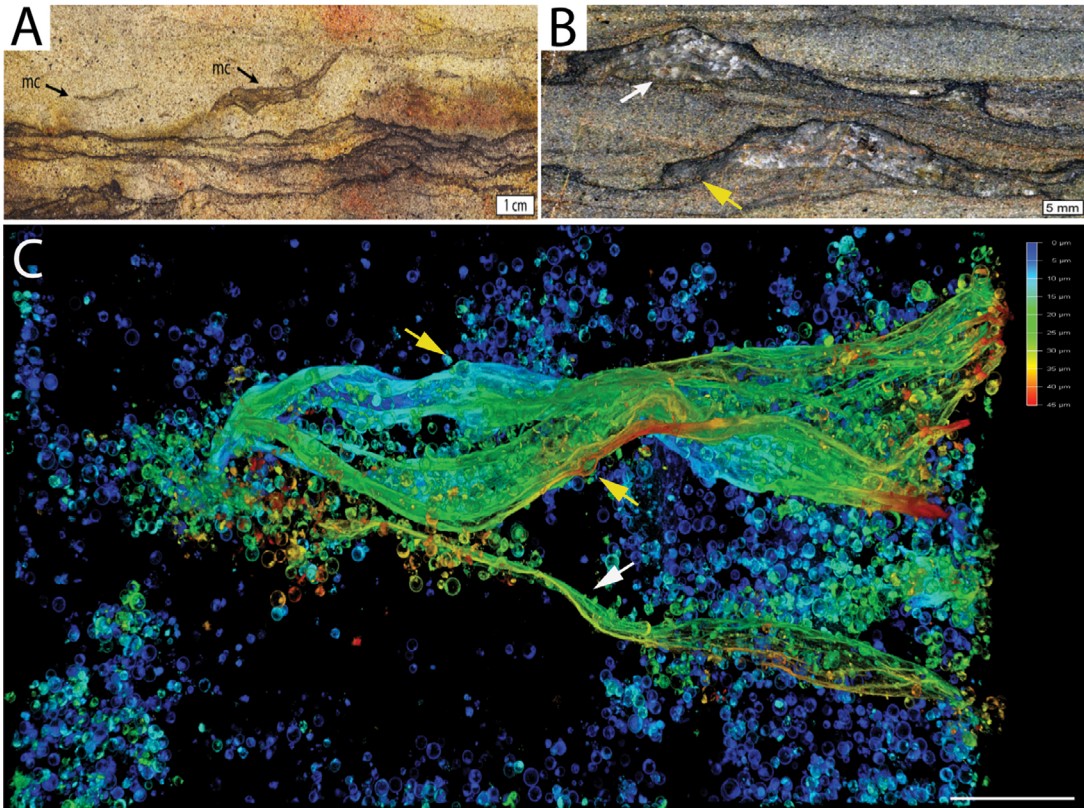

**Appendix 1—figure 50.** Morphological comparison of *EM-P* Membrane debris with laminated structures. Images (**A & B**) show bifurcating laminated structures reported from the Moodies Group (originally published by *Homann et al., 2015*; *Brasier et al., 2005*). Image (**C**) is a 3D-rendered confocal image of image of morphological analogous membrane debris formed by *EM-P* cells. White arrows in B point to hollow lenticular regions within the laminations, similar to the one observed in *EM-P*. Arrows in images (**A & C**) point to the bifurcation of membrane debris. Membranes were stained with Nile red, and imaging was done using a point scanning microscope. Scale bar: 50 µm (**C**).

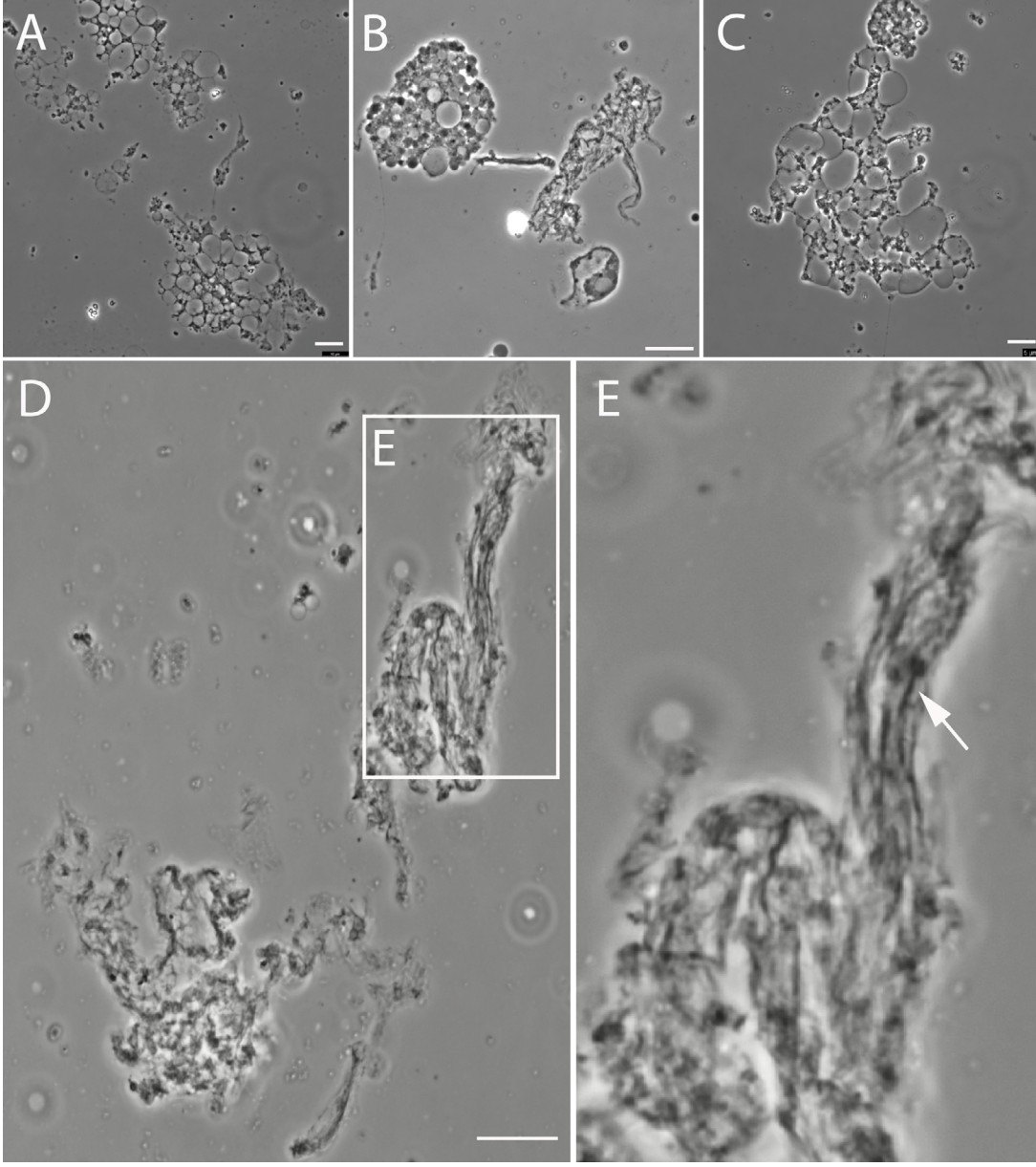

**Appendix 1—figure 51.** Sequential stages involved in the formation of β-laminations. Images (**A-E**) shows the steps involved in the sequential transformation of membrane debris to β-type laminations. Image (**A-C**) shows the aggregation of vacuoles formed from the lysis of large spherical *EM-P* cells. Image (**D**) shows collapsed membrane debris formed by lateral compression or deflation of such aggregations after the release of daughter cells. The arrow in image (**E**) points to the individual membrane layers within the debris. Scale bars:10 μm.

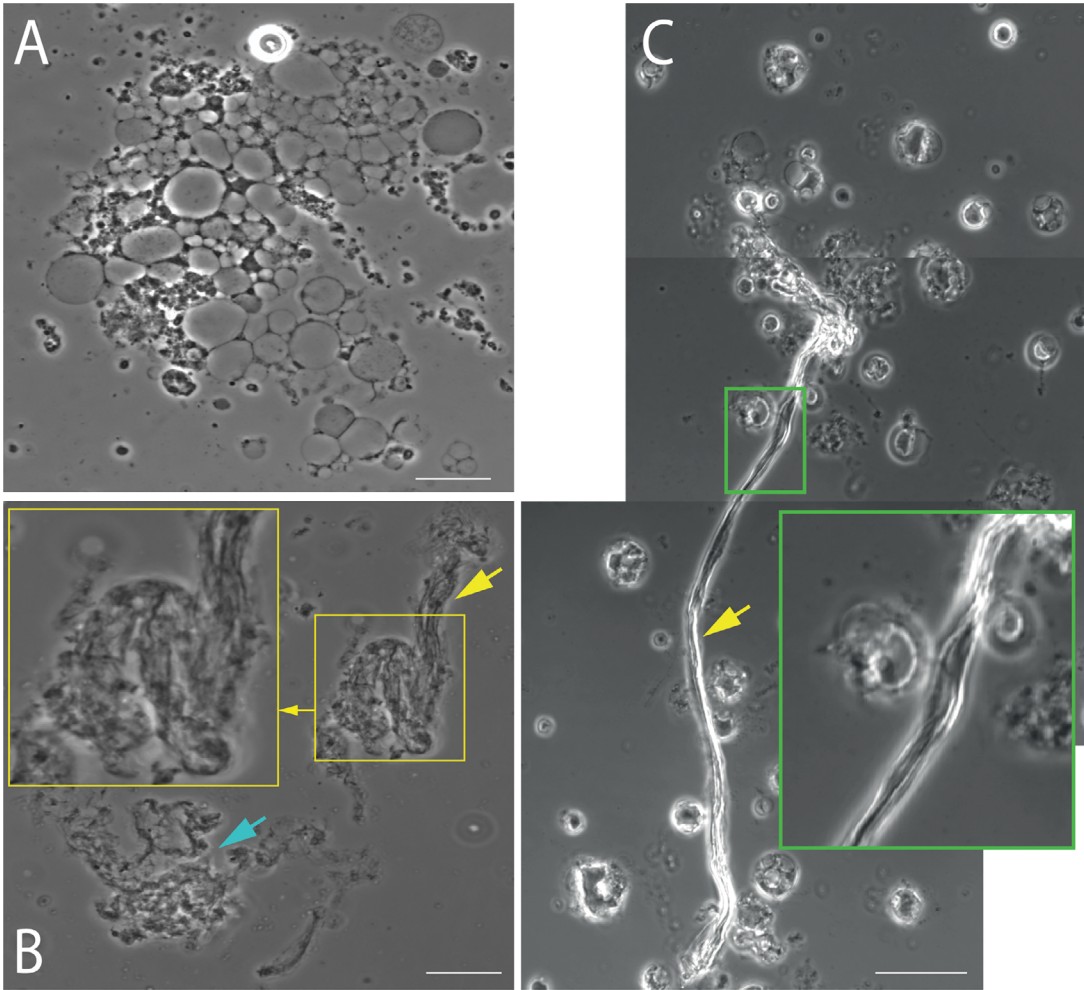

**Appendix 1—figure 52.** Sequential stages involved in the formation of β-laminations. Images (**A-C**) show the steps involved in the sequential transformation of membrane debris to β-type laminations. Image (**A**) shows the aggregation of vacuoles formed from the lysis of large spherical *EM-P* cells. Image (**B**) shows collapsed membrane debris formed by lateral compression or deflation of such aggregations after the release of daughter cells. Cyan and yellow arrows point to membrane debris and compressed membrane debris, respectively. Inset in image (**B**) is the magnified region of compressed membrane debris. Image-c shows structures similar to β-type laminations. Scale bars:10 μm.

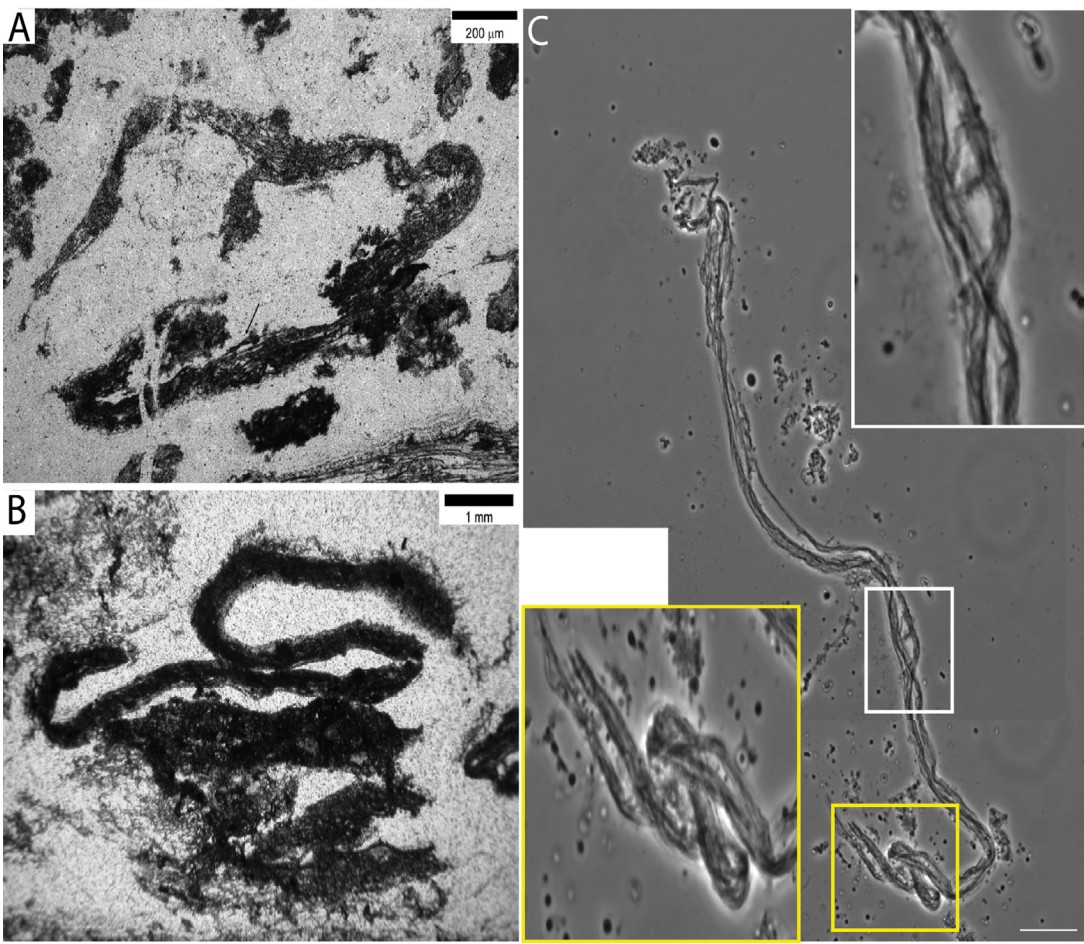

**Appendix 1—figure 53.** Morphological comparison of the BRC β-laminations with *EM-P's* membrane debris. Images (**A & B**) show β-type laminations reported from the BRC (originally published by *Tice and Lowe, 2004*; *Duck et al., 2007*). These laminations are described as rolled-up or bundled filamentous structures. Image (**C**) shows morphologically analogous membrane debris formed by *EM-P*. Insets in image (**C**) are magnified regions of the debris showing bundled-up individual filamentous structures. Scale bars: 10 µm (**C**).

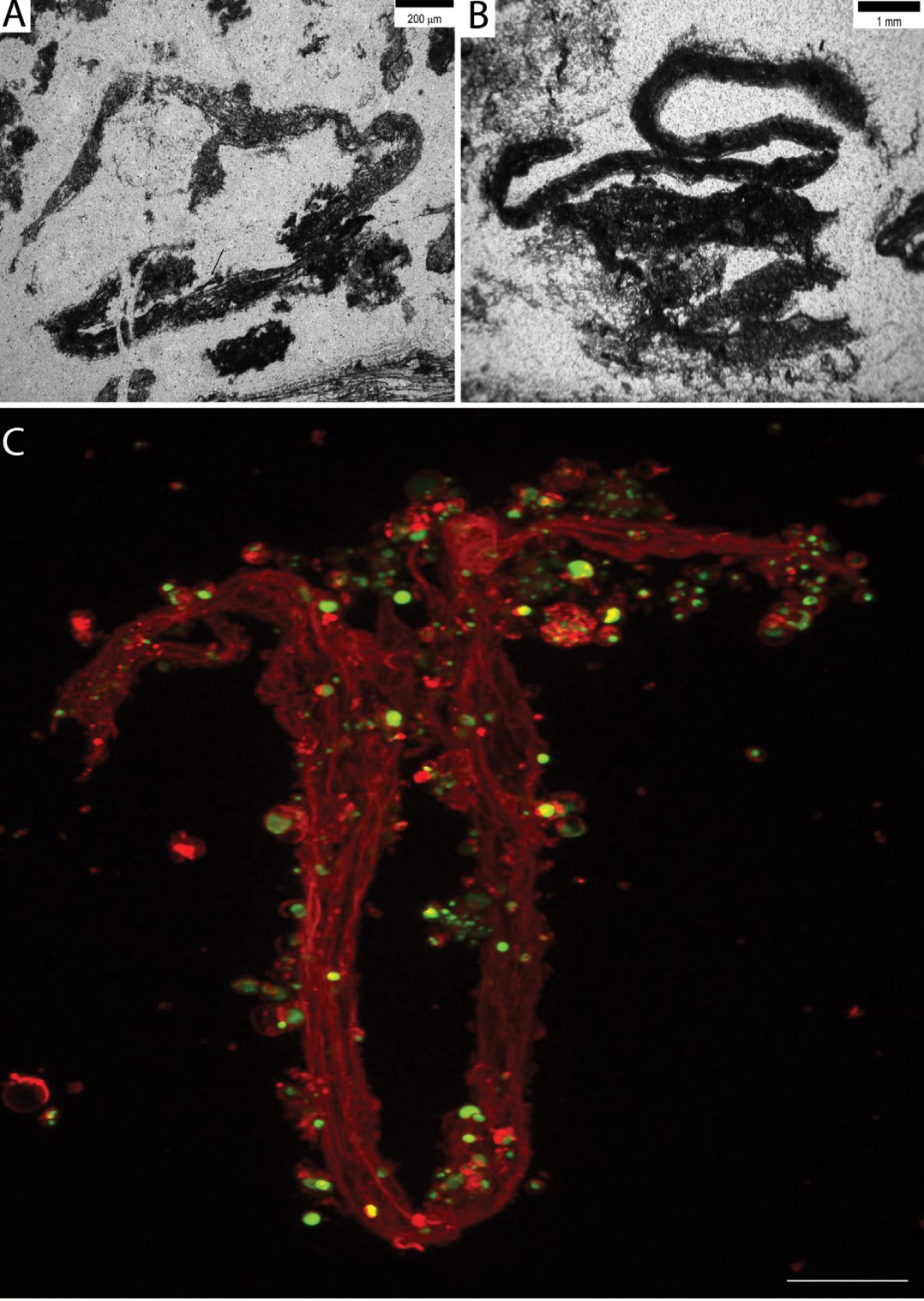

**Appendix 1—figure 54.** Morphological comparison of the BRC β-laminations with *EM-P's* membrane debris.
Images (**A & B**) show β-type laminations reported from the BRC (originally published by *Tice and Lowe, 2004*;
*Duck et al., 2007*). Image (**C**) shows an STED image of morphologically analogous membrane debris formed by
*EM-P*. Spherical daughter cells (evidenced by the presence of DNA) are still attached to the membrane debris,
which can be seen in the image. In the image, the membrane was stained with FM5-95 (red), and DNA was stained
with PicoGreen (green). Scale bar: 10 μm (**C**).

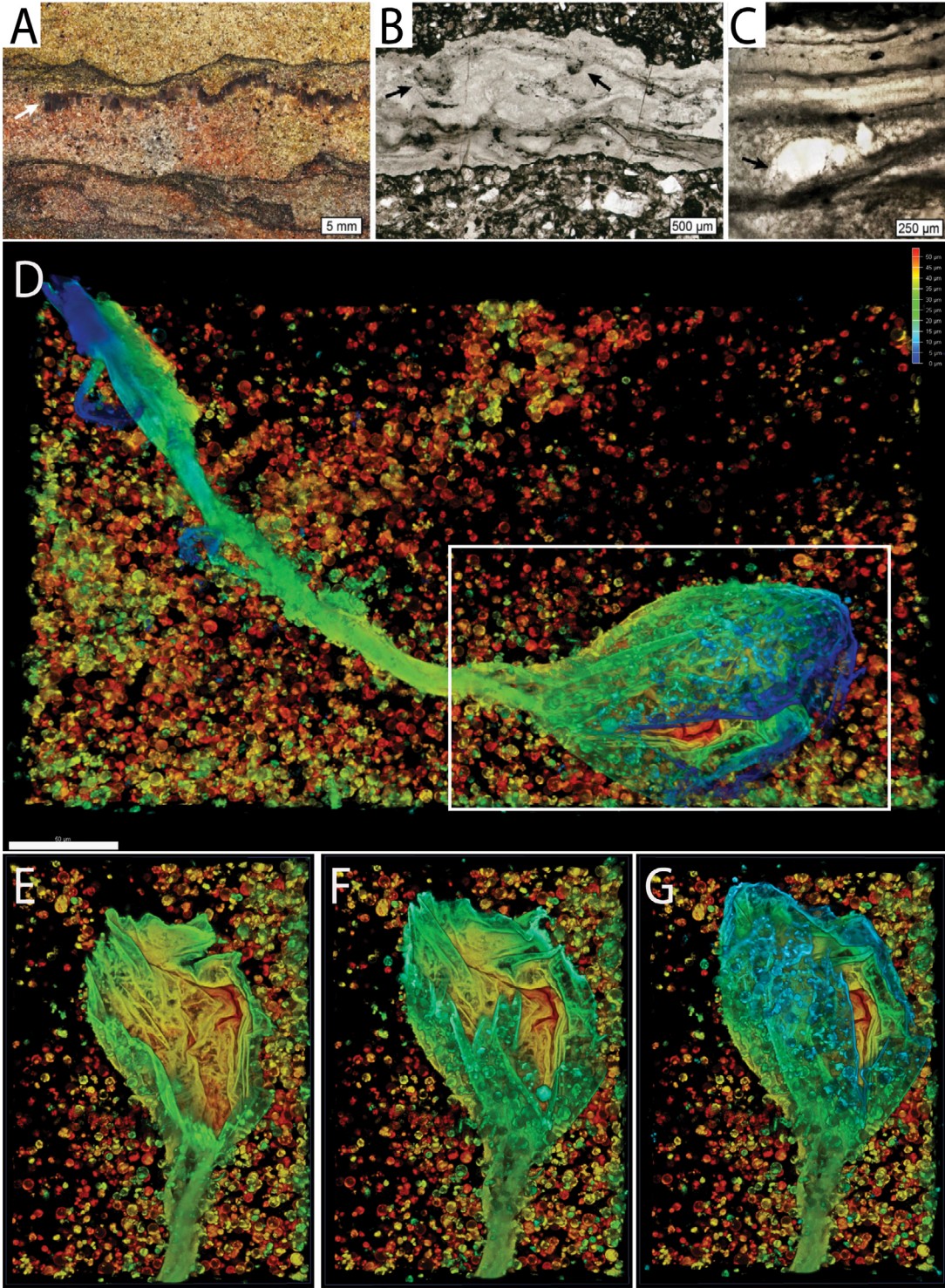

**Appendix 1—figure 55.** Morphological comparison of the Moodies Group laminations with *EM-P's* membrane debris. Image (**A-C**) shows laminations reported from the Moodies Group (originally published by *Homann et al., 2018*; *Kazmierczak et al., 2009*). Image (**D**) is a 3D-rendered confocal image of *EM-P's* membrane debris exhibiting hollow lenticular gaps (*Video 20*). Images (**E-G**) show the hollow lenticular structure at different Z-axis positions. Membranes were stained with Nile red, and imaging was done using an STED microscope. Scale bar: 50 μm.

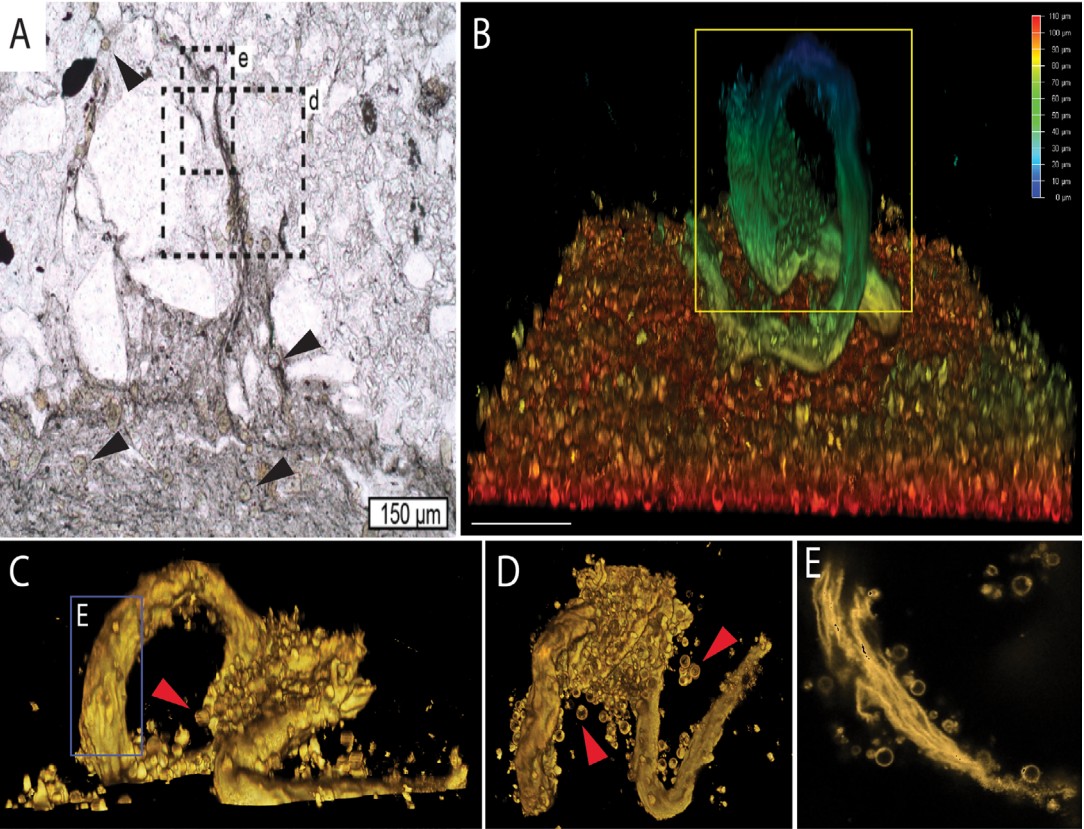

**Appendix 1—figure 56.** Morphological comparison of the Moodies Group laminations with *EM-P's* membrane debris. Image (**A**) show β-type laminations reported from the Moodies Group (originally published by *Homann et al., 2018*; *Kazmierczak et al., 2009*). Laminations from the Moodies Group were shown to form raised filamentous structures (**A**). Arrows in the image point to spherical structures within the laminations. Images (**B-D**) show 3D-rendered confocal images of morphologically analogous *EM-P's* membrane debris. Image (**B**) shows filamentous membrane debris of *EM-P*, that rose above layers of spherical cells. Images (**C & D**) are close-up images of raised filamentous membrane debris of *EM-P* from different viewing angles. Arrows in these images point to spherical cells attached to the membrane debris. Image (**E**) is a non-3D rendered image showing individual membrane layers within the debris, with spherical inclusions attached. Scale bar: 20 μm (**B**).

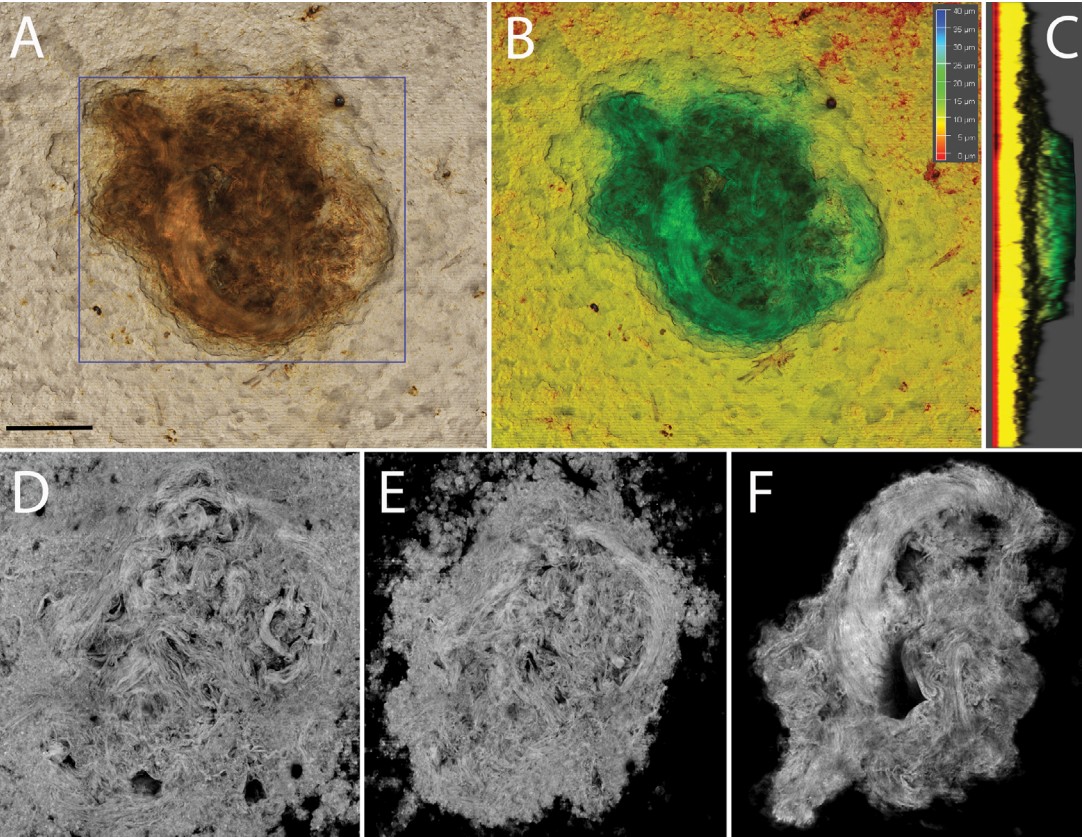

**Appendix 1—figure 57.** Swirl-like structures of *EM-P*'s membrane debris. Images (**A-F**) show a 3D-rendered STED image of membrane debris observed in late-growth stages of *EM-P* (1–2 month-old). This debris typically formed raised mound-like structures. Images (**A & B**) show the top view of such structures, and image C shows the side view. Images (**D-F**) are images of such mounds at different depths (bottom to top). The formation of swirls can be seen in images (**E & F**). In the image, membrane debris is stained with Nile red. A morphological comparison of these structures with Archaean laminations is shown in *Appendix 1—figure 58*. Scale bar: 50 µm (**A**).

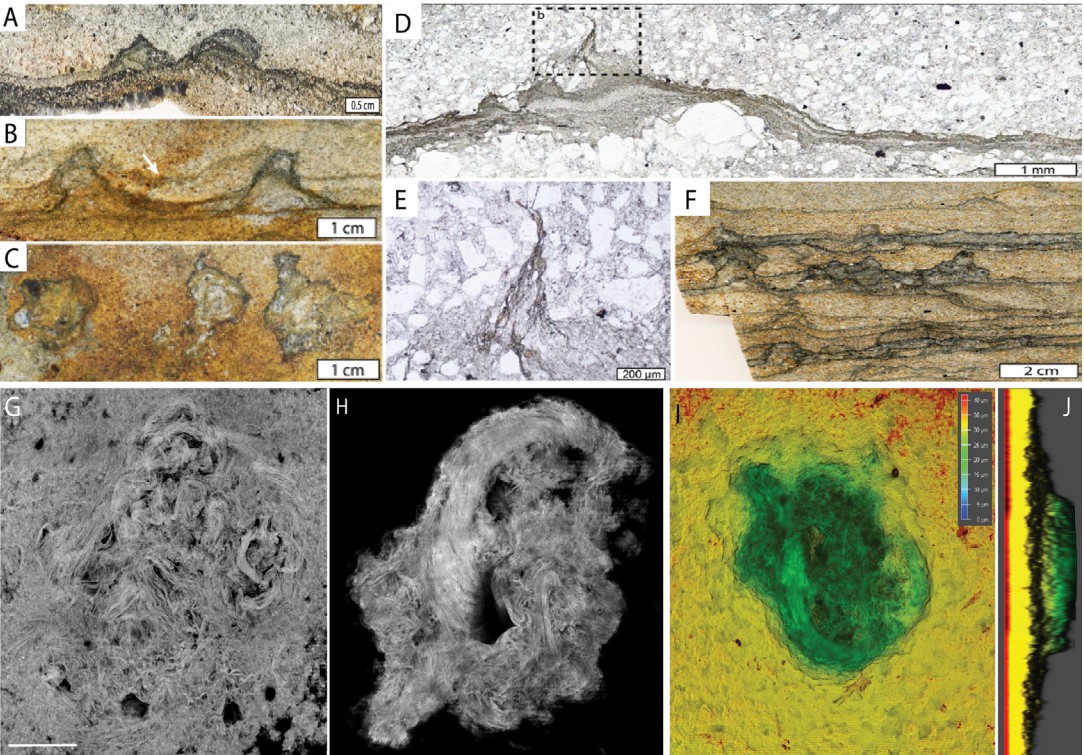

**Appendix 1—figure 58.** Morphological comparison of the Moodies Group laminations with *EM-P's* membrane debris. Images (**A-F**) shows raised mound-like structures reported from the Moodies Group (*Homann et al., 2018*; *Kazmierczak et al., 2009*). Image (**C**) shows the top view of such mounds. These mounds were shown to have filamentous extensions at their peaks. Images (**G-J**) are morphologically analogous structures formed by *EM-P*. In these images, the membrane debris is stained with Nile red. Images (**I & J**) show the top and side views of these raised mounds (depth coloring). Scale bar: 50 µm (**G**).

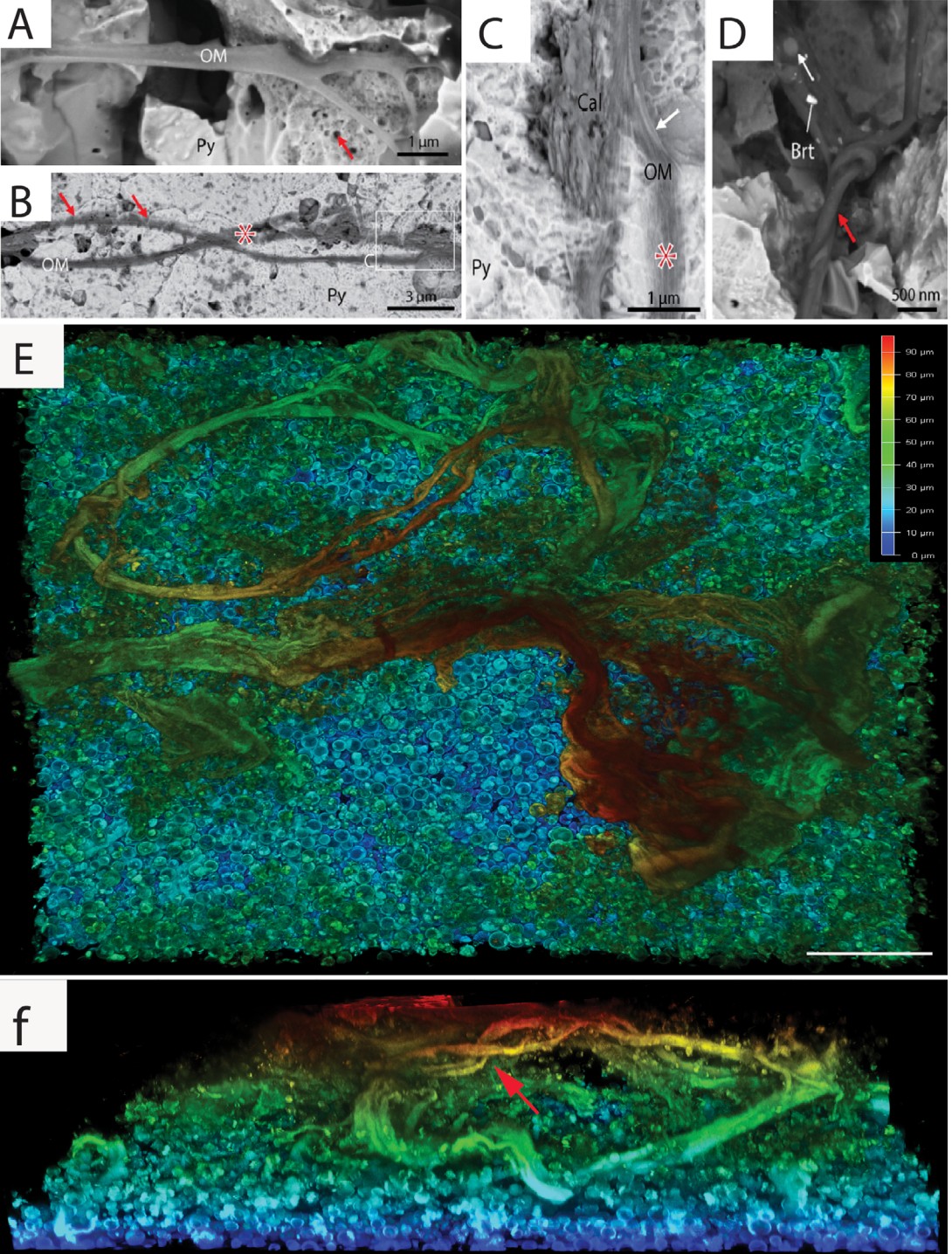

**Appendix 1—figure 59.** Morphological comparison of laminations reported from the BRC with *EM-P's* membrane debris. Images (**A & B**) show β-type laminations reported from the BRC (originally published by *Tice, 2009*; *Tice and Lowe, 2004*). Images (**E & F**) are frontal and lateral view of morphologically analogous structures formed by *EM-P*. Arrows in these images point to helical structures observed in microfossils and *EM-P*. Star signs in images (**B, C, & E**) point to regions showing the filamentous nature of the membrane debris. In the image, the membrane was stained with Nile red. Scale bars: 50 µm (**E**).

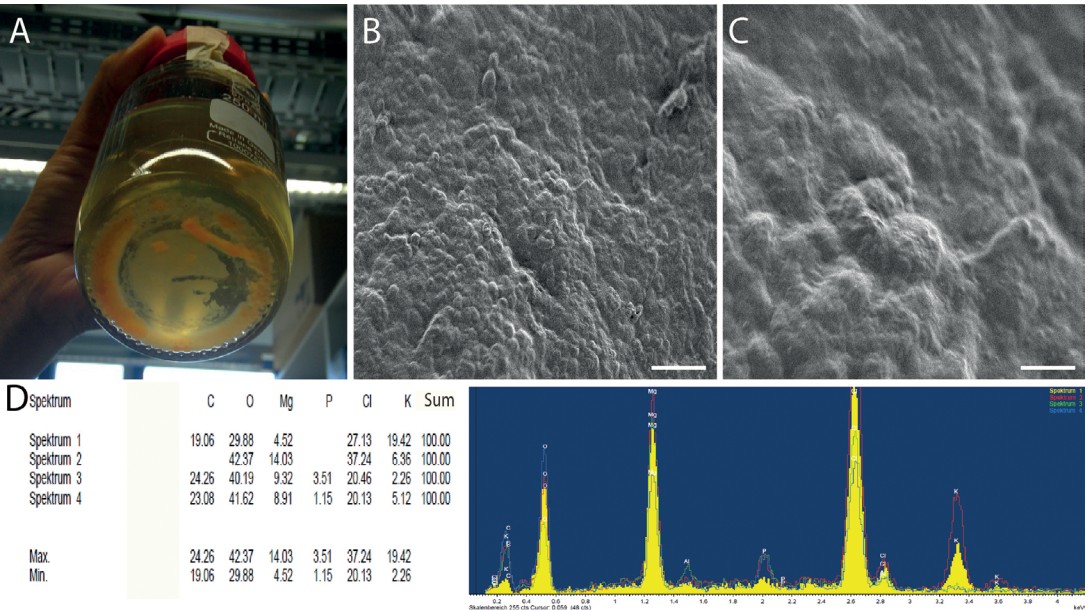

**Appendix 1—figure 60.** Chemical nature of *EM-P*'s crust. Image (**A**) shows a 30-month-old *EM-P* culture at the bottom of the bottle. Over the course of incubation, *EM-P* biofilm transformed into a brittle wafer due to salt encrustation. Images (**B & C**) show SEM images of *EM-P*. Scale bars in these images are 2 μm. The elemental composition of the solidified mat is shown in (**D**) (determined by SEM-EDX).

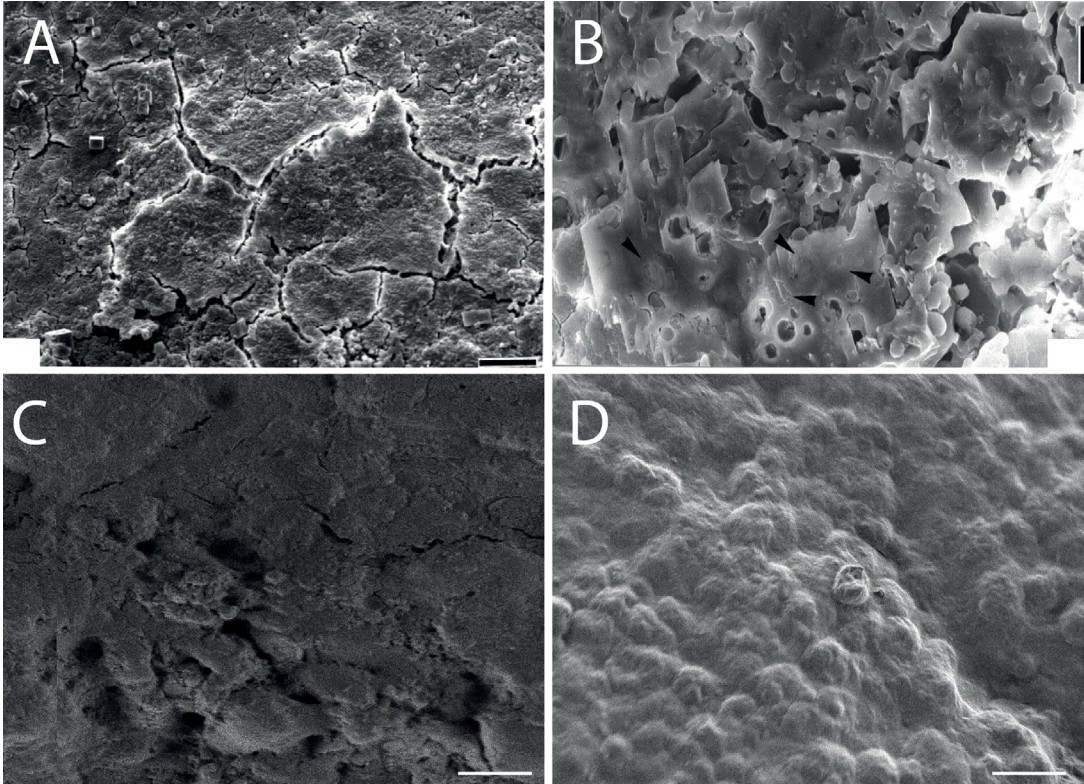

**Appendix 1—figure 61.** Morphological comparison of the Kromberg microfossils with *EM-P*. Images (**A & B**) show mineral-encrusted microbial mats (**A**) and spherical microfossils (**B**) within these encrusted mats reported from the Kromberg Formation (originally published by *Westall et al., 2001*; *Homann et al., 2015*). Scale bars in images (**A & B**) are 20 μm. (**C & D**) show SEM images of morphologically analogous salt-encrusted *EM-P* cells after 30 months of incubation. The elemental composition of such microbial mats and individual cell morphologies are shown in *Appendix 1—figure 60*. Scale bars in images (**C & D**) are 2 μm.

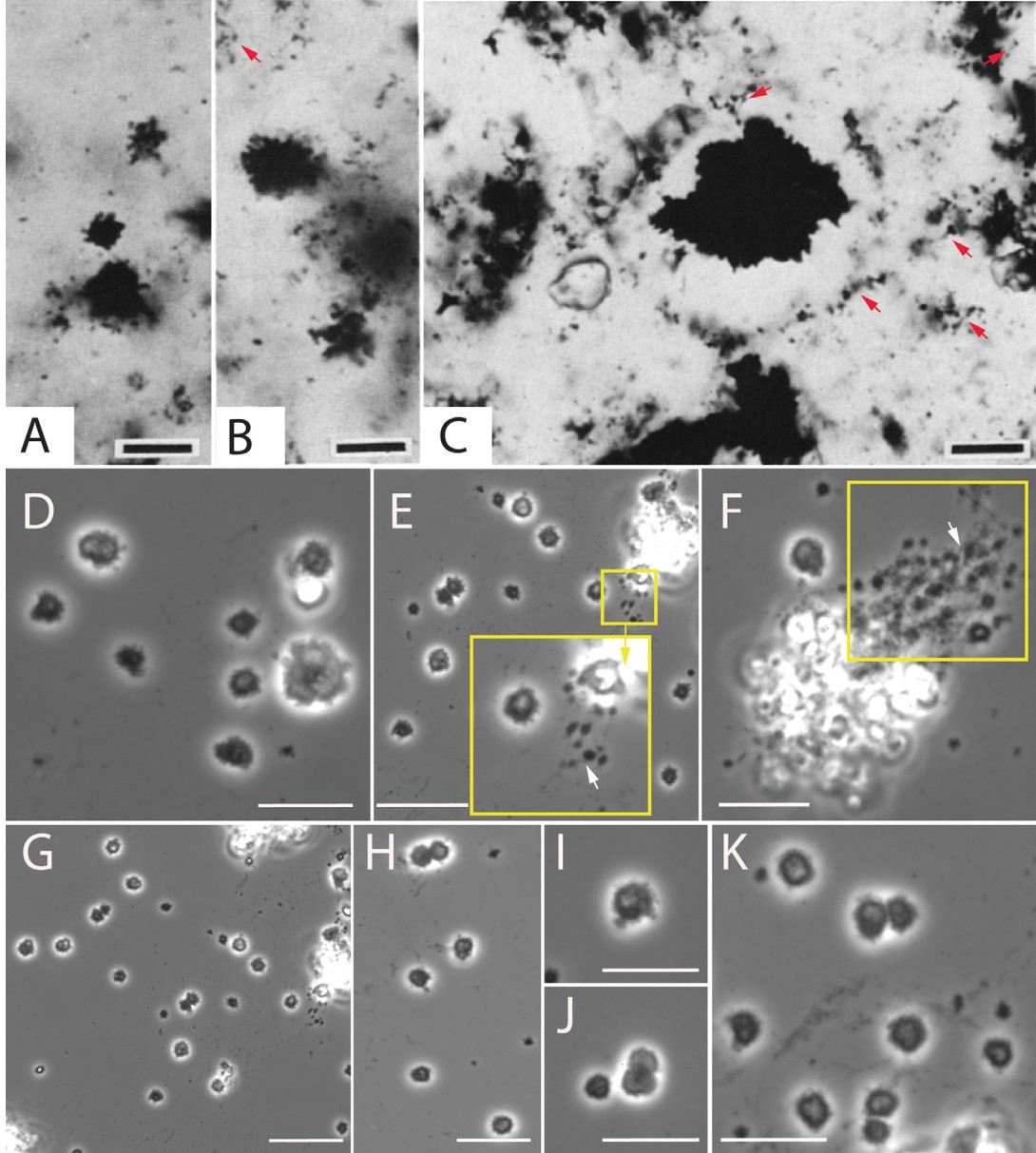

**Appendix 1—figure 62.** Morphological comparison of the North Pole microfossils with *EM-P*. Images (**A-C**) show stellate-shaped structures with filamentous projections reported from the North Pole locality (originally published by *Buick, 1990*; *Kaźmierczak and Kremer, 2019*). Scale bars: 100 μm. (**D-K**) are phase-contrast images of morphologically analogous *EM-P* cells. The stellar-shaped structure could have been the biologically induced mineral formed on the surface of the cells, as shown in *Appendix 1—figure 60*. The filamentous structures were the daughter cells growing out of the crust when encrusted cells were transferred into fresh media. Arrows in images (**C, E, & F**) point to a filamentous string of daughter cells extending out of cell clumps (*Video 22*). Images (**D-K**) also show salt-encrusted individuals or cells in dyads undergoing binary fission. Scale bars: 10 μm.

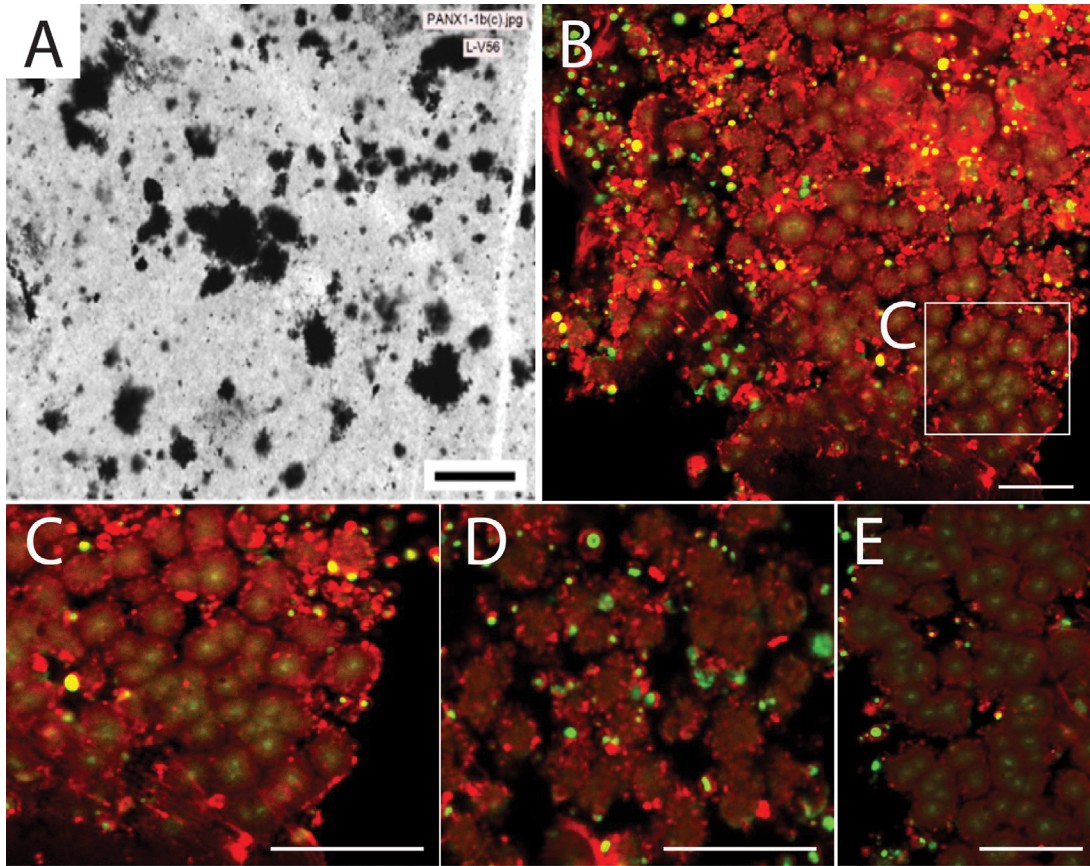

**Appendix 1—figure 63.** Morphological comparison of the Strelley Pool Formation (SPF) microfossils with *EM-P*. Image A shows stellate microfossils with filamentous projections reported from the SPF (originally published by *Sugitani et al., 2013*; *Schopf and Barghoorn, 1967*). Scale bar: 200 µm. Images B-E are spinning-disk confocal images of morphologically analogous *EM-P* cells. Cells in these images are stained with membrane stain, FM5-95 (red), and PicoGreen (DNA, green). Like microfossils, *EM-P* cells revived from the above-described crust (*Appendix 1—figure 62*) formed filamentous daughter cells from below the stellate structures. Scale bars: 10 µm.

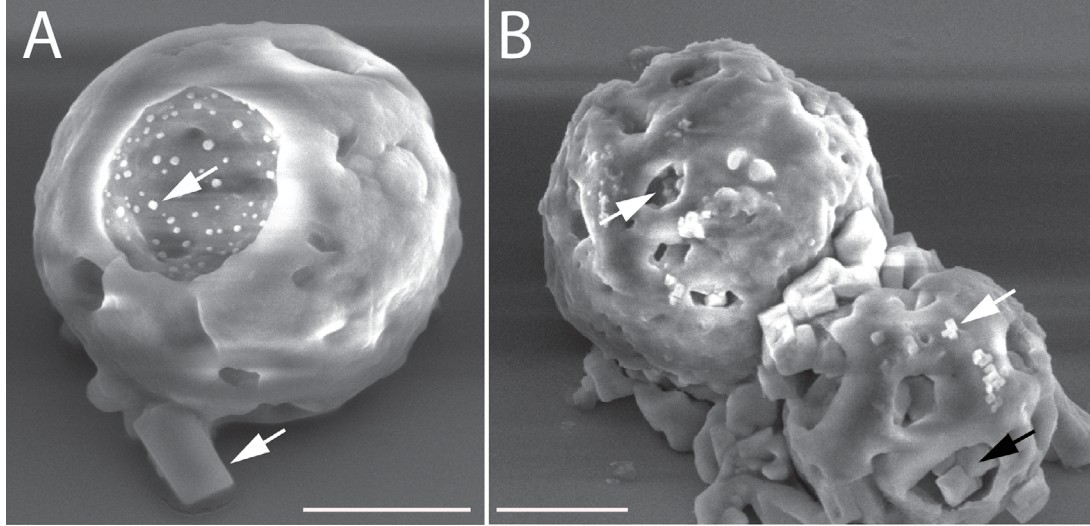

**Appendix 1—figure 64.** Association of salt with *EM-P*. Images (**A & B**) are SEM images of *EM-P*. Image (**A**) shows *EM-P* cells forming membrane invaginations that resemble an endocytosis vesicle. These images *EM-P* with salt crystals within the membrane vesicles (closed arrows). Scale bars: 2 µm.

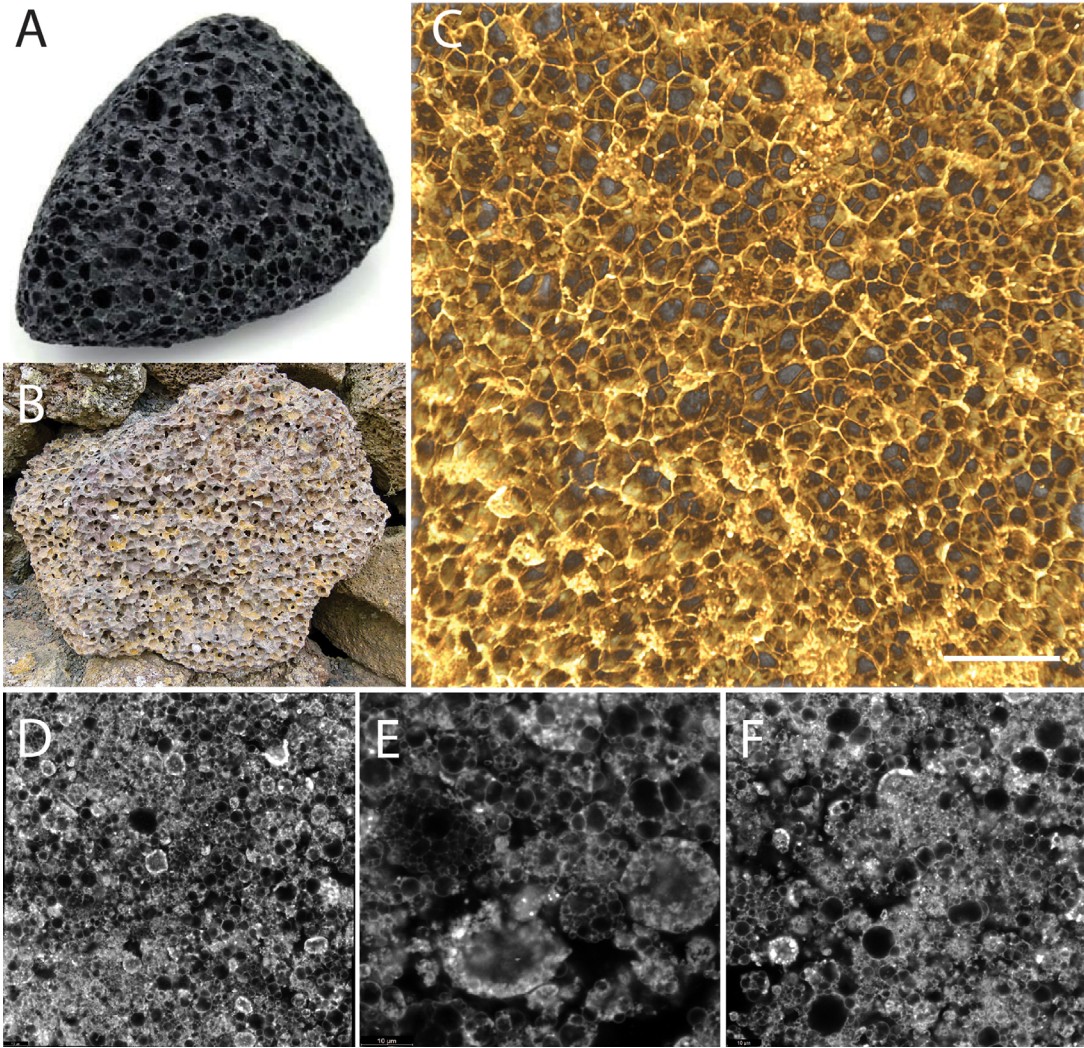

**Appendix 1—figure 65.** Morphological comparison of volcanic pumice with honeycomb-like structures observed in *EM-P* incubations. Images (**A & B**) are the images of volcanic pumice (World Wide Web). Images (**C-F**) are the images of *EM-P* biofilm. Scale bars: 20 µm (**C**) and 10 µm (**D–F**).

