## [Editor Report · eLife Assessment]

This provocative manuscript presents **important** comparisons of the morphologies of Archaean bacterial microfossils to those of microbes transformed under environmental conditions that mimic those present on Earth during the same Eon. The evidence in support of the conclusions is **solid**. The authors' environmental condition selection for their experiment is justified.

---

## [Referee Report · Joint Public Review]

Summary:

Microfossils from the Paleoarchean Eon represent the oldest evidence of life, but their nature has been strongly debated among scientists. To resolve this, the authors reconstructed the lifecycles of Archaean organisms by transforming a Gram-positive bacterium into a primitive lipid vesicle-like state and simulating early Earth conditions. They successfully replicated all morphologies and life cycles of Archaean microfossils and studied cell degradation processes over several years, finding that encrustation with minerals like salt preserved these cells as fossilized organic carbon. Their findings suggest that microfossils from 3.8 to 2.5 billion years ago were likely liposome-like protocells with energy conservation pathways but without regulated morphology.

Strengths:

The authors have crafted a compelling narrative about the morphological similarities between microfossils from various sites and proliferating wall-deficient bacterial cells, providing detailed comparisons that have never been demonstrated in this detail before. The extensive number of supporting figures is impressive, highlighting numerous similarities. While conclusively proving that these microfossils are proliferating protocells morphologically akin to those studied here is challenging, we applaud this effort as the first detailed comparison between microfossils and morphologically primitive cells.

Summary of reviewer comments on this revision:

Each of the original reviewers evaluated the revised manuscript and were complimentary about how the authors addressed their original concerns. One reviewer added: "It is a thought-provoking manuscript that will be well received." We encourage readers of this version of the paper to consider the original reviewer comments and the authors' responses: https://elifesciences.org/reviewed-preprints/98637/reviews

---

## [Author Response]

The following is the authors’ response to the original reviews.

**eLife Assessment**
This provocative manuscript from presents valuable comparisons of the morphologies of Archaean bacterial microfossils to those of microbes transformed under environmental conditions that mimic those present on Earth during the same Eon, although the evidence in support of the conclusions is currently incomplete. The reasons include that taphonomy is not presently considered, and a greater diversity of experimental environmental conditions is not evaluated -- which is important because we ultimately do not know much about Earth's early environments. The authors may want to reframe their conclusions to reflect this work as a first step towards an interpretation of some microfossils as 'proto-cells,' and less so as providing strong support for this hypothesis.

Regarding the taphonomic alterations: The editor and reviewers are correct in pointing out this issue. Taphonomic alteration of the microfossils attains special significance in the case of microorganisms, as they lack rigid structures and are prone to morphological alterations during or after their fossilization. We are acutely aware of this issue and have conducted long-term experiments (lasting two years) to observe how cells die, decay, and get preserved. A large section of the manuscript (pages 11 to 20) and a substantial portion of the supplementary information is dedicated to understanding the taphonomic alterations. To the best of our knowledge, these are among the longest experiments done to understand the taphonomic alterations of the cells within laboratory conditions.

Recent reports by Orange et al. (1,2) showed that under favorable environmental conditions, cells could be fossilized rather rapidly with little morphological modifications. We observed a similar phenomenon in this work. Cells in our study underwent rapid encrustation with cations from the growth media. We have analyzed the morphological changes over a period of 18 months. After 18 months, the softer biofilms got encrusted entirely in salt and turned solid (Fig.). Despite this transformation, morphologically intact cells could still be observed within these structures. This suggests that the cells inhabiting Archaean coastal marine environments could undergo rather rapid encrustation, and their morphological features could be preserved in the geological record with little taphonomic alteration.

Regarding the environmental conditions: We are in total agreement with the reviewers that much is unknown about Archaean geology and its environmental conditions. Like the present-day Earth, Archaean Earth certainly had regions that greatly differed in their environmental conditions—volcanic freshwater ponds, brines, mildly halophilic coastal marine environments, and geothermal and hydrothermal vents, to name a few. Our experimental design focuses on one environment we have a relatively good understanding of rather than the rest of the planet, of which we know little. Below, we list our reasons for restricting to coastal marine environments and studying cells under mildly halophilic experimental conditions.

(1) Very little continental crust from Haden and early Archaean Eon exists on the presentday Earth. Much of our geochemical understanding of this time period was a result of studying the Pilbara Iron Formations and the Barberton Greenstone Belt. Geological investigations suggest that these sites were coastal marine environments. The salinity of coastal marine environments is higher than that of open oceans due to the greater water evaporation within these environments. Moreover, brines were discovered within pillow basalts within the Barberton greenstone belt, suggesting that the salinity within these sites is higher or similar to marine environments.

(2) We are not certain about the environmental conditions that could have supported the origin of life. However, all currently known Archaean microfossils were reported from coastal marine environments (3.8-2.4Ga). This suggests that proto-life likely flourished in mildly halophilic environments, similar to the experimental conditions employed in our study.

(3) The chemical analysis of Archaean microfossils also suggests that they lived in saltrich environments, as most, if not all, microfossils are closely associated, often encrusted in a thin layer of salt.

However, we concur with the reviewers that our interpretations should be reassessed if Archaean microfossils that greatly differ from the currently known microfossils are to be discovered or if new microfossils are to be reported from environments other than coastal marine sites.

**Public Reviews:**

**Reviewer #1 (Public Review):**
Summary:Microfossils from the Paleoarchean Eon represent the oldest evidence of life, but their nature has been strongly debated among scientists. To resolve this, the authors reconstructed the lifecycles of Archaean organisms by transforming a Gram-positive bacterium into a primitive lipid vesicle-like state and simulating early Earth conditions. They successfully replicated all morphologies and life cycles of Archaean microfossils and studied cell degradation processes over several years, finding that encrustation with minerals like salt preserved these cells as fossilized organic carbon. Their findings suggest that microfossils from 3.8 to 2.5 billion years ago were likely liposome-like protocells with energy conservation pathways but without regulated morphology.Strengths:The authors have crafted a compelling narrative about the morphological similarities between microfossils from various sites and proliferating wall-deficient bacterial cells, providing detailed comparisons that have never been demonstrated in this detail before. The extensive number of supporting figures is impressive, highlighting numerous similarities. While conclusively proving that these microfossils are proliferating protocells morphologically akin to those studied here is challenging, we applaud this effort as the first detailed comparison between microfossils and morphologically primitive cells.Weaknesses:Although the species used in this study closely resembles the fossils morphologically, it would be beneficial to provide a clearer explanation for its selection. The literature indicates that many bacteria, if not all, can be rendered cell wall-deficient, making the rationale for choosing this specific species somewhat unclear. While this manuscript includes clear morphological comparisons, we believe the authors do not adequately address the limitations of using modern bacterial species in their study. All contemporary bacteria have undergone extensive evolutionary changes, developing complex and intertwined genetic pathways unlike those of early life forms. Consequently, comparing existing bacteria with fossilized life forms is largely hypothetical, a point that should be more thoroughly emphasized in the discussion.Another weak aspect of the study is the absence of any quantitative data. While we understand that obtaining such data for microfossils may be challenging, it would be helpful to present the frequencies of different proliferative events observed in the bacterium used. Additionally, reflecting on the chemical factors in early life that might cause these distinct proliferation modes would provide valuable context.

Regarding our choice of using modern organisms or this particular bacterial species:

Based on current scientific knowledge, it is logical to infer that cellular life originated as protocells; nevertheless, there has been no direct geological evidence for the existence of such cells on early Earth. Hence, protocells remain an entirely theoretical concept. Moreover, protocells are considered to have been far more primitive than present-day cells. Surprisingly, this lack of sophistication was the biggest challenge in understanding protocells. Designing experiments in which cells are primitive (but not as primitive as non-living lipid vesicles) and still retain a functional resemblance to a living cell does pose some practical challenges. Laboratory experiments with substitute (proxy) protocells almost always come with some limitations. Although not a perfect proxy, we believe protocells and protoplasts share certain characteristics. Having said that, we would like to reemphasize that protoplasts are not protocells. Our reasons for using protoplasts as model organisms and working with this bacterial species (Exiguobacterium Strain-Molly) are based on several scientific and practical criteria listed below.

(1) Irrespective of cell physiology and intracellular complexity, we believe that protoplasts and protocells share certain similarities in the biophysical properties of their cytoplasm. We explained our reasoning in the manuscript introduction and in our previous manuscripts (Kanaparthi et al., 2024 & Kanaparthi et al., 2023). In short, to be classified as a cell, even a protocell should possess minimal biosynthetic pathways, a physiological mechanism of harvesting free energy from the surrounding (energy-yielding pathways), and a means of replicating its genetic material and transferring it to the daughter cells. These minimal physiological processes could incorporate considerable cytoplasmic complexity. Hence, the biophysical properties of the protocell cytoplasm could have resembled those of the cytoplasm of protoplasts, irrespective of the genomic complexity.

(2) Irrespective of their physiology, protoplasts exhibit several key similarities to protocells, such as their inherent inability to regulate their morphology or reproduction. This similarity was pointed out in previous studies (3). Despite possessing all the necessary genetic information, protoplasts undergo reproduction through simple physiochemical processes independent of canonical molecular biological processes. This method of reproduction is considered to have been erratic and rather primitive, akin to the theoretical propositions on protocells. Although protoplasts are fully evolved cells with considerable physiological complexity, the above-mentioned biophysical similarities suggest that the protoplast life cycle could morphologically resemble that of protocells (in no other aspect except for their morphology and reproduction).

(3) Physiologically or genomically different species of Gram-positive protoplasts are shown to exhibit similar morphologies. This suggests that when Gram-positive bacteria lose their cell wall and turn into a protoplast, they reproduce in a similar manner independent of physiological or genome-based differences. As morphology and only morphology is key to our study, at least from the scope of this study, intracellular complexity is not a key consideration.

(4) This specific strain was isolated from submerged freshwater springs in the Dead Sea. This isolate and members of this bacterial genus are known to have been well acclimatized to growing in a wide range of salt concentrations and in different salt species. This is important for our study (this and previous manuscript), in which cells must be grown not only at high salt concentrations (1-15%) but in different salts like NaCl, MgCl_2_, and KCl.

(5) Our initial interest in this isolate was due to its ability to reduce iron at high salt concentrations. Given that most spherical microfossils are found in Archaean-banded iron formations covered in pyrite, this suggests that these microfossils could have been reducing oxidized iron species like Fe(III). Nevertheless, over the course of our study, we realized the complexities of live cell staining and imaging under anoxic conditions. Given that the scope of the manuscript is restricted only to comparing the morphologies, not the physiology, we abandoned the idea of growing cells under anoxic conditions.

Based on these observations, cell physiology may not be a key consideration, at least within the scope of studying microfossil morphology. However, we want to emphasize again that “We do not claim present-day protoplasts are protocells.”

Regarding the absence of quantitative data:

We are unsure what the reviewer meant by the absence of quantitative data. Is it from the cell size/reproductive pathways perspective or from a microfossil/ecological perspective? At the risk of being portrayed in a bad light, we admit that we did not present quantitative data from either of these perspectives. In our defense, this was not due to our lack of effort but due to the practical limitations imposed by our model organism.

If the reviewer means the quantitative data regarding cell sizes and morphology: In our previous work, we studied the relationship between protoplast morphology, growth rate, and environmental conditions. In that study, we proposed that the growth rate is one factor that regulates protoplast morphology. Nevertheless, we did not observe uniformity in the sizes of the cells. This lack of uniformity was not just between the replicates but even among the cells grown within the same culture flask or the cells within the same microscopic field. Moreover, cells are often observed to be reproducing either by forming internal or external or by both these processes at the same time. The size and morphological differences among cells within a growth stage could be explained by the physiological and growth rate heterogenicity among cells.

Bacterial growth curves and their partition into different stages (lag, log & stationary), in general, represent the growth dynamics of an entire bacterial population. Nevertheless, averaging the data obscures the behavior of individual cells (4,5). It is known that genetically identical cells within a single bacterial population could exhibit considerable cell-to-cell variation in gene expression (6,7) and growth rates (8). The reason for such stochastic behavior among monoclonal cells has not been well understood. In the case of normal cells, morphological manifestation of these variations is restricted by a rigid cell wall. Given the absence of a cell wall in protoplasts, we assume such cell-to-cell variations in growth rate is manifested in cell morphology. This makes it challenging to quantitatively determine variations in cell sizes or the size increase in a statically robust manner, even in monoclonal cells.

Although this lack of uniformity in cell sizes should not be perceived as a limitation, this behavior is consistently observed among microfossils. Spherical microfossils of similar morphology but different sizes were reported from different microfossil sites (9,10). In this regard, both protoplasts and microfossils are very similar.

If the reviewer means the quantitative data from an ecological perspective:

Based on the elemental composition and the isotopic signatures of the organic carbon, we can deduce if these structures are of biological origin or not. However, any further interpretation of this data to annotate these microfossils to a particular physiology group is fraught with errors. Hence, we refrain from making any inferences about the physiology and ecological function of these microfossils. This lack of clarity on the physiology of microfossils reduces the chance of quantitative studies on their ecological functions. Moreover, we would like to re-emphasize that the scope of this work is restricted to morphological comparison and is not targeted at understanding the ecological function of these microfossils. This narrow objective also limits the nature of the quantitative data we could present.

Moreover, developing a quantitative understanding of some phenomena could be technically challenging. Many theories on the origin of life, like chemical evolution, started with the qualitative observation that lightning could mediate the synthesis of biologically relevant organic carbon. Our quantitative understanding of this process is still being explored and debated even to this day.

**Reviewer #2 (Public Review):**
Summary:In summary, the manuscript describes life-cycle-related morphologies of primitive vesiclelike states (Em-P) produced in the laboratory from the Gram-positive bacterium Exiguobacterium Strain-Molly under assumed Archean environmental conditions. Em-P morphologies (life cycles) are controlled by the "native environment". In order to mimic Archean environmental conditions, soy broth supplemented with Dead Sea salt was used to cultivate Em-Ps. The manuscript compares Archean microfossils and biofilms from selected photos with those laboratory morphologies. The photos derive from publications on various stratigraphic sections of Paleo- to Neoarchean ages. Based on the similarity of morphologies of microfossils and Em-Ps, the manuscript concludes that all Archean microfossils are in fact not prokaryotes, but merely "sacks of cytoplasm".Strengths:The approach of the authors to recognize the possibility that "real" cells were not around in the Archean time is appealing. The manuscript reflects the very hard work by the authors composing the Em-Ps used for comparison and selecting the appropriate photo material of fossils.Weaknesses:While the basic idea is very interesting, the manuscript includes flaws and falls short in presenting supportive data. The manuscript makes too simplistic assumptions on the "Archean paleoenvironment". First, like in our modern world, the environmental conditions during the Archean time were not globally the same. Second, we do not know much about the Archean paleoenvironment due to the immense lack of rock records. More so, the Archean stratigraphic sections from where the fossil material derived record different paleoenvironments: shelf to tidal flat and lacustrine settings, so differences must have been significant. Finally, the Archean spanned 2.500 billion years and it is unlikely that environmental conditions remained the same. Diurnal or seasonal variations are not considered. Sediment types are not considered. Due to these reasons, the laboratory model of an Archean paleoenvironment and the life therein is too simplistic. Another aspect is that eucaryote cells are described from Archean rocks, so it seems unlikely that prokaryotes were not around at the same time. Considering other fossil evidence preserved in Archean rocks except for microfossils, the many early Archean microbialites that show baffling and trapping cannot be explained without the presence of "real cells". With respect to lithology: chert is a rock predominantly composed of silica, not salt. The formation of Em-Ps in the "salty" laboratory set-up seems therefore not a good fit to evaluate chert fossils. Formation of structures in sediment is one step. The second step is their preservation. However, the second aspect of taphonomy is largely excluded in the manuscript, and the role of fossilization (lithification) of Em-Ps is not discussed. This is important because Archean rock successions are known for their tectonic and hydrothermal overprint, as well as recrystallization over time. Some of the comparisons of laboratory morphologies with fossil microfossils and biofilms are incorrect because scales differ by magnitudes. In general, one has to recognize that prokaryote cell morphologies do not offer many variations. It is possible to arrive at the morphologies described in various ways including abiotic ones.

Regarding the simplistic presumptions on the Archaean Eon environmental conditions, we provided a detailed explanation of this issue in our response to the eLife evaluation. In short, we agree with the reviewer that little is known about the Archaean Eon environmental conditions at a planetary scale. Hence, we restricted our study to one particular environment of which we had a comparatively good understanding. The Archaean Eon spanned 2.5 billion years. However, most of the microfossil sites we discussed in the manuscript are older than 3 billion years, with one exception (2.4 billion years old Turee Creek microfossils). We presume that conditions within this niche (coastal marine) environment could not have changed greatly until 2Ga, after which there have been major changes in the ocean salt composition and salinities.

In the manuscript, we discussed extensively the reasons for restricting our study to these particular environmental conditions. Further explanations of these choices are presented in our response to the eLife evaluation (also see our previous manuscript). In short, the fact that all known microfossils are restricted to coastal marine environments justifies the experimental conditions employed in our study. Nevertheless, we agree with the reviewer that all lab-based studies involve some extent of simplification. This gap/mismatch is even wider when it comes to studies involving origin or early life on Earth.

We are not arguing that prokaryotes are not around at this time. The key message of the manuscript is that they are present, but they have not developed intracellular mechanisms to regulate their morphology and remained primitive in this aspect.

The sizes of the microfossils and cells from our study were similar in most cases. However, we agree with the reviewer that they deviated considerably in some cases, for example, S70, S73, and S83. These size variations are limited to sedimentary structures like laminations rather than cells. These differences should be expected as we try to replicate the real-life morphologies of biofilms that could have extended over large swats of natural environments in a 2ml volume chamber slide. More specifically, in Fig. S70, there is a considerable size mismatch. But, in Fig. S73, the sizes were comparable between A & C (of course, the size of our reproduction did not match B). In the case of Fig. S83, we do not see a huge size mismatch.

**Reviewer #1 (Recommendations For The Authors):**
We would like to provide several suggestions for changes in text and additions to data analysis.39-41: It has been stated that reconstructing the lifecycle is the only way of understanding the nature of these microfossils. First of all, I would rephrase this to 'the most promising way', as there are always multiple approaches to comparing phenomena.

We agree with the reviewer's suggestion. The suggested changes have been made (line 41).

125: Please rephrase "under the environmental condition of early Earth" to "under experimental conditions possibly resembling the conditions of the Paleoarchean Eon". Now it sounds like the exact environmental conditions have been produced, which has already been debated in the discussion.

We agree with the reviewer's suggestion. The suggested changes have been made (line 127).

125: Please mention the fold change in size, the original size in numbers, and whether this change is statistically significant.

In the above sections of this document, we explained our reservations about presenting the exact number.

128: Have you found a difference in the relative percentages of modes of reproduction? In other words, is there a difference in percentage between forming internal daughter cells or a string of external daughter cells?

We explained our reservations about presenting the exact number above. But this has been extensively discussed in our accompaining manuscript. We want to reemphasize that the scope of this manuscript is restricted to comparing morphologies rather than providing a mechanistic explanation of the reproduction process.

151: A similar model for endocytosis has already been described in proliferating wall-less cells (Kapteijn et al., 2023). In the discussion, please compare your results with the observations made in that paper.

This is an oversight on our part. The manuscript suggested by the reviewer has now been added (line 154 & 155).

163: Please use another word for uncanny. We suggest using 'strong resemblance'.

We changed this according to the reviewers' suggestion (line 168).

433: Please elaborate on why the results are not shown. This sounds like a statement that should be substantiated further.

To observe growth and simultaneously image the cells, we conducted these experiments in chamber slides (2ml volume). Over time, we observed cells growing and breaking out of the salt crust (Fig. S86, S87 & Movie 22) and a gradual increase in the turbidity of the media. Although not quantitative, this is a qualitative indication of growth. We did not take precise measurements for several reasons. This sample is precious; it took us almost two years to solidify the biofilm completely, as shown in Fig. S84A. Hence, it was in limited supply, which prevented us from inoculating these salt crusts into large volumes of fresh media. Given a long period of starvation, these cells often exhibited a long lag phase (several days), and there wasn't enough volume to do OD measurements over time.

We also crushed the solidified biofilm with a sterile spatula before transferring it into the chamber slide with growth media. This resulted in debris in the form of small solid particles, which interfered with our OD measurements. These practical considerations made it challenging to determine the growth precisely. Despite these challenges, we measured an OD of 4 in some chamber slides after two weeks of incubation. Given that these measurements were done haphazardly, we chose not to present this data.

456: Could you please double-check whether the description is correct for the figure? 8C and 8D are part of Figure 8B, but this is stated otherwise in the description.

We thank the reviewer for pointing it out. It has now been rectified (line 461-472).

**Reviewer #2 (Recommendations For The Authors):**

We thank Reviewer #2 for carefully reading the manuscript and such an elaborate list of questions. The revisions suggested have definitely improved the quality of the manuscript. Here, we would like to address some of the questions that came up repeatedly below. One frequently asked question is regarding the letters denoting the individual figures within the images. For comparison purposes, we often reproduced previously published images. To maintain a consistent figure style, we often have to block the previous denotations with an opaque square and give a new letter.

The second question that appeared repeatedly below is the missing scale bars in some of the images within a figure. We often did not include a scale bar in the images when this image is an enlarged section of another image within the same figure.

Title: Please consider being more precise in the title. Microfossils are only one fossil group of "oldest life". Perhaps better: "On the nature of some microfossils in Archean rocks". (see also Line 37).

Authors’ response: The title conveys a broader message without quantitative insinuations. If our manuscript had been titled "On the nature of all known Archaean microfossils,” we should have agreed with the reviewer's suggestion and changed it to "On the nature of some microfossils in Archean rocks". As it is not, we respectfully decline to make this modification.

Abstract:Line 41: "one way", not "the only way"

We agree with the reviewer’s comment, and necessary changes have been made (line 41).

Introduction:Line 58f: "oldest sedimentary rock successions", not "oldest known rock formations". There are rocks of much older ages, but those are not well preserved due to metamorphic overprint, or the rocks are igneous to begin with. Minor issue: please note that "formations" are used as stratigraphic units, not so much to describe a rock succession in the field.

We agree with the reviewer’s comment and have made necessary changes (line 58).

Line 67: Microfossils are widely accepted as evidence of life. Please rephrase.

We agree with the reviewer’s comment, and necessary changes have been made.

Line 71 - 74: perhaps add a sentence of information here.

We agree with the reviewer’s comment, and necessary changes have been made (line 71).

Line 76: which "chemical and mineralogical considerations"?

This has been rephrased to “Apart from the chemical and δ^13^C-biomass composition” (line 76).

Line 84ff: This is a somewhat sweeping statement. Please remember that there are microbialites in such rocks that require already a high level of biofilm organization. The existence of cyanobacteria-type microbes in the Archean is also increasingly considered.

We are aware of literature that labeled the clusters of Archaean microfossils as biofilms and layered structures as microbialites or stromatolite-like structures. However, the use of these terms is increasingly being discouraged. A more recent consensus among researchers suggests annotating these structures simply as sedimentary structures, as microbially induced sedimentary structures (MISS).

We respectfully disagree with the reviewer’s comment that Archaean microfossils exhibit a high level of biofilm organization. We are not aware of any studies that have conducted such comprehensive research on the architecture of Archaean biofilms. We are not even certain if these clusters of Archaean cells could even be labeled as biofilms in the true sense of the term. We presently lack an exact definition of a biofilm. In our study, we do see sedimentation and bacteria and their encapsulation in cell debris. From a broader perspective, any such aggregation of cells enclosed in cell debris could be annotated as a biofilm. However, more in-depth studies show that biofilm is not a random but a highly organized structure. Different bacterial species have different biofilm architectures and chemical composition. The multispecies biofilms in natural environments are even more complex. We do agree with the reviewer that these structures could broadly be labeled as biofilms, but we presently lack a good, if any, understanding of the Archaean biofilm architecture.

Regarding the annotation of microfossils as cyanobacteria, we respectfully disagree with the reviewer. This is not a new concept. Many of the Archaean microfossils were annotated as cyanobacteria at the time of their discovery. This annotation is not without controversy. With the advent of genome-based studies, researchers are increasingly moving away from this school of thought.

Line 101ff: The conditions on early Earth are unknown - there are many varying opinions. Perhaps simply state that this laboratory model simulates an Archean Earth environment of these conditions outlined.

This is a good idea. We thank the reviewer for this suggestion, and we made appropriate changes.

Line 112: manuscript to be replaced by "paper"?

This change has been made (line 114).

Line 116: "spanned years" - how many years?

We now added the number of years in the brackets (line 118).

Results:Line 125: see comment for 101ff.

we made appropriate changes.

Figure 1: Caption: Please write out ICV the first time this abbreviation is used. Images: Note that some lettering appears to not fit their white labels underneath. (G, H, I, J0, and M).

We apologize; this is an oversight on our part. We now spell complete expansion of ICV, the first time we used this abbreviation.

We took these images from previously published work (references in the figure legend), so we must block out the previous figure captions. This is necessary to maintain a uniform style throughout the manuscript.

Line 152ff.: here would be a great opportunity to show in a graph the size variations of modern ICVs and to compare the variations with those in the fossil material.

In the above sections of this document, we explained our reservations about presenting the exact number.

Line 159f.: Fig.1K - what is to see here? Maybe a close-up or - better - a small sketch would help?

Fig. 1K shows the surface depressions formed during the vesicle formation. The surface characteristics of EM-P and microfossils is very similar.

Line 161f.: reference?

The paragraph spanning lines 159 to 172 discusses the morphological similarities between EM-P and SPF microfossils. We rechecked the reference no 35 (Delarue 2019). This is the correct reference. We do not see a mistake if the reviewer meant the reference to the figures.

Line 164ff.: A question may be asked, how many fossils of the Strelley Pool population would look similar to the "modeled" ones. Questions may rise in which way the environmental conditions control such morphology variations. Perhaps more details?

This relationship between the environmental conditions and the morphology is discussed extensively in our previous work (11).

Line 193: what is meant by "similar discontinuous distribution of organic carbon"?

This statement highlights similarities between EM-P and microfossils. The distribution of cytoplasm within the cells is not uniform. There are regions with and devoid of cytoplasm, which is quite unusual for bacteria. Some previous studies argued that this could indicate that these organic structures are of abiotic origin. Here, we show that EMP-like cells could exhibit such a patchy distribution of cytoplasm within the cell.

Line 218 - 291: The observations are very nice, however, the figures of fossil material in Figures 3 A, B, and C appear not to conform. Perhaps use D, E and I to K. Also, S48 does not show features as described here (see below).

We did not completely understand the reviewer’s question. As mentioned in the figure legend, both the microfossils and the cells exhibit string with spherical daughter cells within them. Moreover, there are also other similarities like the presence of hollow spherical structures devoid of organic carbon. We also saw several mistakes in the Fig. S48 legend. We have rectified them, and we thank the reviewer for pointing them out.

Line 293f: Title with "." at end?

This change has been made.

Line 298: predominantly in chert. In clastic material preservation of cells and pores is unlikely due to the common lack of in situ entombment by silica.

We rephrased this entire paragraph to better convey our message. Either way, we are not arguing that hollow pore spaces exist. As the reviewer mentioned, they will, of course, be filled up with silica. In this entire paragraph, we did not refer to hollow spaces. So, we are not entirely sure what the question was.

Line 324, 328-349: Please see below comments on the supplementary figures 51-62. Some of the interpretations of morphologies may be incorrect.

Please find our response to the reviewer’s comments on individual figures below.

Figure 5 A to D look interesting, however E to J appear to be unconvincing. What is the grey frame in D (not the white insert).

The grey color is just the background that was added during the 3D rendering process.

Figure 6 does not appear to be convincing. - Erase?

We did not understand the reviewer’s reservations regarding this figure. Images A-F within the figure show the gradual transformation of cells into honeycomb-like structures, and images G-J show such structures from the Archaean that are closely associated with microfossils. Moreover, we did not come up with this terminology (honeycomb-like). Previous manuscripts proposed it.

Line 379ff: S66 and 69, please see my comments below. Microfossils "were often discovered" in layers of organic carbon.

Please see our response below.

Line 393-403: Laminae? There are many ways to arrive at C-rich laminae, especially, if the material was compressed during burial. Basically, any type of biofilm would appear as laminae, if compressed. The appearance of thin layers is a mere coincidence. Note that the scale difference in S70, S73, as well as S83, is way too high (cm versus μm!) to allow any such sweeping conclusions. What are α- and β- laminations, the one described by Tice et al.? The arguments are not convincing.

We propose that cells be compressed to form laminae. We answered this question above about the differences in the scale bars. Yes, we are referring to α- and β- laminations described by Tice et al.

Figure 7: This is an interesting figure, but what are the arguments for B and C, the fossil material, being a membrane? Debris cannot be distinguished with certainty at this scale in the insert of C. B could also be a shriveled-up set of trichomes.

We agree with the reviewer that debris cannot be definitely differentiated. Traditionally, annotations given to microfossil structures such as biofilm, intact cells, or laminations were all based on morphological similarities with existing structures observed in microorganisms. Given that the structures observed in our study are very similar to the microfossil structures, it is logical to make such inferences. Scales in A & B match perfectly well. The structure in C is much larger, but, as we mentioned in reply to one of the reviewer’s earlier questions, some of the structures from natural environments could not be reproduced at scale in lab experiments. Working in a 2 ml chamber slides does impose some restrictions.

Figure 8: The figure does not show any honeycomb patterns. The "gaps" in the Moodies laminae are known as lenticular particles in biofilms. They form by desiccated and shriveledup biofilm that mineralizes in situ. Sometimes also entrapped gases induce precipitation. Note also that the modelled material shows a kind of skin around the blobs that are not present in the Moodies material.

We agree that entrapped gas bubbles could have formed lenticular gaps. In the manuscript, we did not discount this possibility. However, if that is the case, one should explain why we often find clumps of organic carbon within these gaps. As we presented a step-by-step transformation of parallel layers of cells into laminations, which also had similar lenticular gaps, we believe this is a more plausible way such structures could have formed. In the end, there could have been more than one way such structures could have been formed.

We do see the honeycomb pattern in the hollow gaps. Often, the 3D-rendering of the STED images obscures some details. Hence, in the figure legend, we referred to the supplementary figures also show the sequence of steps involved in the formation of such a pattern.

Line 405-417: During deposition of clastic sediment any hollow space would be compressed during burial and settling. It is rare that additional pore space (except between the graingrain-contacts) remains visible, especially after consolidation. The exception would be if very early silicification took place filling in any pore space. What about EPS being replaced by mineralic substance? The arguments are not convincing.

We are suggesting that EPS or cell debris is rapidly encrusted by cations from the surrounding environment and gets solidified into rigid structures. This makes it possible for the structures to be preserved in the fossil record. We believe that hollow structures like the lenticular gaps will be filled up with silica.

We do not agree with the reviewer’s comment that all biological structures will be compressed. If this is true, there should be no intact microfossils in the Archaean sedimentary structures, which is definitely not the case.

Line 419-430: Lithification takes place within the sediment and therefore is commonly controlled by the chemistry of pore water and chemical compounds that derive from the dissolution of minerals close by. Another aspect to consider is whether "desiccation cracks" on that small scale may be artefacts related to sample preparation (?).

We agree that desiccation cracks could have formed during the sample preparation for SEM imaging, as this involves drying the biofilms. However, we observed that the sample we used for SEM is a completely solidified biofilm (Fig. S84), so we expect little change in its morphology during drying. Moreover, visible cracks and pointy edges were also observed in wet samples, as shown in Fig. S87.

Line 432 - 439: Please see comments on the supplementary material below.

Please find our response to the reviewer’s comments on individual figures below.

Discussion:Line 477f: "all known microfossil morphologies" - is this a correct statement? Also, would the Archean world provide only one kind of "EM-P type"? Morphologies of prokaryote cells (spherical, rod-shaped, filamentous) in general are very simple, and any researcher of Precambrian material will appreciate the difficulties in concluding on taxonomy. There are papers that investigate putative microfossils in chert as features related to life cycles. Microfossil-papers commonly appear not to be controversial give and take some specific cases.

We made a mistake in using the term “all known microfossil morphologies.” We have now changed it to “all known spherical microfossils” from this statement (line 483). However, we do not agree with the statement that microfossil manuscripts tend not to be controversial. Assigning taxonomy to microfossils is anything but controversial. This has been intensely debated among the scientific community.

Line 494-496: This statement should be in the Introduction.

We agree with the reviewer’s comment. In an earlier version of the manuscript this statement was in the introduction. To put this statement in its proper context, it needs to be associated with a discussion about the importance of morphology in the identification of microfossils. The present version of the manuscript do not permit moving an entire paragraph into the introduction. Hence, we think making this statement in the discussion section is appropriate.

Line 484ff. The discussion on biogenicity of microfossils is long-standing (e.g., biogenicity criteria by Buick 1990 and other papers), and nothing new. In paleontology, modern prokaryotes may serve as models but everyone working on Archean microfossils will agree that these cannot correspond to modern groups. An example is fossil "cyanobacteria" that is thought to have been around already in the early Archean. While morphologically very similar to modern cyanobacteria, their genetic information certainly differed - how much will perhaps remain undisclosed by material of that high age.

Yes, we agree with the reviewer that there has been a longstanding conflict on the topic of biogenicity of microfossils. However, we have never come across manuscripts suggesting that modern microorganisms should only be used as models. If at all, there have been numerous manuscripts suggesting that these microfossils represent cyanobacteria, streptomycetes, and methanotrophs. Regarding the annotation of microfossils as cyanobacteria, we addressed this issue in one of the previous questions raised by the reviewer.

Line 498ff: Can the variation of morphology and sizes of the EM-Ps be demonstrated statistically? Line 505ff are very speculative statements. Relabeling of what could be vesicles as "microfossils" appears inappropriate. Contrary to what is stated in the manuscript, the morphologies of the Dresser Formation vesicles do not resemble the S3 to S14 spheroids from the Strelley Pool, the Waterfall, and Mt Goldsworthy sites listed in the manuscript. The spindle-shaped vesicles in Wacey et al are not addressed by this manuscript. What roles in mineral and element composition would have played diagenetic alteration and the extreme hydrothermal overprint and weathering typical for Dresser material? S59, S60 do not show what is stated, and the material derives from the Barberton Greenstone Belt, not the Pilbara.Please see the comments below regarding the supplementary images.

We did not observe huge variations in the cell morphology. Morphologies, in most cases, were restricted to spherical cells with intracellular vesicles or filamentous extensions. Regarding the sizes of the cells, we see some variations. However, we are reluctant to provide exact numbers. We have presented our reasons above.

We respectfully disagree with the reviewer’s comments. We see quite some similarities between Dresser formation microfossils and our cells. Not just the similarities, we have provided step-by-step transformation of cells that resulted in these morphologies. We fail to see what exactly is the speculation here. The argument that they should be classified as abiotic structures is based on the opinion that cells do form such structures. We clearly show here that they can, and these biological structures resemble Dresser formation microfossils more closely than the abiotic structures.

Regarding the figures S3-S14. We think they are morphologically very similar. Often, it's not just comparing both images or making exact reproductions (which is not possible). We should focus on reproducing the distinctive morphological features of these microfossils.

We agree with the reviewer that we did not reproduce all the structures reported by Wacey’s original manuscript, such as spherical structures. We are currently preparing another manuscript to address the filamentous microfossils. These spindle-like structures will be addressed in this subsequent work.

We agree with the reviewer, we often have difficulties differentiating between cells and vesicles. This is not a problem in the early stages of growth. During the log phase, a significant volume of the cell consists of the cytoplasm, with hollow vesicles constituting only a minor volume (Fig. 1B or S1A). During the later growth stages (Fig. 1E7F or S11), cells were almost hollow, with numerous daughter cells within them. These cells often resemble hollow vesicles rather than cells. However, given these are biologically formed structures, and one could argue that these vesicles are still alive as there is still a minimal amount of cytoplasm (Fig. S27). Hence, we should consider them as cells until they break apart to release daughter cells.

Regarding Figures S59 and S60, we did not claim either of these microfossils is from Pilbara Iron Formations. The legend of Figure S59 clearly states that these structures are from Buck Reef Chert, originally reported by Tice et al., 2006 (Figure 16 in the original manuscript). The legend of Figure S60 says these structures were originally reported by Barlow et al., 2018, from the Turee Creek Formation.

Line 546f and 552: The sites including microfossils in the Archean represent different paleoenvironments ranging from marine to terrestrial to lacustrine. References 6 and 66 are well-developed studies focusing on specific stratigraphic successions, but cannot include information covering other Archean worlds of the over 2.5 Ga years Archean time.

All the Archaean microfossils reported to date are from volcanic coastal marine environments. We are aware that there are rocky terrestrial environments, but no microfossils have been reported from these sites. We are unaware of any Archaean microfossils reported from freshwater environments.

Line 570ff: The statements may represent a hypothesis, but the data presented are too preliminary to substantiate the assumptions.

We believe this is a correct inference from an evolutionary, genomic, and now from a morphological perspective.

Figures:Please check all text and supplementary figures, whether scale bars are of different styles within the figure (minor quibble).S3 (no scale in C, D); S4, S5: Note that scale bars are of different styles.

We believe we addressed this issue above.

S6 D: depressions here are well visible - perhaps exchange with a photo in the main text? Note that scale bars are of different styles.

We agree that depressions are well visible in E. The same image of EM-P cell in E is also present in Fig. 1D in the main text.

S7: Scale bars should all be of the same style, if anyhow possible. Scale in D?

We believe we addressed this issue above.

S9: F appears to be distorted. Is the fossil like this? The figure would need additional indicators (arrows) pointing toward what the reader needs to see - not clear in this version. More explanation in the figure caption could be offered.

We rechecked the figure from the original publication to check if by mistake the figure was distorted during the assembly of this image. We can assure you that this is not the case. We are not sure what further could be said in the figure legend.

S13: What is shown in the inserts of D and E that is also visible in A and B? Here a sketch of the steps would help.

We did not understand the question.

S14: Scale in A, B?

We believe we addressed this issue above.

S15: Scales in A, E, C, D

We believe we addressed this issue above.

S16: scales in D, E, G, H, I, J?

We believe we addressed this issue above.

S17: "I" appears squeezed, is that so? If morphology is an important message, perhaps reduce the entire figure so it fits the layout. Note that labels A, B, C, and D are displaced.

As shown in several subsequent figures, the hollow spherical vesicles are compressed first into honeycomb-like structures, and they often undergo further compression to form lamination-like structures. Such images often give the impression that the entire figure is squashed, but this is not the case. If one examines the figure closely, you could see perfectly spherical vesicles together with laterally sqeezed structures. Regarding the figure labels, we addressed this issue above.

S18: The filamentous feature in C could also be the grain boundaries of the crystals. Can this be excluded as an interpretation? Are there microfossils with the cell membranes? That would be an excellent contribution to this figure. Note that scale bars are of different styles.

If this is a one-off observation, we could have arrived at the reviewer's opinion. But spherical cells in a “string of beads” configuration were frequently reported from several sites, to be discounted as mere interpretation.

S19: The morphologies in A - insert appear to be similar to E - insert in the lower left corner. The chain of cells in A may look similar to the morphologies in E - insert upper right of the image. B - what is to see here? D - the inclusions do not appear spherical (?). Does C look similar to the cluster with the arrow in the lower part of image E? Note that scale bars are of different styles (minor quibble). A, B, C, and D appear compressed. Perhaps reduce the size of the overall image?

The structures highlighted (yellow box) in C are similar to the highlighted regions in E—the agglomeration of hollow vesicles. It is hard to get understand this similarity in one figure. The similarities are apparent when one sees the Movie 4 and Fig. S12, clearly showing the spherical daughter cells within the hollow vesicle. We now added the movie reference to the figure legend.

S20: A appears not to contribute much. The lineations in B appear to be diagenetic. However, C is suitable. Perhaps use only C, D, E?

We believe too many unrecognizable structures are being labeled as diagenetic. Nevertheless, we do not subscribe to the notion that these are too lenient interpretations. These interpretations are justified as such structures have not been reported from live cells. This is the first study to report that cells could form such structures. As we now reproduced these structures, an alternate interpretation that these are organic structures derived from microfossils should be entertained.

S 21: Note that scale bars are of different styles.

We believe we addressed this issue above.

S22: Perhaps add an arrow in F, where the cell opened, and add "see arrow" in the caption? Is this the same situation as shown in C (white arrow)? What is shown by the white arrow in A? Note that scale bars are of different styles.

We did the necessary changes.

S23: In the caption and main text, please replace "&" with "and" (please check also the other figure captions, e.g. S24). Note that scale bars are of different styles. What is shown in F? A, D - what is shown here?

We replaced “&” with “and.”

S24: Note that scale bars are of different styles. Note that Wacey et al. describe the vesicles as abiotic not as "microfossils"; please correct in figure caption [same also S26; 25; 28].

We are aware of Prof. Dr. Wacey’s interpretations. We discuss it at length in the discussion section our manuscript. Based on the similarities between the Dresser formation structures and structures formed by EM-P, we contest that these are abiotic structures.

S25: Appears compressed; note different scale bars.

We believe we addressed this issue above.

S28: The label in B is still in the upper right corner; scale in D? What is to see in rectangles (blue and red) in A, B? In fossil material, this could be anything.

These figures are taken from a previous manuscript cited in the figure legend. We could not erase or modify these figures.

S33: "L"ewis; G appears a bit too diffuse - erase? Note that scale bars are of different styles.

We believe we addressed this issue above.

S34: This figure appears unconvincing. Erase?

There are considerable similarities between the microfossils and structures formed by EM-P. If the reviewer expands a bit on what he finds unconvincing, we can address his reservations.

S35: It would be more convincing to show only the morphological similarities between the cell clusters. B and C are too blurry to distinguish much. Scales in D to F and in sketches? A appears compressed (?).

We rechecked the original manuscript to see if image A was distorted while making this figure, but this is not the case. Regarding B & C, cells in this image are faint as they are hollow vesicles and, by nature, do not generate too much contrast when imaged with a phase-contrast microscope. There are some limitations on how much we can improve the contrast. We now added scale bars for D-I. Similarly, faint hollow vesicles can be seen in Fig. S21 C & D, and Fig. 3H.

S36: Very nice; in B no purple arrow is visible. Note that scale bars are of different styles. S37 and S36 are very much the same - fuse, perhaps?

We are sorry for the confusion. There are purple arrows in Fig. S37B-D.

S38: this is a more unconvincing figure - erase?

Unconvincing in wahy sense. There are considerable similarities between the microfossils and structures formed by EM-P. If the reviewer expands a bit on what he finds unconvincing, we can address his reservations.

S39: white rectangle in A? Arrow in A? Note that scale bars are of different styles.

These are some of the unavoidable remnants from the image from the original publication.

S40: in F: CM, V = ?; Note that scale bars are of different style.

It’s an oversite on our part. We now added the definitions to the figure legaend. We thank the reviewer for pointing it out.

S41: Rectangles in D, E, F, G can be deleted? Scales and labels missing in photos lower right.

Those rectangles are added by the image processing software to the 3Drendered images. Regarding the missing scale bars in H & I they are the magnified regions of F. The scale bar is already present in F.

S42: appears compressed. G could be trimmed. Labels too small; scale in G?

This is a curled-up folded membrane. We needed to lower the resolution of some images to restrict the size of the supplement to journal size restrictions. It is not possible to present 85 figures in high resolution. But we assure you that the image is not laterally compressed in any manner.

S43: This figure appears to be unconvincing. Reducing to pairing B, C, D with L, K? Spherical inclusions in B? Scales in E to G? Similar in S44: A, B, E only? Note that scale bars are of different styles.

Figures I to K are important. They show not just the morphological similarities but also the sequence of steps through which such structures are formed. We addressed the issue of the scale bars above.

S45: A, B, and C appear to show live or subrecent material. How was this isolated of a rock? Note that scale bars are of different styles.

It is common to treat rocks with acids to dissolve them and then retrieve organic structures within them. This technique is becoming increasingly common. The procedure is quite extensively discussed in the original manuscript. We don’t see much differences in the scale bars of microfossils and EM-P cells, they are quite similar.

S46: A: what is to see here? Note that scale bars are of different styles.

There are considerable similarities between the folded fabric like organic structures with spherical inclusions and structures formed by EM-P. If the reviewer expands a bit on what he finds unconvincing, we can address his reservations.

S47: Perhaps enlarge B and erase A. Note that scale bars are of different styles.S48: Image B appears to show the fossil material - is the figure caption inconsistent? There are no aggregations visible in the boxes in A. H is described in the figure caption but missing in the figure. Overall, F and G do not appear to mirror anything in A to E (which may be fossil material?).S51; S52 B, C, E; S53: these figures appear unconvincing - erase?

Unconvincing in what sense? The structures from our study are very similar to the microfossils.

S54: North "Pole; scale bars in A to C = ?

These figures were borrowed from an earlier publication referenced in the figure legend. That is the reason for the differences in the styles of scale bars.

S55: D and E appear not to contribute anything. Perhaps add arrow(s) and more explanation? Check the spelling in the caption, please.

D & E show morphological similarities between cells from our study and microfossils (A).

S56: Hexagonal morphologies may also be a consequence of diagenesis. Overall, perhaps erase this figure?

I certainly agree that could be one of the reasons for the hexagonal morphologies. Such geometric polygonal morphologies have not been observed in living organisms. Nevertheless, as you can see from the figure, such morphologies could also be formed by living organisms. Hence, this alternate interpretation should not be discounted.

S57: The figure caption needs improvement. Please add more description. What show arrows in A, what are the numbers in A? What is the relation between the image attached to the right side of A? Is this a close-up? Note that scale bars are of different styles.

We expanded a bit on our original description of the figure. However, we request the reviewer to keep in mind that the parts of the figure are taken from previous publication. We are not at liberty to modifiy them, like removing the arrows. This imposes some constrains.

S58: There are no honeycomb-shaped features visible. What is to see here? Erase this figure?

Clearly, one can see spherical and polygonal shapes within the Archaean organic structures and mat-like structures formed by EM-P.

S59 and S60: What is to see here? - Erase?

Clearly, one can see spherical and polygonal shapes within the Archaean organic structures and mat-like structures formed by EM-P in Fig. S59. Further disintegration of these honeycomb shaped mats into filamentous struructures with spherical cells attached to them can be seen in both Archaean organic structures and structures formed by EM-P.

S61: This figure appears to be unconvincing. B and F may be a good pairing. Note that scale bars are of different styles.

There are considerable similarities between the microfossils and structures formed by EM-P. If the reviewer expands a bit on what he finds unconvincing, we might be able to address his reservations.

S62: This figure appears to be unconvincing - erase?

There are considerable similarities between the microfossils and structures formed by EM-P. If the reviewer expands a bit on what he finds unconvincing, we might be able to address his reservations.

S66: This figure is unconvincing - erase?

There are considerable similarities between the microfossils and structures formed by EM-P. If the reviewer expands a bit on what he finds unconvincing, we might be able to address his reservations.

S68: Scale in B, D, and E?

Image B is just a magnified image of a small portion of image A. Hence, there is no need for an additional scale bar. The same is true for images D and E.

S69: This figure appears to be unconvincing, at least the fossil part. Filamentous features are visible in fossil material as well, but nothing else.

We are not sure what filamentous features the reviewer is referring to. Both the figures show morphologically similar spherical cells covered in membrane debris.

S70 [as well as S82]: Good thinking here, but scales differ by magnitudes (cm to μm). Erase this figure? Very similar to Figure S73: Insert in C has which scale in comparison to B? Note that scale bars are of different styles.

We realize the scale bars are of different sizes. In our defense, our experiments are conducted in 1ml volume chamber slides. We don’t have the luxury of doing these experiments on a scale similar to the natural environments. The size differences are to be expected.

S71: Scale in E?

Image E is just a magnified image of a small portion of image D. Hence, we believe a scale bar is unnecessary.

S72: Scale in insert?

The insert is just a magnified region of A & C

S75: This figure appears to be unconvincing. This is clastic sediment, not chert. Lenticular gaps would collapse during burial by subsequent sediment. - Erase?

Regarding the similarities, we see similar lenticular gaps within the parallel layers of organic carbon in both microfossils, and structures formed by EM-P.

S76: A, C, D do not look similar to B - erase? Similar to S79, also with respect to the differences in scale. Erase?

Regarding the similarities, we see similar lenticular gaps within the parallel layers of organic carbon in both microfossils, and structures formed by EM-P. We believe we addressed the issue of scale bars above.

S80: A appears to be diagenetic, not primary. Erase?

These two structures share too many resemblances to ignore or discount just as diagenic structures - Raised filamentous structures originate out of parallel layers of organic carbon (laminations), with spherical cells within this filamentous organic carbon.

S85: What role would diagenesis play here? This figure appears unconvincing. Erase?

We do believe that diagenesis plays a major role in microfossil preservation. However, we also do not suscribe to the notion that we should by default assign diagenesis to all microfossil features. Our study shows that there could be an alternate explanation to some of the observations.

S86 and S87: These appear unconvincing. What is to see here? Erase?

The morphological similarities between these two structures. Stellarshaped organic structures with strings of spherical daughter cells growing out of them.

S88: Does this image suggest the preservation of "salt" in organic material once preserved in chert?

That is one inference we conclude from this observation. Crystaline NaCl was previously reported from within the microfossil cells.

S89: What is to see here? Spherical phenomena in different materials?

At present, the presence of honeycomb-like structures is often considered to have been an indication of volcanic pumice. We meant to show that biofilms of living organisms could result in honeycomb-shaped patterns similar to volcanic pumice.

ReferencesPlease check the spelling in the references.

We found a few references that required corrention. We now rectified them.

References

(1) Orange F, Westall F, Disnar JR, Prieur D, Bienvenu N, Le Romancer M, et al. Experimental silicification of the extremophilic archaea pyrococcus abyssi and methanocaldococcus jannaschii: Applications in the search for evidence of life in early earth and extraterrestrial rocks. Geobiology. 2009;7(4).

(2) Orange F, Disnar JR, Westall F, Prieur D, Baillif P. Metal cation binding by the hyperthermophilic microorganism, Archaea Methanocaldococcus Jannaschii, and its effects on silicification. Palaeontology. 2011;54(5).

(3) Errington J. L-form bacteria, cell walls and the origins of life. Open Biol. 2013;3(1):120143.

(4) Cooper S. Distinguishing between linear and exponential cell growth during the division cycle: Single-cell studies, cell-culture studies, and the object of cell-cycle research. Theor Biol Med Model. 2006;

(5) Mitchison JM. Single cell studies of the cell cycle and some models. Theor Biol Med Model. 2005;

(6) Kærn M, Elston TC, Blake WJ, Collins JJ. Stochasticity in gene expression: From theories to phenotypes. Nat Rev Genet. 2005;

(7) Elowitz MB, Levine AJ, Siggia ED, Swain PS. Stochastic gene expression in a single cell. Science. 2002;

(8) Strovas TJ, Sauter LM, Guo X, Lidstrom ME. Cell-to-cell heterogeneity in growth rate and gene expression in Methylobacterium extorquens AM1. J Bacteriol. 2007;

(9) Knoll AH, Barghoorn ES. Archean microfossils showing cell division from the Swaziland System of South Africa. Science. 1977;198(4315):396–8.

(10) Sugitani K, Grey K, Allwood A, Nagaoka T, Mimura K, Minami M, et al. Diverse microstructures from Archaean chert from the Mount Goldsworthy–Mount Grant area, Pilbara Craton, Western Australia: microfossils, dubiofossils, or pseudofossils? Precambrian Res. 2007;158(3–4):228–62.

(11) Kanaparthi D, Lampe M, Krohn JH, Zhu B, Hildebrand F, Boesen T, et al. The reproduction process of Gram-positive protocells. Sci Rep. 2024 Mar 25;14(1):7075.